# MINIBATCH OPTIMAL TRANSPORT AND PERPLEXITY BOUND ESTIMATION IN DISCRETE FLOW MATCHING

## ABSTRACT

Discrete flow matching, a recent framework for modeling categorical data, has shown competitive performance with autoregressive models. However, unlike continuous flow matching, the rectification strategy cannot be applied due to the stochasticity of discrete paths, necessitating alternative methods to minimize state transitions. We propose a dynamic-optimal-transport-like minimization objective and derive its Kantorovich formulation for discrete flows with convex interpolants, where transport cost depends solely on inter-state similarity and can be optimized via minibatch strategies. In the case of bag-of-words (BoW) sourced flows, we show that such methods can reduce the number of transitions up to 8 times (1024 to 128) to reach the same generative perplexity without compromising diversity. Additionally, path nondeterminism in discrete flows precludes an instantaneous change-of-variables analogue, preventing precise probability estimation available to continuous flows. We therefore propose two upper bounds on perplexity, enabling principled training, evaluation and model comparison. Finally, we introduce Multimask Flow which outperforms masked flows in generative perplexity without sacrificing diversity, particularly when utilizing minibatch Optimal Transport.

## 1 INTRODUCTION

Modeling data distributions is central to machine learning. For continuous data, diffusion and flow models (Sohl-Dickstein et al., 2015; Ho et al., 2020; Song et al., 2020b; Lipman et al., 2023) have shown impressive results in generation and density estimation (Song et al., 2021a;c; Chen et al., 2018). Rectified flows particularly excel by enabling high-quality generation with few integration steps. However, these continuous models lag behind autoregressive models on categorical data (Chen et al., 2023; Gulrajani & Hashimoto, 2024; Li et al., 2022; Dieleman et al., 2022; Strudel et al., 2022).

To address this, recent work has developed discrete diffusion (Austin et al., 2021; Campbell et al., 2022; Meng et al., 2022; Lou et al., 2024; Sahoo et al., 2024; Ou et al., 2024; Shi et al., 2024) and discrete flow models (Campbell et al., 2024; Gat et al., 2024), better suited for categorical data. These models can accelerate generation and, unlike autoregressive models, naturally enable infilling. We focus on discrete flow matching (DFM), which expands the design space beyond discrete diffusion by allowing arbitrary couplings and inner dynamics. While discrete and continuous flows share similarities that facilitate adapting continuous flow matching results, fundamental differences remain. These arise primarily from the nonexistence of a DFM formulation with deterministic sample paths.

A major implication of non-deterministic sample paths is that we cannot use the rectification strategy from Liu et al. (2022). Since paths in discrete flows are sequences of states, we explore minimizing the number of jumps between states, which can be interpreted as the discrete analogue of path length minimization. Using similarity measures between states, we minimize jumps weighted by dissimilarity, yielding a weighted path-length-oriented dynamic formulation of optimal transport (OT) for discrete flow matching. We derive its Kantorovich formulation, where the cost function depends only on the similarity measure and can be optimized using minibatch strategies (Tong et al., 2024; Fatras et al., 2021). This gives a categorical Benamou-Brenier-type theorem when conditional flows are convex interpolants, i.e., the categorical equivalent of shortest-path continuous flows. When the similarity measure in the dynamic formulation is the discrete metric, the cost function in the Kantorovich formulation becomes the Hamming distance. When the similarity measure is the $L_2$ norm, the cost function is also $L_2$, mirroring the continuous case.

An additional implication of stochastic crossing paths in DFM is that we cannot use an equivalent of the instantaneous change of variable formula (Chen et al., 2018) for probability estimation. Thus, other approaches are needed for estimating the perplexity. Inspired by bounds in Lou et al. (2024); Haxholli et al. (2025), we derive two upper bounds on perplexity for discrete flow matching. These bounds enable theoretically grounded training, model evaluation and comparison with other methods.

Experiments show that minibatch-OT significantly reduces jumps in small-scale experiments and, in realistic settings (GPT2-sized model on OWT), reduces inference steps up to 8-fold (from 1024 to 128) to achieve the same generative perplexity. We also introduce multimask flow (DFM-MM), which outperforms masked DFM in terms of generative perplexity without sacrificing diversity, in particular when combined with minibatch-OT. Finally, we demonstrate that our derived bounds enable comparisons with autoregressive and discrete diffusion models.

In summary, the main **contributions** of this paper include:

- We formulate a weighted path-length-oriented dynamic OT objective that minimizes dissimilarity-weighted jumps between states. We derive its Kantorovich formulation for convex interpolant flows, establishing a categorical Benamou-Brenier-type theorem.
- We extend two discrete diffusion bounds to DFM, providing principled training objectives and enabling comparisons with autoregressive and discrete diffusion models.
- We show minibatch OT reduces inference steps up to 8-fold (1024 to 128) while maintaining generative perplexity on GPT2-scale models. Finally, we introduce multimask flow (DFM-MM), which surpasses masked DFM models in generative perplexity without compromising diversity, with further gains achieved when applying OT.

## 2 PRELIMINARIES AND NOTATION

A summary of Discrete Flow Matching is provided below. While the following preliminary is self-contained, we also provide an introduction to the discrete diffusion framework in Appendix D.

### 2.1 DISCRETE FLOW MATCHING

To expand the design space of discrete diffusion models, Campbell et al. (2024); Gat et al. (2024) introduce discrete flow matching. We follow the approach and notation of Gat et al. (2024). In discrete sequence modeling, a sequence (state) $x$ consists of $L$ elements $(x^1, x^2, \ldots, x^L)$. Each position $i$ contains an element $x^i$ from a vocabulary $\mathcal{V} = [V] = \{1, \ldots, V\}$ of size $V$. Thus, the set of possible sequences is $\mathcal{D} = \mathcal{V}^L$. Two sequences are neighbors if they differ in only one position.

We denote with $p^i(x^i)$ the marginal of $p$ at position $i$, i.e., $p^i(x^i) = \sum_{x^{-i}} p(x)$, where $x^{-i} = (x^1 \ldots, x^{i-1}, x^{i+1}, \ldots x^L)$. The following delta function notation will be particularly useful,

$$\delta_y(x) = \prod_{i=1}^{N} \delta_{y^i}(x^i), \text{ where } \delta_{y^i}(x^i) = \begin{cases} 1 & \text{if } x^i = y^i \\ 0 & \text{if } x^i \neq y^i \end{cases}. \tag{1}$$

### 2.1.1 PROBABILITY FLOWS AND VELOCITIES

In discrete flow matching (Gat et al., 2024), the goal is to acquire a flow $p_t(z) : [0, 1] \times [V]^L \to [0, 1]$ constrained by $\sum_{z \in [V]^L} p_t(z) = 1$ that transforms source (reference) distributions $X_0 \sim p$ to target (data) distributions $X_1 \sim q$. The flow is completely defined by the choice of a probability velocity $u_t(x) : [0, 1] \times [V]^L \to \mathbb{R}^{L \times V}$, such that $u_t(z) = (u_t^1(z), \ldots, u_t^i(z), \ldots, u_t^L(z))$ and $u_t^i : [0, 1] \times [V]^L \to \mathbb{R}^V$, where $u_t^i(z)[x^i \neq z^i] \geq 0$ and $\sum_{x^i \in [V]} u_t^i(z)[x^i] = 0$, for each $i$. The update rule of the probability over states when going from time $t$ to $t + \epsilon$ is defined independently for each position in the sequence as follows $p_{t+\epsilon|t}^i(x^i|x_t) = \delta_{x_t^i}(x^i) + \epsilon u_t^i(x^i, x_t)$, where we used $u_t^i(x^i, z) := u_t^i(z)[x^i]$. Therefore, we can see that as in the framework of Markov chains, the probability over the states in the next step depends solely on the current state, and that $u_t$ plays a similar role to a transition-rate matrix $Q_t$, completely determining the flow. As such, if we approximate the probability velocity $u_t(z)$ using a neural network $u_t(z; \theta) : [0, 1] \times [V]^L \to \mathbb{R}^{L \times V}$, we can sample from $p$ and generate data from $q$, using the previous update rule. Before modeling the

probability velocity $u_t(z)$ however, one must first design an appropriate flow $p_t(z)$ that has a suitable, practically learnable corresponding $u_t(z)$.

### 2.1.2 CONDITIONAL PROBABILITY FLOWS

Since at time $t = 0$ and $t = 1$ we must have $p_0 = p$ and $p_1 = q$ respectively, we are already restricted regarding the endpoints of the flow. A trivial way to satisfy such constraints is to define

$$p_t(x) = \sum_{x_0, x_1 \in \mathcal{D}} p_t(x|x_0, x_1)\pi(x_0, x_1), \tag{2}$$

where $p_0(x|x_0, x_1) = \delta_{x_0}(x)$, $p_1(x|x_0, x_1) = \delta_{x_1}(x)$ and $\pi(X_0, X_1)$ is an arbitrary joint distribution of $X_0, X_1$ satisfying the marginals constraints $p(x) = \sum_{y \in \mathcal{D}} \pi(x, y)$, $q(y) = \sum_{x \in \mathcal{D}} \pi(x, y)$. Since the probability velocities update the probability independently for each position, it is natural to define $p_t(x|x_0, x_1)$ independently for each dimension as in Gat et al. (2024):

$$p_t(x|x_0, x_1) = \prod_{i=1}^{N} p_t^i(x^i|x_0, x_1), \tag{3}$$

where $p_t^i(x^i|x_0, x_1) = (1 - k_t)\delta_{x_0^i}(x^i) + k_t\delta_{x_1^i}(x^i)$, with $k_0 = 0, k_1 = 1$ and increasing $k_t$. (4)

It is clear that this definition of $p_t(x|x_0, x_1)$ satisfies the conditions $p_0(x|x_0, x_1) = \delta_{x_0}(x)$ and $p_1(x|x_0, x_1) = \delta_{x_1}(x)$. In addition, Gat et al. (2024) show that component $i$ of the conditional probability velocity $u_t(x, z|x_0, x_1)$ corresponding to the flow defined in Equations (3) and (4) is

$$u_t^i(x^i, z|x_0, x_1) = \frac{\dot{k}_t}{1 - k_t}\left[\delta_{x_1^i}(x^i) - \delta_{z^i}(x^i)\right]. \tag{5}$$

Furthermore, they show that the probability velocity corresponding to the unconditional flow $p_t(z)$ can be written as

$$u_t^i(x^i, z) = \sum_{x_0, x_1 \in \mathcal{D}} u_t^i(x^i, z|x_0, x_1)p(x_0, x_1|z)dx_0dx_1, \tag{6}$$

which in the case of Equations (4) and (5) implies, $u_t^i(x^i, z) = \frac{\dot{k}_t}{1 - k_t}\left[p_{1|t}^i(x^i|z) - \delta_z(x^i)\right]$. One then approximates $u_t^i(x^i, z)$ by simply modeling $p_{1|t}^i(x^i|z)$ with a neural network $p_{1|t}^i(x^i|z; \theta)$ using the cross entropy loss $L$,

$$-\mathbb{E}_{t \sim U(0,1)}\mathbb{E}_{x_0, x_1 \sim \pi(x_0, x_1)}\mathbb{E}_{x_t \sim p_{t|0}(\cdot|x_0, x_1)}\sum_{i=1}^{L}\log p_{1|t}^i(x_1^i|x_t; \theta). \tag{7}$$

It should be mentioned that in Gat et al. (2024), the definition of $p_t^i(x^i|x_0, x_1)$ is given in a more general form, but here we focus on this specific case for the sake of simplicity and since this formulation corresponds to shortest path conditional flows in the continuous framework, that is $X_t = (1 - t)X_0 + tX_1$.

### 2.1.3 SOURCE AND TARGET DISTRIBUTIONS

As mentioned, points $X_0$ and $X_1$ are sampled from a joint distribution $\pi(x, y)$, i.e. $(X_0, X_1) \sim \pi(X_0, X_1)$, satisfying the marginals constraints $p(x) = \sum_{y \in \mathcal{D}} \pi(x, y)$, $q(y) = \sum_{x \in \mathcal{D}} \pi(x, y)$. As a special case, the training pairs $X_0$ and $X_1$ can be sampled independently, $(X_0, X_1) \sim p(X_0)q(X_1)$. Common instantiations of source distribution $p$ are:
(i) adding a special token value often referred to as a *mask* token, denoted here by $m$, and setting the source distribution to contain only the fully masked sequence, i.e., $(X_0, X_1) = ((m, \ldots, m), X_1)$.
(ii) using uniform distribution over $\mathcal{D}$, which is equivalent to drawing each $x^i$ independently to be some value in $[V]$ with equal probability, denoted $p_u(x^i)$.

# 3 TRANSITION REDUCTION OBJECTIVES IN DISCRETE FLOW MATCHING

We first introduce our dynamic optimal transport formulation for discrete flows with per-position convex interpolants and derive its equivalent static (Kantorovich) formulation. We then explain how our formulation and equivalence result differ from and improve upon classical OT results in this specific flow setting. Finally, since masked flows (Gat et al., 2024) achieve the best practical performance but cannot leverage OT, we introduce multimasked flows. These maintain the performance and time-independent predictive probabilities of masked flows while enabling OT enhancement.

## 3.1 DYNAMIC OT FORMULATION OF DISCRETE FLOWS WITH CONVEX INTERPOLANTS AND ITS STATIC EQUIVALENT

A central aim in flow-matching research is to cut the number of steps needed for high-quality generation, that is, to simplify the trajectories from the source to the target distribution. Such simplified paths are easier for neural networks to model and, empirically, yield higher-quality models. One principled way to simplify these paths is to minimize kinetic energy, as in the dynamic formulation of optimal transport:

$$\int_0^1 \int \frac{1}{2} p(x_t) \|v_t(x_t)\|^2 dx_t dt, \tag{8}$$

with endpoints fixed at the source and target distributions. This objective is closely related to minimizing expected path length, but the squared speed penalizes large velocities more strongly. Moreover, by the Benamou–Brenier theorem (Benamou & Brenier, 2000; Tong et al., 2024), the infimum of Equation (8) equals the infimum of the Kantorovich transport with quadratic cost,

$$\int c(x_0, x_1) \pi(x_0, x_1) dx_0 dx_1, \text{ where } c(x_0, x_1) = \|x_0 - x_1\|^2, \tag{9}$$

taken over all couplings $\pi$ with marginals $p_0$ and $p_1$. We observe that minimizing $\|v_t(x_t)\|^2 = v_{1,t}^2 + ... + v_{d,t}^2$ corresponds to minimizing the instantaneous movement of particles from their current positions. In the discrete flow setting, there is a natural analogue: we seek to minimize the expected outflowing mass $u_t^i(x^i, x_t)$ for transitions where $x^i \neq x_t^i$. Equivalently, this amounts to maximizing $u_t^i(x_t^i, x_t)$, favoring trajectories where the mass predominantly stays in place rather than flowing between states.

Therefore, the dynamic formulation for DFM minimizes:

$$\int_0^1 \sum_{x_t} \frac{1}{2} p(x_t) \left[ \sum_{i=1}^L \left( \sum_{x^i \neq x_t^i} u_t^i(x^i, x_t) - u_t^i(x_t^i, x_t) \right) \right] dt = \int_0^1 \sum_{x_t} p(x_t) \sum_{i=1}^L \sum_{x^i \neq x_t^i} u_t^i(x^i, x_t) dt, \tag{10}$$

where $x^i \neq x_t^i$ denotes $x^i \in \mathcal{V} \setminus x_t^i$ and $x_t \in \mathcal{D}$. We prove this equals the Kantorovich formulation in Equation (9) when $c(x_0, x_1)$ is the Hamming distance ($d_H$) between sequences (Corollary 1). The categorical dynamic formulation above treats all tokens equally, yet in practice tokens have varying similarities reflected in their embeddings. We should weight the outflow by token similarity, penalizing transitions to dissimilar states more heavily. Moreover, for large vocabularies, sequences sampled from the source distribution $p(x_0)$ and the target data distribution $q(x_1)$ likely share few matching positions. Consequently, optimizing this expression using OT-minibatches as in Tong et al. (2024), should not offer substantial improvements in realistic DFM settings.

For these reasons, we define the categorical dynamic objective more generally as follows:

$$\int_0^1 \sum_{x_t} p(x_t) \left[ \sum_{i=1}^L \sum_{x^i} u_t^i(x^i, x_t) s(x^i, x_t^i) \right] dt = \int_0^1 \sum_{x_t} p(x_t) \left[ \sum_{i=1}^L \sum_{x^i \neq x_t^i} u_t^i(x^i, x_t) s(x^i, x_t^i) \right] dt, \tag{11}$$

where $s(x^i, x_t^i) \geq 0$ is a similarity measure between two tokens $x^i$ and $x_t^i$ that is symmetric and satisfies $s(a, a) = 0$. Our previous formulation in Equation (10) used the discrete metric $s(x^i, x_t^i) = 1 - \delta_{x_t^i}(x^i)$. Another natural choice is the squared $L_2$ distance between token embeddings: $s(x^i, x_t^i) = \|e_m(x^i) - e_m(x_t^i)\|^2$. For any choice of similarity measure (typically a metric), there exists a corresponding Kantorovich formulation with a cost function determined by that measure.

**Theorem 3.1.** *Let $\pi(x_0, x_1)$ be the joint distribution of $x_0$ and $x_1$, and let $p_t$ be a flow defined as in Equations (2, 3, 4) that transforms $p = \int \pi(x_0, x_1)dx_1$ into $q = \int \pi(x_0, x_1)dx_0$. In this setting, the dynamic formulation given in Equation (11) equals the Kantorovich formulation:*

$$\int_0^1 \sum_{x_t} p(x_t) \sum_{i=1}^L \sum_{x^i \neq x_t^i} u_t^i(x^i, x_t)s(x^i, x_t^i)dt = \sum_{x_0, x_1} c(x_0, x_1)\pi(x_0, x_1), \quad (12)$$

*where the cost function is $c(x_0, x_1) = \sum_{i=1}^L s(x_0^i, x_1^i)$.*

We provide a proof in Appendix A.1. The theorem trivially extends to position-specific schedulers: $p_t^i(x^i|x_0, x_1) = (1 - k_t^i)\delta_{x_0^i}(x^i) + k_t^i\delta_{x_1^i}(x^i)$. Algorithm 1 describes training with minibatch OT for optimizing the Kantorovich formulation. For the categorical dynamic formulation (10), the corresponding Kantorovich cost function is the Hamming distance $d_H$:

**Corollary 3.2.** *If in Theorem 3.1 we choose $s(x^i, x_t^i) = 1 - \delta_{x_t^i}(x^i)$ then $c(x_0, x_1) = \sum_{i=1}^L s(x_0^i, x_1^i) = \sum_{i=1}^L (1 - \delta_{x_0^i}(x_1^i)) = \sum_{i=1}^L \delta_{x_0^i \neq x_1^i} = d_H(x_0, x_1)$.*

Interestingly, if $s(x^i, x_t^i) = \|e_m(x^i) - e_m(x_t^i)\|^2$, the cost function becomes the $L_2$ norm between sequence embeddings, mirroring continuous flow matching:

**Corollary 3.3.** *If in Theorem 3.1 we choose $s(x^i, x_t^i) = \|e_m(x^i) - e_m(x_t^i)\|^2$ then $c(x_0, x_1) = \sum_{i=1}^L \|e_m(x_1^i) - e_m(x_0^i)\|^2$ that is $c(x_0, x_1) = \|e_m(x_1) - e_m(x_0)\|^2$.*

### 3.2 Differences with Classical Optimal Transport Results

We work in a restricted setting: per-position discrete flows on sequence space, specifically with convex interpolants. This structured approach enables results for a broader class of cost functions than classical Benamou–Brenier theorem. Within this framework, we introduce a general similarity function $s(\cdot, \cdot)$ and define our dynamic objective using path length rather than kinetic energy, that is, discrete jump rates and weighted jump counts instead of quadratic velocity fields. This naturally accommodates more general discrepancy measures beyond $\| \cdot \|^2$. Our restriction pays off as our Benamou–Brenier-type theorem holds for *arbitrary* similarity functions $s$, not just squared Euclidean distances. Since the flow dynamics are fully determined by the coupling $\pi$, we optimize only over couplings, not entire flows. For each $\pi$, the induced convex-interpolant flow yields a dynamic loss that exactly equals the static Kantorovich loss for the same $\pi$. Taking the infimum over all couplings is therefore purely formal since equality holds pointwise before optimization.

### 3.3 Multi-mask Flows

Standard masked DFM (Gat et al., 2024) uses a Dirac distribution at the fully masked sequence as its source, admitting only the trivial coupling. To enable meaningful couplings, we introduce multimask flow (DFM-MMLM) with $V_s = 50, 257$ special mask tokens $m_1, m_2, ..., m_{V_s}$, all distinct from data tokens. Source sequences are uniformly sampled combinations of these masks. Our total vocabulary thus contains $V = V_s + V_d$ tokens: $V_d = 50, 257$ data tokens $x_1, ..., x_{V_d}$ from our tokenizer and $V_s = 50, 257$ mask tokens. At $t = 0$, only mask tokens appear while data tokens are assigned zero probability. This design provides two advantages: denoising probabilities remain time-independent as in masked flow (see Appendix A.6), and mask embeddings are fully unrestricted since they are untied from data-token embeddings. Effectively, we create a "fictitious grid" where each $L$-length sequence carries mass $\frac{1}{50257^L}$. The flow transports this mass to the data grid, enabling minibatch OT.

## 4 Upper Bounds on the Perplexity in Discrete Flow Matching

Perplexity is a key metric for language models, making it essential to calculate or bound it in the DFM framework. While Appendix A.5 provides a precise but computationally intractable formula, the next two subsections present practical bounds. These bounds serve as both principled training objectives and effective evaluation metrics, offering intrinsic and objective assessment.

## 4.1 An Upper Bound on the Perplexity

To derive the first upper bound, we first provide an expression for the KL divergence between the end distributions $\bar{p}_1$ and $\bar{q}_1$ of two flows $\bar{p}_t$ and $\bar{q}_t$. To derive this expression, we extend the approaches of Opper & Sanguinetti (2007, Equation 3) and Haxholli et al. (2025) to DFM models.

**Theorem 4.1.** *For two discrete flows $\bar{p}_t$ and $\bar{q}_t$ with corresponding probability velocities $v_t(x^i, x_t)$ and $w_t(x^i, x_t)$, the following equality holds*

$$D_{KL}(\bar{q}_1\|\bar{p}_1) = \int_0^1 \sum_{x_t} \bar{q}_t(x_t) \sum_{i=1}^{L} \sum_{x^i \neq x_t^i} \left( w_t^i(x^i, x_t) \log \frac{w_t^i(x^i, x_t)}{v_t^i(x^i, x_t)} + v_t^i(x^i, x_t) - w_t^i(x^i, x_t) \right) dt$$

$$- \int_0^1 \sum_{x_t} \bar{q}_t(x_t) \sum_{i=1}^{L} \sum_{x^i \neq x_t^i} \left( \tilde{w}_t^i(x^i, x_t) \log \frac{\tilde{w}_t^i(x^i, x_t)}{\tilde{v}_t^i(x^i, x_t)} + \tilde{v}_t^i(x^i, x_t) - \tilde{w}_t^i(x^i, x_t) \right) dt + D_{KL}(\bar{q}_0\|\bar{p}_0),$$

*where $\tilde{v}_t(x^i, x_t)$, $\tilde{w}_t(x^i, x_t)$ are the respective reverse probability velocities, which generate the identical distributions of paths as the forward ones.*

A proof is provided in Appendix A.1. The key idea of the extension is to see the space as a grid, wherein the flow between non-neighbor states becomes negligible for small step sizes.
By Proposition A.1 in Appendix A, $D_{KL}(\bar{q}_1\|\bar{p}_1)$ depends only on the forward probability velocities and the learned probability ratios between neighbor states. Unfortunately, we lack access to these probability ratios. However, the following statement provides a computable upper bound,

**Theorem 4.2.** *Under the conditions of Theorem 4.1. $D_{KL}(\bar{q}_1\|\bar{p}_1)$ is bounded from above by*

$$D_{KL}(\bar{q}_0\|\bar{p}_0) + \int_0^1 \sum_{x_t} \bar{q}_t(x_t) \sum_{i=1}^{L} \sum_{x^i \neq x_t^i} \left( w_t^i(x^i, x_t) \log \frac{w_t^i(x^i, x_t)}{v_t^i(x^i, x_t)} + v_t^i(x^i, x_t) - w_t^i(x^i, x_t) \right) dt.$$

A proof is provided in Appendix A.1. Motivated by the last result and Lou et al. (2024), we choose $\bar{p}_t(x)$ to be the learned approximation of flow $p_t$ in Equation (2) with the coupling $\pi(x_0, x_1)$, i.e., $\bar{p}_t(x) = p_t(x; \theta)$ and $v_t = u_t(x^i, x_t; \theta)$. On the other hand, we choose $\bar{q}_t(x)$ to have the dynamics of $p_t$, but with the coupling $\bar{\pi}(x, y) = p_0(x)\delta_{x_1}(y) = \int \pi(x, z)dz\delta_{x_1}(y)$. Clearly, $\bar{q}_0(x) = p_0(x)$, $\bar{q}_1(x) = \delta_{x_1}(x)$ and $\bar{q}_t(x) = p_{t|1}(x|x_1)$. We notice that since $\bar{q}_0(x) = p_0(x)$ and $\bar{p}_0(x) = p_0(x)$, then $D_{KL}(\bar{q}_0\|\bar{p}_0) = 0$. Furthermore $D_{KL}(\bar{q}_1(x)\|\bar{p}_1(x)) = D_{KL}(\delta_{x_1}(x)\|p_1(x; \theta)) = -\log p_1(x_1; \theta)$. Thus, for such choices, $-\log p_t(x_1; \theta)$ is bounded from above by

$$\int_0^1 \sum_{x_t} p_{t|1}(x_t|x_1) \sum_{i=1}^{L} \sum_{x^i \neq x_t^i} \left( w_t^i(x^i, x_t) \log \frac{w_t^i(x^i, x_t)}{u_t^i(x^i, x_t; \theta)} + u_t^i(x^i, x_t; \theta) - w_t^i(x^i, x_t) \right) dt. \quad (13)$$

This bounds the negative-log-likelihood (NLL) for *general* DFM models. Shaul et al. (2025) concurrently obtained a similar result via an ELBO-based derivation. Taking expectations over $p_1(x_1)$ on both sides gives a general bound on cross entropy $H(p_1, p_1(\theta))$. For the dynamics from Equation (4),

$$H(p_1, p_1(\theta)) \leq \mathcal{B} := \int_0^1 \frac{\dot{k}_t}{1 - k_t} \sum_{x_1, x_0} \pi(x_1, x_0) \sum_{x_t} p_{t|1,0}(x_t|x_1, x_0) \sum_{i=1}^{L}$$

$$\left( -\delta_{x_1^i \neq x_t^i} \log p_{1|t}^i(x_1^i|x_t; \theta) + 1 - p_{1|t}^i(x_t^i|x_t; \theta) - \delta_{x_1^i \neq x_t^i} \right) dt. \quad (14)$$

A detailed derivation is provided in Appendix A.2. Hence $e^{\frac{\mathcal{B}}{L}}$ is a computable upper bound of the perplexity $e^{\frac{H(p_1, p_1(\theta))}{L}}$ that can be used for training and evaluation (Algorithm 2 in Appendix B). Additionally, we provide an expression for the exact perplexity in Appendix A.5, but this cannot be used in practice as it requires knowing the learned probability ratios between neighbor states.

## 4.2 An Alternative Upper Bound on the Perplexity

Analogous to Haxholli et al. (2025)'s findings for discrete diffusion models, using the continuity equation, we show that the distribution entropy at the flow's endpoint can be expressed as follows:

**Proposition 4.3.** *Given a discrete flow $\bar{q}_t$ with a corresponding forward velocity field $w_t$, the entropy of distribution $\bar{q}_1$ can be written as*

$$H(\bar{q}_1) = H(\bar{q}_0) + \int_0^1 \sum_{x_t} \bar{q}_t(x_t) \sum_{i=1}^L \sum_{x^i} w_t^i(x_t^i, x) \frac{\bar{q}_t(x)}{\bar{q}_t(x_t)} \left( \log \frac{\bar{q}_t(x)}{\bar{q}_t(x_t)} - 1 \right) dt, \qquad (15)$$

where $x$ is such that $x^{-i} = x_t^{-i}$ and $x^i$ varies in the third sum. Combining this with Theorem 4.2 yields a direct upper bound on the cross-entropy between the terminal distributions of two flows.

**Proposition 4.4.** *Under the conditions of Theorem 4.1, the following inequality holds*

$$H(\bar{q}_1, \bar{p}_1) \leq H(\bar{q}_0) - \sum_{x_t} \bar{q}_t(x_t) \sum_{i=1}^L \sum_{x^i \neq x_t^i} \tilde{w}_t^i(x^i, x_t) + D_{KL}(\bar{q}_0 \| \bar{p}_0) + \int_0^1 \sum_{x_0, x_1} \pi(x_0, x_1)$$

$$\sum_{x_t} \bar{q}_t(x_t | x_0, x_1) \sum_{i=1}^L \sum_{x^i \neq x_t^i} \left( \frac{\bar{q}_t^i(x | x_0, x_1)}{\bar{q}_t^i(x_t | x_0, x_1)} \tilde{w}_t^i(x_t^i, x) \log \frac{\tilde{w}_t^i(x_t^i, x)}{v_t^i(x^i, x_t)} + v_t^i(x^i, x_t) \right) dt, \qquad (16)$$

where $x$ is defined as in Proposition 4.3. This provides another upper bound on perplexity. Indeed, by setting $\bar{q}_t$ as $p_t$ from Equation (2) with coupling $\pi(x_0, x_1)$, and $\bar{p}_t(x) = p_t(\theta)$, so that $H(\bar{q}_1, \bar{p}_1) = H(p_1, p_1(\theta))$, we obtain the DFM extension of the discrete diffusion bound of Haxholli et al. (2025). See Appendix A.3 for details. As shown in Gat et al. (2024), the backward probability velocity $\tilde{w}_t$ can be computed explicitly in important cases: when the coupling is independent $\pi(x_0, x_1) = p_0(x_0)q_1(x_1)$, and when the source is either masked or has i.i.d. dimensions $p_0(x_0) = \prod_{i=1}^N p_0(x_0^i)$. In these cases,

$$\tilde{w}_t(x^i, x_t) = -\frac{\check{k}_t}{k_t} \left[ \delta_{x_t^i}(x^i) - p_0^i(x^i) \right]. \qquad (17)$$

Since we can compute all terms on the RHS of Inequality (16), it provides an alternative practical upper bound of the perplexity, as described in Algorithm 3, Appendix B. For the special masked dynamics, $p_0^i(x^i) = \delta_m(x^i)$, the two bounds coincide. A derivation can be found in Appendix A.3.1. These bounds provide principled training objectives and serve as effective evaluation metrics.

## 5 EXPERIMENTS

This section empirically validates our results. As proof of concept, we first show on small-vocabulary datasets that applying Theorem 3.1 effectively reduces jumps. Therein, we use the similarity metric $s(x^i, x_t^i) = 1 - \delta_{x_t^i}(x^i)$. We then confirm our bounds empirically, and in simple settings, estimate their tightness. In Section 5.3, we use $s(x^i, x_t^i) = \|e_m(x^i) - e_m(x_t^i)\|^2$, with learnable embeddings, where $e_m(k)$ denotes the $k$-th column of a learnable embedding matrix $E_{d \times V}$. Therein, we demonstrate that in realistic scenarios, applying Theorem 3.1 can reduce generation steps up to 8-fold to reach the same generative perplexity for BoW source distributions. Additionally, we introduce a new flow type (multimask-flow) and show it outperforms masked flows, especially when combined with OT. Further, we calculate OT overhead, showing minibatch OT is practical. Finally, we conduct ablation studies on three key factors: OT batch size, similarity metric, and OT solver. We use the POT implementation of Flamary et al. (2021) to compute optimal minibatch couplings throughout these experiments.

### 5.1 PROOF OF CONCEPT EXPERIMENTS

We trained a time-conditioned GPT-2 transformer with full attention on the Morse-code converted Shakespeare dataset, where non-convertible characters were left unchanged. The source sequence used was Bag-of-Words (BoW). A sample sequence from the BoW is constructed by sampling independently per position from the token frequencies in the training set. Training consisted of 100k iterations, character-level tokenization, sequence length 128, and batch size 64. We compared standard training with minibatch-OT. Minibatch OT increased training time by $0.3\%$ without affecting inference. We used Hamming distance and the Sinkhorn algorithm with entropy regularization parameter $\epsilon$. During inference with 1024 Euclidean steps, we counted token changes at each position across 3,000 generated sequences. Results appear in Table 1.

Since unstructured source sequences require modifying most tokens to generate structured data, there is a natural lower bound on required modifications. In Shakespeare Morse, the vocabulary contains three main tokens, each with probability $\sim 1/3$. Thus, any given token has probability $2/3$ of needing change, yielding an expected minimum of $128 \times \frac{2}{3} = 85.33$ jumps for sequence length 128. The standard method's 85.47 jumps nearly matches this theoretical minimum, suggesting near-optimal performance. That OT reduces this to 74.84 is significant, demonstrating that OT-trained models generate samples closer to the source sequence while maintaining unbiased sampling when marginalizing across source sequences. Beyond jump counts, we report perplexity using the bound in Equation 14. For OT-trained models, we apply exact minibatch OT during testing to prevent sample repetition. We also report relative jumps, calculated as the ratio of each model's jump count to that of the best performing model.

We also performed the same experiments on Shakespeare using a character-level tokenizer (see Appendix C.1). As discussed in Section 3, the increased vocabulary size makes Hamming distance less effective, necessitating the usage of the $L_2$ metric (Section 5.3) or other specialized measures.

Table 1: Using minibatch OT reduces the number of jumps by $\sim 14\%$. We notice that by increasing the entropy regularization we get closer to the results of training without OT.

| Model (L=128) | Jumps | Relative Jumps | Perplexity |
|---|---|---|---|
| Normal | $85.47 \pm 0.1$ | 1.14 | 2.35 |
| With OT $\epsilon = 0.1$ | $82.86 \pm 0.1$ | 1.1 | 2.14 |
| With OT $\epsilon = 0.01$ | $\mathbf{74.87} \pm 0.1$ | 1 | **2.12** |

## 5.2 Utilizing the Bounds and Estimating Their Tightness

We test in practice the utility of our bounds as optimization targets and evaluation metrics. In Section 4.2, we mentioned that for masked DFM, both bounds coincide and simplify to $\int_0^1 \frac{1}{1-t} \sum_{x_1,x_0} \pi(x_1,x_0) \sum_{x_t} p_{t|1,0}(x_t|x_1,x_0) \sum_{i=1}^{L} -\delta_m(x_t^i) \log p_{1|t}^i(x_1^i|x_t;\theta) dt$. This matches the MD4 bound of Shi et al. (2024) for masked discrete diffusion (Appendix A.4). We denote models trained with this loss as DFM-S, those trained with the loss multiplied by $(1-t)$ as DFM-N, and those trained with cross-entropy as DFM-O. Using the architecture from Section 5.1 with GPT2 tokenization, we trained on OpenWebText (OWT) (Gokaslan & Cohen, 2019) for 400K steps (batch size 512, sequence length 128). Testing on datasets from Lou et al. (2024), DFM-N performed best (Appendix C.2), so we compared DFM-N against SEDD and GPT-2 for longer sequences (L=1024). The performance gap between DFM-N and DFM-S mirrors that of CEDD and CEDDT in diffusion (Haxholli et al., 2025), further validating our bounds. Table 2 demonstrates that our bounds enable direct comparison with autoregressive models, with DFM-N and SEDD performing similarly.

Table 2: Results comparing SEDD, DFM-N, and GPT2.

| Model (L=1024) | Lambada | Wikitext2 | PTB | Wikitext103 | LM1B |
|---|---|---|---|---|---|
| SEDD | 52.18 | 42.02 | 117.00 | 41.83 | 80.79 |
| DFM-N | 53.19 | 42.00 | **111.58** | 41.64 | 77.87 |
| GPT2 | **49.02** | **37.68** | 134.13 | **37.55** | **58.92** |

A natural question is how tight our bounds are and their implications for the GPT-2/DFM-N performance gap. In small-scale masked flow settings, the bound exceeds the ground-truth value by roughly 11%. The NLL differences are similar to those reported by Song et al. (2021b) in the case of continuous diffusion models. See Appendix C.5 for full details.

## 5.3 Multi-masked Flows and Minibatch-OT on OpenWebText

To test minibatch-OT in practice, we trained a time-conditioned GPT-2-sized model with full attention for 400k iterations on OWT, using batch size 512 and sequence length 128. We compared: (1) a baseline (DFM-B) without OT, and (2) DFM-B-OT, that is, DFM-B trained using minibatch-OT with

$L_2$ metric (Corollary 3.3). Both used the GPT-2 tokenizer (vocabulary size 50,257). The source distribution was BoW as in Section 5.1, but with OWT as the training set. The cross-entropy loss was used in both cases. After training, we generated 10,240 samples and evaluated quality using GPT2-large and Llama3.1 8B. OT significantly improved generative perplexity, reducing by 8-fold the generation steps (1024 to 128) needed to match the non-OT model's score. Additionally, we measured the total transport cost in both dynamic and Kantorovich formulations for models trained with and without OT. The two formulations yielded similar values, and models trained with OT show lower transport costs as expected. See Appendix C.6 for details.

Table 3: Result differences between SOTA masked-flows, DFM-B, and DFM-MMLM with/without *minibatch* OT. GPT-2 Large was used as a judge. Asterisks denote the best results across all categories.

| Generation Steps: | 8 | 16 | 32 | 64 | 128 | 1024 |
|---|---|---|---|---|---|---|
| DFM-B | 345.94 | 241.16 | 211.99 | 197.48 | 192.75 | 185.12 |
| DFM-B-OT | **331.88*** | **233.24*** | **203.08** | **191.17** | **185.06** | **178.53** |
| DFM-S (MD4 loss) | 587.80 | 316.25 | 222.46 | 188.62 | 169.81 | 156.81 |
| DFM-N | **556.73** | **296.25** | 210.11 | 176.34 | 160.17 | 147.07 |
| DFM-O | 560.67 | 300.06 | **208.06** | **175.59** | **159.03** | **146.54** |
| DFM-MMLM | 536.50 | 288.38 | 204.77 | 170.61 | 155.45 | 143.48 |
| DFM-MMLM-OT | **525.83** | **283.10** | **199.55*** | **167.86*** | **153.51*** | **141.92*** |

Further, using the same experimental settings as DFM-B, we test multimasked flow with and without OT. Table 3 presents all generative perplexity results using GPT2 for evaluation, and the perplexity bound results are provided in Appendix C.3. In Appendix C.4, we provide the standard deviations, Llama evaluation results, and demonstrate through entropy scores that OT preserves the diversity. The Pearson correlation of perplexity and generative perplexity results is 0.96 (see Appendix C.7).

## 5.4 Scaling Properties of Minibatch-OT

We examine the computational overhead introduced by minibatch OT during training. While OT computation is vocabulary-size independent, its requirements increase with batch size under Sinkhorn and each parameter update requires computing a minibatch OT coupling. Table 4 compares the time for 1000 couplings (CPU or GPU) against 1000 diffusion updates without OT. OT adds only 3.4% overhead in our experiments. Larger batch sizes require GPU acceleration, maintaining the overhead between 10-15%. All experiments use fixed sequence length $L = 128$.

Table 4: Timing (seconds) for 1000 batches with sequence length 128. The symbol 'E' indicates extrapolated values due to memory constraints (Nvidia GH200 reaches its maximum capacity).

| Batch size: | 32 | 64 | 128 | 256 | 512 | 1024 | 2048 | 4096 |
|---|---|---|---|---|---|---|---|---|
| POT (CPU) | 1.94 | 2.23 | 2.99 | 4.91 | 12.57 | 78.90 | 275.91 | 834.77 |
| POT (GPU) | 8.93 | 43.26 | 93.11 | 156.64 | 149.30 | 150.31 | 179.13 | 265.34 |
| Pure diffusion | 54.7 | 63.6 | 104.9 | 173.0 | 367.8 | 634.4 | $1129^E$ | $2010^E$ |

Sequence length does not have a negative impact on computational scaling. Normally, the primary overhead stems from the Sinkhorn operation, which processes pre-computed pairwise sequence distances. Consequently, sequence length does not affect Sinkhorn's computational cost. We tested the overall role of the length empirically by increasing the sequence length 8 times (from $L = 128$ to $L = 1024$), which yielded only a 4.6x increase in OT computation time (from 12,57 to 57,99 seconds per thousand minibatches). This scaling behavior has important implications for training efficiency: At $L = 128$, OT adds 3.4% to the total training time. At $L = 1024$, this overhead should drop below 1.9% because diffusion-only training time scales at best roughly linearly (and up to quadratically if attention dominates) with sequence length, whereas OT in our experiments scaled more slowly. Consequently, the relative cost of mini-batch OT is not expected to increase with sequence length.

## 5.5 ABLATION EXPERIMENTS

We conduct ablation studies on three key factors: OT batch size, similarity metric, and OT solver. Switching from Sinkhorn to exact solver in multimask flow yields minor improvements for small step sizes (8 and 16), with more pronounced gains in bound values. Increasing minibatch OT size to 4096 (while maintaining diffusion batch size) slightly improves results at 32 and 64 steps, again with more pronounced improvements in bound values. Replacing the $L_2$ metric with a heuristically chosen learned metric $-\sum_{i=1}^{L} \log p_{1|0}^i(x^i|x_0;\theta)$ greatly improves generative perplexity at small step sizes but underperforms overall. More details and the results can be found in Appendix C.8.

## 6 RELATED WORK

Diffusion-based models have proven highly effective in capturing the structure of continuous data distributions, leading to significant advancements in generative modeling (Song et al., 2020a; 2021c; Kingma et al., 2021; Nichol & Dhariwal, 2021; Saharia et al., 2022; Ramesh et al., 2022). Given their success in image, video and audio synthesis, researchers have explored their applicability to language modeling (Chen et al., 2023; Gulrajani & Hashimoto, 2024; Li et al., 2022; Dieleman et al., 2022; Strudel et al., 2022; Gong et al., 2022; Mahabadi et al., 2023).

An alternative paradigm for discrete data, particularly in NLP, is discrete diffusion. Introduced by Hoogeboom et al. (2021); Austin et al. (2021) and extended to continuous-time settings (Campbell et al., 2022; Lou et al., 2024), these models offer a structured approach to learning categorical distributions. Training typically uses the variational lower bound or cross-entropy loss, similar to continuous diffusion (Dieleman et al., 2022).

To expand the design space of discrete diffusion, Campbell et al. (2024); Gat et al. (2024) introduce discrete flow matching, notably avoiding conditional score ratio calculations during training and thus bypassing matrix exponential computation. Instead of focusing on a path-length-oriented objective, Shaul et al. (2025) define a kinetic energy OT objective, derive the optimum for specific DFM classes, and independently obtain a bound similar to ours from an ELBO perspective. While in this work we focus on pure DFM models, Arriola et al. (2025) introduce block diffusion language models that interpolate between discrete denoising diffusion and autoregressive models. Regarding scaling, Nie et al. (2025) train masked diffusion models up to 1.1B parameters to systematically evaluate against comparable or larger ARMs. Their 1.1B MDM outperforms the 1.1B TinyLlama trained on the same data across four of eight zero-shot benchmarks.

## 7 LIMITATIONS AND FUTURE WORK

Flow matching with minibatch OT involves two interacting optimization procedures: choosing optimal minibatch coupling and training the flow model. The coupling affects flow dynamics, while embedding updates during training alter the optimal coupling when using embedding-based similarities. Though the model's local view of the flow at each step provides stability, we can enhance it by decoupling network embeddings from those used in minibatch coupling, for instance, using moving-average embeddings for OT. In addition, future work could explore connections between the fictitious grid in DFM-MMLM and VQ-VAEs, potentially defining source distributions using the fictitious grid state closest to the encoding of each data point in $L_2$ distance.

## 8 CONCLUSION

We developed a weighted path-length dynamic OT objective for DFM that minimizes dissimilarity-weighted jumps between states, derived its Kantorovich formulation establishing a categorical Benamou-Brenier-type theorem. We extended two discrete diffusion bounds to DFM, enabling comparisons with autoregressive and discrete diffusion models. Experiments show minibatch OT reduces inference steps up to 8-fold (1024 to 128) while maintaining generative perplexity on GPT2-scale models. Our multimask flow (DFM-MM) surpasses masked DFM in generative perplexity without sacrificing diversity, with further gains under OT.

## LLM USAGE STATEMENT

Large Language Models were used in this paper to improve the conciseness and quality of the text at the sentence level.

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

## APPENDIX

## A  THEORETICAL RESULTS

### A.1  PROOFS

**Proof of Theorem 3.1:**

We begin with

$$\int_0^1 \sum_{x_t} p(x_t) \left( \sum_{i=1}^L \sum_{x^i \neq x_t^i} u_t^i(x^i, x_t) s(x^i, x_t^i) \right) dt \tag{18}$$

which due to Equation (6) can be rewritten as

$$= \int_0^1 \sum_{x_t} p(x_t) \left( \sum_{i=1}^L \sum_{x^i \neq x_t^i} \sum_{x_0,x_1} u_t^i(x^i, x_t | x_0, x_1) p(x_0, x_1 | x_t) s(x^i, x_t^i) \right) dt \tag{19}$$

$$= \int_0^1 \sum_{x_t} \sum_{i=1}^L \sum_{x^i \neq x_t^i} \sum_{x_0,x_1} u_t^i(x^i, x_t | x_0, x_1) s(x^i, x_t^i) p(x_0, x_1, x_t) dt \tag{20}$$

$$= \int_0^1 \sum_{x_0,x_1} \sum_{i=1}^L \sum_{x_t} \sum_{x^i \neq x_t^i} u_t^i(x^i, x_t | x_0, x_1) s(x^i, x_t^i) p(x_t | x_0, x_1) p(x_0, x_1) dt. \tag{21}$$

For $p(x_t | x_0, x_1)$ as in Equation (4), by Equation (5) we have that

$$u_t^i(x^i, x_t | x_0, x_1) = \frac{\dot{k}_t}{1 - k_t} \left( \delta_{x_1^i}(x^i) - \delta_{x_t^i}(x^i) \right). \tag{22}$$

Continuing from Equation (21)

$$\int_0^1 \sum_{x_0,x_1} \sum_{i=1}^L \sum_{x_t} \sum_{x^i \neq x_t^i} \frac{\dot{k}_t}{1 - k_t} \left( \delta_{x_1^i}(x^i) - \delta_{x_t^i}(x^i) \right) s(x^i, x_t^i) p(x_t | x_0, x_1) p(x_0, x_1) dt \tag{23}$$

$$= \int_0^1 \sum_{x_0,x_1} \sum_{i=1}^L \sum_{x_t^i} \sum_{x^i \neq x_t^i} \frac{\dot{k}_t}{1 - k_t} \left( \delta_{x_1^i}(x^i) - \delta_{x_t^i}(x^i) \right) s(x^i, x_t^i) p^i(x_t^i | x_0, x_1) p(x_0, x_1) dt \tag{24}$$

$$= \int_0^1 \sum_{x_0,x_1} \sum_{i=1}^L \sum_{\substack{x_t^i, x^i \\ x_t^i \neq x^i}} \frac{\dot{k}_t}{1 - k_t} \left( \delta_{x_1^i}(x^i) - \delta_{x_t^i}(x^i) \right) s(x^i, x_t^i) p^i(x_t^i | x_0, x_1) p(x_0, x_1) dt \tag{25}$$

$$= \int_0^1 \sum_{x_0,x_1} \sum_{i=1}^L \sum_{\substack{x_t^i, x^i \\ x_t^i \neq x^i}} \frac{\dot{k}_t}{1 - k_t} \delta_{x_1^i}(x^i) s(x^i, x_t^i) p^i(x_t^i | x_0, x_1) p(x_0, x_1) dt \tag{26}$$

$$= \int_0^1 \sum_{x_0,x_1} \sum_{i=1}^L \sum_{\substack{x_t^i, x^i \\ x_t^i \neq x^i \\ x^i = x_1^i}} \frac{\dot{k}_t}{1 - k_t} s(x^i, x_t^i) p^i(x_t^i | x_0, x_1) p(x_0, x_1) dt, \tag{27}$$

where again due to the choice of Equation (4)

$$\int_0^1 \sum_{x_0,x_1} \sum_{i=1}^L \sum_{\substack{x_t^i, x^i \\ x_t^i \neq x^i \\ x^i = x_1^i}} \frac{\dot{k}_t}{1 - k_t} s(x^i, x_t^i) \left( (1 - k_t) \delta_{x_0^i}(x_t^i) + k_t \delta_{x_1^i}(x_t^i) \right) p(x_0, x_1) dt \tag{28}$$

$$= \int_0^1 \sum_{x_0,x_1} \sum_{i=1}^L \left( \sum_{\substack{x_t^i,x^i \\ x_t^i \neq x^i \\ x^i = x_1^i}} \frac{\dot{k}_t}{1-k_t}(1-k_t)\delta_{x_0^i}(x_t^i) + \sum_{\substack{x_t^i,x^i \\ x_t^i \neq x^i \\ x^i = x_1^i}} \frac{\dot{k}_t}{1-k_t}k_t\delta_{x_1^i}(x_t^i) \right) s(x^i,x_t^i)p(x_0,x_1)dt \tag{29}$$

$$= \int_0^1 \sum_{x_0,x_1} \sum_{i=1}^L \left( \sum_{\substack{x_t^i,x^i \\ x_t^i \neq x^i \\ x^i = x_1^i}} \dot{k}_t\delta_{x_0^i}(x_t^i) + 0 \right) s(x^i,x_t^i)p(x_0,x_1)dt, \tag{30}$$

where the second expression is zero since in the sum one must have $x_t^i \neq x^i$ and $x^i = x_1^i$, therefore $x_t^i \neq x_1^i$ which sets $\delta_{x_1^i}(x_t^i)$ to 0. Hence we only have

$$\int_0^1 \sum_{x_0,x_1} \sum_{i=1}^L \sum_{\substack{x_t^i,x^i \\ x_t^i \neq x^i \\ x^i = x_1^i}} \dot{k}_t\delta_{x_0^i}(x_t^i)s(x^i,x_t^i)p(x_0,x_1)dt = \int_0^1 \sum_{x_0,x_1} \sum_{i=1}^L \sum_{\substack{x_t^i,x^i \\ x_t^i \neq x^i \\ x^i = x_1^i \\ x_t^i = x_0^i}} s(x^i,x_t^i)\dot{k}_tp(x_0,x_1)dt. \tag{31}$$

Expression

$$\sum_{\substack{x_t^i,x^i \\ x_t^i \neq x^i \\ x^i = x_1^i \\ x_t^i = x_0^i}} s(x^i,x_t^i) \tag{32}$$

is clearly $s(x_1^i,x_0^i)$ when $x_0^i \neq x_1^i$ and zero otherwise. Thus, we have

$$\int_0^1 \sum_{x_t} p(x_t) \left( \sum_{i=1}^L \sum_{x^i \neq x_t^i} s(x^i,x_t^i)u_t^i(x^i,x_t) \right) dt = \int_0^1 \sum_{x_0,x_1} \sum_{i=1}^L s(x_1^i,x_0^i)\dot{k}_tp(x_0,x_1)dt \tag{33}$$

$$= \sum_{x_0,x_1} \sum_{i=1}^L s(x_1^i,x_0^i)(k_1 - k_0)p(x_0,x_1) = \sum_{x_0,x_1} \sum_{i=1}^L s(x_1^i,x_0^i)(1-0)p(x_0,x_1) \tag{34}$$

We conclude that

$$\int_0^1 \sum_{x_t} p(x_t) \left( \sum_{i=1}^L \sum_{x^i \neq x_t^i} u_t^i(x^i,x_t)s(x^i,x_t^i) \right) dt = \sum_{x_0,x_1} c(x_0,x_1)p(x_0,x_1), \tag{35}$$

where $c(x_0,x_1) = \sum_{i=1}^L s(x_1^i,x_0^i)$. $\qquad\square$

**Proof of Theorem 4.1:**

We begin by defining two discrete time Markov chains $\hat{p}_t$ and $\hat{q}_t$ whose timestep sizes are $\epsilon$ and the total number of steps is $K = \lfloor \frac{1}{\epsilon} \rfloor$, such that when $\epsilon \to 0$, their marginal distributions converge to those of the flows $\bar{p}_t$ and $\bar{q}_t$. The KL divergence between the paths of such Markov chains can be written as below:

$$D_{KL}(\hat{q},\hat{p}) = \sum_{x_{0:K\epsilon}} \hat{q}(x_{0:K\epsilon}) \log \frac{\hat{q}(x_{0:K\epsilon})}{\hat{p}(x_{0:K\epsilon})} = \sum_{x_{0:K\epsilon}} \hat{q}(x_{0:K\epsilon}) \log \prod_{k=1}^K \frac{\hat{q}(x_{k\epsilon}|x_{(k-1)\epsilon},...,x_0)}{\hat{p}(x_{k\epsilon}|x_{(k-1)\epsilon},...,x_0)} \frac{\hat{q}(x_0)}{\hat{p}(x_0)} \tag{36}$$

$$= \sum_{x_{0:K\epsilon}} \hat{q}(x_{0:K\epsilon}) \left( \sum_{k=1}^{K} \log \frac{\hat{q}(x_{k\epsilon}|x_{(k-1)\epsilon})}{\hat{p}(x_{k\epsilon}|x_{(k-1)\epsilon})} + \log \frac{\hat{q}(x_0)}{\hat{p}(x_0)} \right) \tag{37}$$

$$= \sum_{k=1}^{K} \sum_{\substack{x_{k\epsilon} \\ x_{(k-1)\epsilon}}} \hat{q}(x_{k\epsilon}, x_{(k-1)\epsilon}) \log \frac{\hat{q}(x_{k\epsilon}|x_{(k-1)\epsilon})}{\hat{p}(x_{k\epsilon}|x_{(k-1)\epsilon})} + \sum_{x_0} \hat{q}(x_0) \log \frac{\hat{q}(x_0)}{\hat{p}(x_0)} \tag{38}$$

$$= \sum_{k=1}^{K} \sum_{x_{(k-1)\epsilon}} \hat{q}(x_{(k-1)\epsilon}) \sum_{x_{k\epsilon}} \hat{q}(x_{k\epsilon}|x_{(k-1)\epsilon}) \log \frac{\hat{q}(x_{k\epsilon}|x_{(k-1)\epsilon})}{\hat{p}(x_{k\epsilon}|x_{(k-1)\epsilon})} + \sum_{x_0} \hat{q}(x_0) \log \frac{\hat{q}(x_0)}{\hat{p}(x_0)} \tag{39}$$

$$= I + D_{KL}(\hat{q}(x_0)\|\hat{p}(x_0)), \tag{40}$$

where

$$I = \sum_{k=1}^{K} \sum_{x_{(k-1)\epsilon}} \hat{q}(x_{(k-1)\epsilon}) \sum_{x_{k\epsilon}} \hat{q}(x_{k\epsilon}|x_{(k-1)\epsilon}) \log \frac{\hat{q}(x_{k\epsilon}|x_{(k-1)\epsilon})}{\hat{p}(x_{k\epsilon}|x_{(k-1)\epsilon})} \tag{41}$$

is a weighted sum of KL divergences with non-negative weights, that is

$$I = \sum_{k=1}^{K} \sum_{x_{(k-1)\epsilon}} \hat{q}(x_{(k-1)\epsilon}) D_{KL}(\hat{q}(x_{k\epsilon}|x_{(k-1)\epsilon})\|\hat{p}(x_{k\epsilon}|x_{(k-1)\epsilon})). \tag{42}$$

First we will simplify notation and write $t_k = (k-1)\epsilon$, as well as $\hat{q}(x_{k\epsilon} = x | x_{(k-1)\epsilon} = z) = \hat{q}_{t_k+\epsilon|t_k}(x|z)$, where $z$ and $x$ are states. Therefore the previous Expression (41) becomes

$$I = \sum_{k=1}^{K} \sum_{z} \hat{q}_{t_k}(z) \sum_{x} \hat{q}_{t_k+\epsilon|t_k}(x|z) \log \frac{\hat{q}_{t_k+\epsilon|t_k}(x|z)}{\hat{p}_{t_k+\epsilon|t_k}(x|z)}. \tag{43}$$

Now, we focus on computing expression $D_{KL}(\hat{q}_{t_k+\epsilon|t_k}(x|z)\|\hat{p}_{t_k+\epsilon|t_k}(x|z))$. The sum

$$\sum_{x} \hat{q}_{t_k+\epsilon|t_k}(x|z) \log \frac{\hat{q}_{t_k+\epsilon|t_k}(x|z)}{\hat{p}_{t_k+\epsilon|t_k}(x|z)}. \tag{44}$$

can be separated into three sums:

$$\sum_{\substack{x \\ d_H(x,z)=0}} \hat{q}_{t_k+\epsilon|t_k}(x|z) \log \frac{\hat{q}_{t_k+\epsilon|t_k}(x|z)}{\hat{p}_{t_k+\epsilon|t_k}(x|z)} + \sum_{\substack{x \\ d_H(x,z)=1}} \hat{q}_{t_k+\epsilon|t_k}(x|z) \log \frac{\hat{q}_{t_k+\epsilon|t_k}(x|z)}{\hat{p}_{t_k+\epsilon|t_k}(x|z)} \tag{45}$$

$$+ \sum_{\substack{x \\ d_H(x,z)>1}} \hat{q}_{t_k+\epsilon|t_k}(x|z) \log \frac{\hat{q}_{t_k+\epsilon|t_k}(x|z)}{\hat{p}_{t_k+\epsilon|t_k}(x|z)} \tag{46}$$

We first analyze the second sum. Since $x$ and $z$ differ at exactly one neighbor (say position $j$), from the flow matching update rule $p^i_{t_k+\epsilon|t_k}(y^i|z) = \delta_{z^i}(y^i) + \epsilon u^i_{t_k}(y^i, z)$ applied independently to each position, we can infer that

$$p_{t_k+\epsilon|t_k}(x|z) = \epsilon u^j_{t_k}(x^j, z) \prod_{\substack{i=1 \\ i \neq j}}^{L} \left(1 + u^i_{t_k}(x^i, z)\epsilon\right) = \epsilon u^j_{t_k}(x^j, z) + O(\epsilon^2) \tag{47}$$

therefore

$$\sum_{\substack{x \\ d_H(x,z)=1}} \hat{q}_{t_k+\epsilon|t_k}(x|z) \log \frac{\hat{q}_{t_k+\epsilon|t_k}(x|z)}{\hat{p}_{t_k+\epsilon|t_k}(x|z)} \tag{48}$$

$$= \sum_{j=1}^{L} \sum_{x^j \neq z^j} \epsilon w^j_{t_k}(x^j, z) \log \frac{w^j_{t_k}(x^j, z) + O(\epsilon)}{v^j_{t_k}(x^j, z) + O(\epsilon)} + O(\epsilon^2). \tag{49}$$

For the third sum, since

$$p_{t_k+\epsilon|t_k}(x|z) = \epsilon^2 u_{t_k}^j(x^j, z) u_{t_k}^l(x^j, z) \prod_{\substack{i=1 \\ i \neq j,l}}^{L} \left(1 + u_{t_k}^i(x^i, z)\epsilon\right) = O(\epsilon^2) + O(\epsilon^3) = O(\epsilon^2) \quad (50)$$

we conclude that

$$\sum_{\substack{x \\ d_H(x,z)>1}} \hat{q}_{t_k+\epsilon|t_k}(x|z) \log \frac{\hat{q}_{t_k+\epsilon|t_k}(x|z)}{\hat{p}_{t_k+\epsilon|t_k}(x|z)} = O(\epsilon^2). \quad (51)$$

Therefore the only sum left is the first one

$$\sum_{\substack{x \\ d_H(x,z)=0}} \hat{q}_{t_k+\epsilon|t_k}(x|z) \log \frac{\hat{q}_{t_k+\epsilon|t_k}(x|z)}{\hat{p}_{t_k+\epsilon|t_k}(x|z)} = \hat{q}_{t_k+\epsilon|t_k}(z|z) \log \frac{\hat{q}_{t_k+\epsilon|t_k}(z|z)}{\hat{p}_{t_k+\epsilon|t_k}(z|z)}. \quad (52)$$

In this special case ($x = z$),

$$p_{t_k+\epsilon|t_k}(z|z) = \prod_{i=1}^{L} \left(1 + u_{t_k}^i(z^i, z)\epsilon\right) = 1 + \epsilon \sum_{i=1}^{L} u_{t_k}^i(z^i, z) + O(\epsilon^2), \quad (53)$$

thus

$$\hat{q}_{t_k+\epsilon|t_k}(z, z) = 1 + \epsilon \sum_{i=1}^{L} w_{t_k}^i(z^i, z) + O(\epsilon^2), \quad (54)$$

$$\log \hat{q}_{t_k+\epsilon|t_k}(z, z) = \epsilon \sum_{i=1}^{L} w_{t_k}^i(z^i, z) + O(\epsilon^2), \quad (55)$$

$$\log \hat{p}_{t_k+\epsilon|t_k}(z, z) = \epsilon \sum_{i=1}^{L} v_{t_k}^i(z^i, z) + O(\epsilon^2), \quad (56)$$

implying

$$\hat{q}_{t_k+\epsilon|t_k}(z|z) \log \frac{\hat{q}_{t_k+\epsilon|t_k}(z|z)}{\hat{p}_{t_k+\epsilon|t_k}(z|z)} = \hat{q}_{t_k+\epsilon|t_k}(z|z) \left(\log \hat{q}_{t_k+\epsilon|t_k}(z|z) - \log \hat{p}_{t_k+\epsilon|t_k}(z|z)\right) \quad (57)$$

$$= \epsilon \sum_{i=1}^{L} w_{t_k}^i(z^i, z) - \epsilon \sum_{i=1}^{L} v_{t_k}^i(z^i, z) + O(\epsilon^2). \quad (58)$$

When accounting for the fact that $u_{t_k}^i(z^i, z) = -\sum_{x^i \neq z^i} u_{t_k}^i(x^i, z)$, we finally have

$$\hat{q}_{t_k+\epsilon|t_k}(z|z) \log \frac{\hat{q}_{t_k+\epsilon|t_k}(z|z)}{\hat{p}_{t_k+\epsilon|t_k}(z|z)} = \epsilon \sum_{i=1}^{L} \sum_{x^i \neq z^i} \left(v_{t_k}^i(x^i, z) - w_{t_k}^i(x^i, z)\right) + O(\epsilon^2). \quad (59)$$

We get the expression for $D_{KL}(\hat{q}_{t_k+\epsilon|t_k}(x|z) \| \hat{p}_{t_k+\epsilon|t_k}(x|z)))$ by adding all three sums,

$$\epsilon \sum_{i=1}^{L} \sum_{x^i \neq z^i} \left(w_{t_k}^i(x^i, z) \log \frac{w_{t_k}^i(x^i, z) + O(\epsilon)}{v_{t_k}^i(x^i, z) + O(\epsilon)} + v_{t_k}^i(x^i, z) - w_{t_k}^i(x^i, z)\right) + O(\epsilon^2). \quad (60)$$

Plugging this last expression in $I$, one gets

$$I = \sum_{k=0}^{K-1} \epsilon \sum_z \hat{q}_{t_k}(z) \sum_{i=1}^{L} \sum_{x^i \neq z^i} \left(w_{t_k}^i(x^i, z) \log \frac{w_{t_k}^i(x^i, z) + O(\epsilon)}{v_{t_k}^i(x^i, z) + O(\epsilon)} + v_{t_k}^i(x^i, z) - w_{t_k}^i(x^i, z)\right) + O(\epsilon).$$

$$(61)$$

Finally taking the limit $\epsilon \to 0$

$$\int_0^1 \sum_z \bar{q}_t(z) \sum_{i=1}^L \sum_{x^i \neq z^i} \left( w_t^i(x^i, z) \log \frac{w_t^i(x^i, z)}{v_t^i(x^i, z)} + v_t^i(x^i, z) - w_t^i(x^i, z) \right) dt. \qquad (62)$$

Based on the last formula, and by replacing $z$ with $x_t$, we can write,

$$D_{KL}(\bar{q}, \bar{p}) = D_{KL}(\bar{q}_0 \| \bar{p}_0)$$

$$+ \int_0^1 \sum_{x_t} \bar{q}_t(x_t) \sum_{i=1}^L \sum_{x^i \neq x_t^i} \left( w_t^i(x^i, x_t) \log \frac{w_t^i(x^i, x_t)}{v_t^i(x^i, x_t)} + v_t^i(x^i, x_t) - w_t^i(x^i, x_t) \right) dt. \qquad (63)$$

Applying this result when considering flow paths $\bar{p}$ and $\bar{q}$ to have been generated in the opposite direction by the reverse probability velocities $\tilde{v}_t^i(x^i, x_t)$ and $\tilde{w}_t^i(x^i, x_t)$,

$$D_{KL}(\bar{q}, \bar{p}) = D_{KL}(\bar{q}_1 \| \bar{p}_1)$$

$$+ \int_0^1 \sum_{x_t} \bar{q}_t(x_t) \sum_{i=1}^L \sum_{x^i \neq x_t^i} \left( \tilde{w}_t^i(x^i, x_t) \log \frac{\tilde{w}_t^i(x^i, x_t)}{\tilde{v}_t^i(x^i, x_t)} + \tilde{v}_t^i(x^i, x_t) - \tilde{w}_t^i(x^i, x_t) \right) dt \qquad (64)$$

Finally, by combining Equations (63) and (64),

$$\int_0^1 \sum_{x_t} \bar{q}_t(x_t) \sum_{i=1}^L \sum_{x^i \neq x_t^i} \left( \tilde{w}_t^i(x^i, x_t) \log \frac{\tilde{w}_t^i(x^i, x_t)}{\tilde{v}_t^i(x^i, x_t)} + \tilde{v}_t^i(x^i, x_t) - \tilde{w}_t^i(x^i, x_t) \right) dt + D_{KL}(\bar{q}_1 \| \bar{p}_1)$$

$$= \int_0^1 \sum_{x_t} \bar{q}_t(x_t) \sum_{i=1}^L \sum_{x^i \neq x_t^i} \left( w_t^i(x^i, x_t) \log \frac{w_t^i(x^i, x_t)}{v_t^i(x^i, x_t)} + v_t^i(x^i, x_t) - w_t^i(x^i, x_t) \right) dt + D_{KL}(\bar{q}_0 \| \bar{p}_0)$$

$$(65)$$

therefore,

$$D_{KL}(\bar{q}_1 \| \bar{p}_1) = \int_0^1 \sum_{x_t} \bar{q}_t(x_t) \sum_{i=1}^L \sum_{x^i \neq x_t^i} \left( w_t^i(x^i, x_t) \log \frac{w_t^i(x^i, x_t)}{v_t^i(x^i, x_t)} + v_t^i(x^i, x_t) - w_t^i(x^i, x_t) \right) dt$$

$$- \int_0^1 \sum_{x_t} \bar{q}_t(x_t) \sum_{i=1}^L \sum_{x^i \neq x_t^i} \left( \tilde{w}_t^i(x^i, x_t) \log \frac{\tilde{w}_t^i(x^i, x_t)}{\tilde{v}_t^i(x^i, x_t)} + \tilde{v}_t^i(x^i, x_t) - \tilde{w}_t^i(x^i, x_t) \right) dt + D_{KL}(\bar{q}_0 \| \bar{p}_0).$$

$$(66)$$

$\square$

**Proposition A.1.** *For two discrete flows $\bar{p}_t$ and $\bar{q}_t$ with corresponding probability velocities $v_t(x^i, x_t)$ and $w_t(x^i, x_t)$, the following equality holds*

$$D_{KL}(\bar{q}_1 \| \bar{p}_1) = D_{KL}(\bar{q}_0 \| \bar{p}_0) +$$

$$\int_0^1 \sum_{x_t} \bar{q}_t(x_t) \sum_{i=1}^L \sum_{x^i \neq x_t^i} \left( w_t^i(x^i, x_t) \log \frac{w_t^i(x^i, x_t)}{v_t^i(x^i, x_t)} + v_t^i(x^i, x_t) - w_t^i(x^i, x_t) \right) dt -$$

$$\int_0^1 \sum_{x_t} \bar{q}_t(x_t) \sum_{i=1}^L \sum_{x^i \neq x_t^i} \left( r_{\bar{q}_t} w_t^i(x_t^i, x) \log \frac{r_{\bar{q}_t} w_t^i(x_t^i, x)}{r_{\bar{p}_t} v_t^i(x_t^i, x)} + r_{\bar{p}_t} v_t^i(x_t^i, x) - r_{\bar{q}_t} w_t^i(x_t^i, x) \right) dt, \qquad (67)$$

*where $r_{\bar{p}_t} = r_{\bar{p}_t}(x, x_t) = \frac{\bar{p}_t(x)}{\bar{p}_t(x_t)}$, $r_{\bar{q}_t} = r_{\bar{q}_t}(x, x_t) = \frac{\bar{q}_t(x)}{\bar{q}_t(x_t)}$, and where $x$ is a state identical to the current position $x_t$, except for position (dimension) $i$.*

**Proof of Proposition A.1:** Similarly to the proof of Theorem 4.1 above, one can write

$$D_{KL}(\tilde{q}, \tilde{p}) = J + D_{KL}(\tilde{q}_1 \| \tilde{p}_1), \qquad (68)$$

where

$$J = \sum_{k=1}^K \sum_z \tilde{q}_{\tau_k}(z) \sum_x \tilde{q}_{\tau_{k+1} | \tau_k}(x | z) \log \frac{\tilde{q}_{\tau_{k+1} | \tau_k}(x | z)}{\tilde{p}_{\tau_{k+1} | \tau_k}(x | z)}, \qquad (69)$$

for $\tau_k = 1 - (k-1)\epsilon = 1 - t_k$. As before, we can break this expression into three sums and then focus on the ones that concern states $x, z$ that do not differ on more than one dimension. In case that $x$ and $z$ differ in exactly one dimension $(j)$ then as previously we have

$$p_{\tau_{k+1}|\tau_k}(x|z) = \frac{p_{\tau_{k+1}}(x)}{p_{\tau_k}(z)} p_{\tau_k|\tau_{k+1}}(z|x) = \frac{p_{1-t_k-\epsilon}(x)}{p_{1-t_k}(z)} p_{1-t_k|1-t_k-\epsilon}(z|x)$$

$$= \epsilon \frac{p_{1-t_k-\epsilon}(x)}{p_{1-t_k}(z)} u^j_{1-t_k-\epsilon}(z^j, x) \prod_{\substack{i=1 \\ i \neq j}}^{L} \left(1 + u^i_{1-t_k-\epsilon}(z^i, x)\epsilon\right) = \epsilon \frac{p_{1-t_k-\epsilon}(x)}{p_{1-t_k}(z)} u^j_{1-t_k-\epsilon}(z^j, x) + O(\epsilon^2).$$

(70)

Similarly as before, we can develop the expression for the case when the Hamming distance between $x$ and $z$ is 0. By combining these cases as in the previous theorem and taking $\epsilon \to 0$, we derive an expression for $J$:

$$D_{KL}(\bar{q}, \bar{p}) =$$

$$\int_0^1 \sum_z \bar{q}_{1-t}(z) \sum_{i=1}^{L} \sum_{x^i \neq z^i} \left( r_{\bar{q}_{1-t}} w^i_{1-t}(z^i, x) \log \frac{r_{\bar{q}_{1-t}} w^i_{1-t}(z^i, x)}{r_{\bar{p}_{1-t}} v^i_{1-t}(z^i, x)} \right.$$

$$\left. + r_{\bar{p}_{1-t}} v^i_{1-t}(z^i, x) - r_{\bar{q}_{1-t}} w^i_{1-t}(z^i, x) \right) dt$$

$$+ D_{KL}(\bar{q}_1, \bar{p}_1). \tag{71}$$

We conclude the proof by setting $\tau = 1 - t$,

$$D_{KL}(\bar{q}, \bar{p}) = D_{KL}(\bar{q}_1, \bar{p}_1)$$

$$+ \int_0^1 \sum_z \bar{q}_\tau(z) \sum_{i=1}^{L} \sum_{x^i \neq z^i} \left( r_{\bar{q}_\tau} w^i_\tau(z^i, x) \log \frac{r_{\bar{q}_\tau} w^i_\tau(z^i, x)}{r_{\bar{p}_\tau} v^i_\tau(z^i, x)} + r_{\bar{p}_\tau} v^i_\tau(z^i, x) - r_{\bar{q}_\tau} w^i_\tau(z^i, x) \right) d\tau$$

(72)

followed by $z = x_\tau$, where $r_{\bar{p}_\tau} = r_{\bar{p}_\tau}(x, z) = \frac{\bar{p}_\tau(x)}{\bar{p}_\tau(z)}$ and $r_{\bar{q}_\tau} = r_{\bar{q}_\tau}(x, z) = \frac{\bar{q}_\tau(x)}{\bar{q}_\tau(z)}$. □

**Proof of Theorem 4.2:**

We now set to prove that $D_{KL}(\bar{q}(x_1) \| \bar{p}(x_1)) \leq D_{KL}(\bar{q}, \bar{p})$. Since in Equation (66), the term

$$\int_0^1 \sum_{x_t} \bar{q}_t(x_t) \sum_{i=1}^{L} \sum_{x^i \neq x^i_t} \left( \tilde{w}^i_t(x^i, x_t) \log \frac{\tilde{w}^i_t(x^i, x_t)}{\tilde{v}^i_t(x^i, x_t)} + \tilde{v}^i_t(x^i, x_t) - \tilde{w}^i_t(x^i, x_t) \right) dt \tag{73}$$

is a positively weighted sum of KL divergences this immediately implies that

$$D_{KL}(\bar{q}_1 \| \bar{p}_1) \leq D_{KL}(\bar{q}_0 \| \bar{p}_0)$$

$$+ \int_0^1 \sum_{x_t} \bar{q}_t(x_t) \sum_{i=1}^{L} \sum_{x^i \neq x^i_t} \left( w^i_t(x^i, x_t) \log \frac{w^i_t(x^i, x_t)}{v^i_t(x^i, x_t)} + v^i_t(x^i, x_t) - w^i_t(x^i, x_t) \right) dt \tag{74}$$

We can also show this result to be an immediate consequence of the Jensen inequality. Indeed,

$$-D_{KL}(\hat{q} \| \hat{p}) = \sum_{x_{0:K\epsilon}} \hat{q}(x_{0:K\epsilon}) \log \frac{\hat{p}(x_{0:K\epsilon})}{\hat{q}(x_{0:K\epsilon})} = \sum_{x_{0:K\epsilon}} \hat{q}(x_0) \hat{q}(x_{\epsilon:K\epsilon}|x_0) \log \frac{\hat{p}(x_{0:K\epsilon})}{\hat{q}(x_{0:K\epsilon})} \tag{75}$$

$$= \sum_{x_0} \hat{q}(x_0) \sum_{x_{\epsilon:K\epsilon}} \hat{q}(x_{\epsilon:K\epsilon}|x_0) \log \frac{\hat{p}(x_{0:K\epsilon})}{\hat{q}(x_{0:K\epsilon})} \leq \sum_{x_0} \hat{q}(x_0) \log \sum_{x_{\epsilon:K\epsilon}} \hat{q}(x_{\epsilon:K\epsilon}|x_0) \frac{\hat{p}(x_{0:K\epsilon})}{\hat{q}(x_{0:K\epsilon})} \tag{76}$$

$$\sum_{x_0} \hat{q}(x_0) \log \sum_{x_{\epsilon:K\epsilon}} \frac{\hat{p}(x_{0:K\epsilon})}{\hat{q}(x_0)} = -D_{KL}(\hat{q}(x_0) \| \hat{p}(x_0)) \tag{77}$$

Therefore, $D_{KL}(\hat{q}_0 \| \hat{p}_0) \leq D_{KL}(\hat{q} \| \hat{p})$, and taking the limit $\epsilon \to 0$ we get $D_{KL}(\bar{q}_0 \| \bar{p}_0) \leq D_{KL}(\bar{q} \| \bar{p})$. Applying this result to the reverse processes that generate the marginals $\bar{p}$ and $\bar{q}$ gives $D_{KL}(\bar{q}_1 \| \bar{p}_1) \leq D_{KL}(\bar{q}, \bar{p})$.

In total we have proved that

$$D_{KL}(\bar{q}_1 \| \bar{p}_1) \leq D_{KL}(\bar{q}_0 \| \bar{p}_0)$$

$$+ \int_0^1 \sum_{x_t} \bar{q}_t(x_t) \sum_{i=1}^{L} \sum_{x^i \neq x_t^i} \left( w_t^i(x^i, x_t) \log \frac{w_t^i(x^i, x_t)}{v_t^i(x^i, x_t)} + v_t^i(x^i, x_t) - w_t^i(x^i, x_t) \right) dt \qquad (78)$$

**Proof of Proposition 4.3:**

First we define

$$\delta_x(y^{-i}) = \prod_{j \in \{1,2,\dots,i-1,i+1,\dots L\}} \delta_{x^j}(y^j). \qquad (79)$$

From the definition of entropy,

$$\frac{\partial H(\bar{q}_t)}{\partial t} = -\frac{\partial}{\partial t} \sum_{x_t} \bar{q}_t(x_t) \log \bar{q}_t(x_t) = -\sum_{x_t} \frac{\partial \bar{q}_t(x_t)}{\partial t} \left( \log \bar{q}_t(x_t) + 1 \right) \qquad (80)$$

$$= \sum_{x_t} \frac{\partial \bar{q}_t(x_t)}{\partial t} \left( \log \frac{\bar{q}_t(x)}{\bar{q}_t(x_t)} - 1 \right) - \sum_{x_t} \frac{\partial \bar{q}_t(x_t)}{\partial t} \log \bar{q}_t(x). \qquad (81)$$

We prove the last term $\sum_{x_t} \frac{\partial \bar{q}_t(x_t)}{\partial t} \log \bar{q}_t(x) dx_t$ is 0. From the Continuity Equation (Gat et al., 2024),

$$\frac{\partial \bar{q}_t(x_t)}{\partial t} = \sum_x \bar{q}_t(x) \sum_{i=1}^{L} \delta_x(x_t^{-i}) w_t^i(x_t^i, x), \qquad (82)$$

we get that

$$\sum_{x_t} \frac{\partial \bar{q}_t(x_t)}{\partial t} \log \bar{q}_t(x)$$

$$= \sum_x \bar{q}_t(x) \sum_{i=1}^{L} \sum_{x_t} \left( \delta_x(x_t^{-i}) w_t^i(x_t^i, x) \right) \log \bar{q}_t(x) = \sum_x \bar{q}_t(x) \sum_{i=1}^{L} 0 \log \bar{q}_t(x) = 0. \qquad (83)$$

This implies that

$$\frac{\partial H(\bar{q}_t)}{\partial t} = \sum_{x_t} \frac{\partial \bar{q}_t(x_t)}{\partial t} \left( \log \frac{\bar{q}_t(x)}{\bar{q}_t(x_t)} - 1 \right) = \sum_{x_t} \sum_x \bar{q}_t(x) \sum_{i=1}^{L} \delta_x(x_t^{-i}) w_t^i(x_t^i, x) \left( \log \frac{\bar{q}_t(x)}{\bar{q}_t(x_t)} - 1 \right) \qquad (84)$$

$$= \sum_{x_t} \bar{q}_t(x_t) \sum_{i=1}^{L} \sum_{x^i} w_t^i(x_t^i, x) \frac{\bar{q}_t(x)}{\bar{q}_t(x_t)} \left( \log \frac{\bar{q}_t(x)}{\bar{q}_t(x_t)} - 1 \right). \qquad (85)$$

Integrating from time 0 to 1 on both sides, we get

$$H(\bar{q}_1) = H(\bar{q}_0) + \int_0^1 \sum_{x_t} \bar{q}_t(x_t) \sum_{i=1}^{L} \sum_{x^i} w_t^i(x_t^i, x) \frac{\bar{q}_t(x)}{\bar{q}_t(x_t)} \left( \log \frac{\bar{q}_t(x)}{\bar{q}_t(x_t)} - 1 \right) dt. \qquad (86)$$

**Proof of Proposition 4.4:**

Using the same strategy as in Proposition A.1, we can rewrite the inequality in Theorem 4.2, as

$$D_{KL}(\bar{q}_1 \| \bar{p}_1) \leq D_{KL}(\bar{q}_0 \| \bar{p}_0) + \qquad (87)$$

$$\int_0^1 \sum_{x_t} \bar{q}_t(x_t) \sum_{i=1}^{L} \sum_{x^i \neq x_t^i} \left( r_{\bar{q}_t} \tilde{w}_t^i(x_t^i, x) \log \frac{r_{\bar{q}_t} \tilde{w}_t^i(x_t^i, x)}{v_t^i(x^i, x_t)} + v_t^i(x^i, x_t) - r_{\bar{q}_t} \tilde{w}_t^i(x_t^i, x) \right) dt. \qquad (88)$$

where $r_{\bar{q}_t} = r_{\bar{q}_t}(x, x_t) = \frac{\bar{q}_t(x)}{\bar{q}_t(x_t)}$. Expression

$$\int_0^1 \sum_{x_t} \bar{q}_t(x_t) \sum_{i=1}^{L} \sum_{x^i \neq x_t^i} \left( r_{\bar{q}_t} \tilde{w}_t^i(x_t^i, x) \log \frac{r_{\bar{q}_t} \tilde{w}_t^i(x_t^i, x)}{v_t^i(x^i, x_t)} + v_t^i(x^i, x_t) - r_{\bar{q}_t} \tilde{w}_t^i(x_t^i, x) \right) dt \qquad (89)$$

can be rewritten as

$$\int_0^1 \sum_{x_t} \bar{q}_t(x_t) \sum_{i=1}^L \sum_{x^i \neq x_t^i} \left( r_{\bar{q}_t} \tilde{w}_t^i(x_t^i, x) \log r_{\bar{q}_t} - r_{\bar{q}_t} \tilde{w}_t^i(x_t^i, x) \right) dt \tag{90}$$

$$+ \int_0^1 \sum_{x_t} \bar{q}_t(x_t) \sum_{i=1}^L \sum_{x^i \neq x_t^i} \left( r_{\bar{q}_t} \tilde{w}_t^i(x_t^i, x) \log \frac{\tilde{w}_t^i(x_t^i, x)}{v_t^i(x^i, x_t)} + v_t^i(x^i, x_t) \right) dt \tag{91}$$

and therefore as

$$\int_0^1 \sum_{x_t} \bar{q}_t(x_t) \sum_{i=1}^L \sum_{x^i} \left( r_{\bar{q}_t} \tilde{w}_t^i(x_t^i, x) \log r_{\bar{q}_t} - r_{\bar{q}_t} \tilde{w}_t^i(x_t^i, x) \right) dt + \int_0^1 \sum_{x_t} \sum_{i=1}^L \bar{q}_t(x_t) \tilde{w}_t^i(x_t^i, x_t) dt$$

$$\tag{92}$$

$$+ \int_0^1 \sum_{x_t} \bar{q}_t(x_t) \sum_{i=1}^L \sum_{x^i \neq x_t^i} \left( r_{\bar{q}_t} \tilde{w}_t^i(x_t^i, x) \log \frac{\tilde{w}_t^i(x_t^i, x)}{v_t^i(x^i, x_t)} + v_t^i(x^i, x_t) \right) dt. \tag{93}$$

Therefore, the initial Inequality (88), can be rewritten as

$$H(\bar{q}_1, \bar{p}_1) - H(\bar{q}_1) \leq \tag{94}$$

$$\int_0^1 \sum_{x_t} \bar{q}_t(x_t) \sum_{i=1}^L \sum_{x^i} \left( r_{\bar{q}_t} \tilde{w}_t^i(x_t^i, x) \log r_{\bar{q}_t} - r_{\bar{q}_t} \tilde{w}_t^i(x_t^i, x) \right) dt + \int_0^1 \sum_{x_t} \sum_{i=1}^L \bar{q}_t(x_t) \tilde{w}_t^i(x_t^i, x_t) dt$$

$$\tag{95}$$

$$+ \int_0^1 \sum_{x_t} \bar{q}_t(x_t) \sum_{i=1}^L \sum_{x^i \neq x_t^i} \left( r_{\bar{q}_t} \tilde{w}_t^i(x_t^i, x) \log \frac{\tilde{w}_t^i(x_t^i, x)}{v_t^i(x^i, x_t)} + v_t^i(x^i, x_t) \right) dt + D_{KL}(\bar{q}_0 \| \bar{p}_0). \tag{96}$$

and since $\tilde{w}_t^i(x_t^i, x)$ denotes the reverse probability velocity, then

$$H(\bar{q}_0) = H(\bar{q}_1) + \int_0^1 \sum_{x_t} \bar{q}_t(x_t) \sum_{i=1}^L \sum_{x^i} \left( r_{\bar{q}_t} \tilde{w}_t^i(x_t^i, x) \log r_{\bar{q}_t} - r_{\bar{q}_t} \tilde{w}_t^i(x_t^i, x) \right) dt, \tag{97}$$

and therefore we can calculate the cross entropy as follows

$$H(\bar{q}_1, \bar{p}_1) \leq H(\bar{q}_0) - \int_0^1 \sum_{x_t} \bar{q}_t(x_t) \sum_{i=1}^L \sum_{x^i \neq x_t^i} \tilde{w}_t^i(x^i, x_t) dt + D_{KL}(\bar{q}_0 \| \bar{p}_0) + \int_0^1 \sum_{x_0, x_1} \pi(x_0, x_1)$$

$$\tag{98}$$

$$\sum_{x_t} \bar{q}_t(x_t | x_0, x_1) \sum_{i=1}^L \sum_{x^i \neq x_t^i} \left( \frac{\bar{q}_t(x | x_0, x_1)}{\bar{q}_t(x_t | x_0, x_1)} \tilde{w}_t^i(x_t^i, x) \log \frac{\tilde{w}_t^i(x_t^i, x)}{v_t^i(x^i, x_t)} + v_t^i(x^i, x_t) \right) dt. \tag{99}$$

## A.2 First Upper Bound Derivation Details

From

$$- \log p_t(x_1; \theta) \leq$$

$$\int_0^1 \sum_{x_t} p_{t|1}(x_t | x_1) \sum_{i=1}^L \sum_{x^i \neq x_t^i} \left( w_t^i(x^i, x_t) \log \frac{w_t^i(x^i, x_t)}{u_t^i(x^i, x_t; \theta)} + u_t^i(x^i, x_t; \theta) - w_t^i(x^i, x_t) \right) dt, \tag{100}$$

in the case of the special discrete flow matching dynamics from Equation (4), the probability velocity for $\bar{p}_t(x) = p_t(x; \theta)$ is given in Equation (5), with the learned velocity being $u_t^i(x^i, x_t; \theta) = \frac{\dot{k}_t}{1 - k_t} \left[ p_{1|t}(x^i | x_t; \theta) - \delta_{x_t}(x^i) \right]$. The probability velocity for $\bar{q}_t(x) = p_{t|1}(x | x_1)$ can be calculated by first calculating $p_{t|1}^i(x | x_1)$ using $p_{t|1,0}^i(x | x_1, x_0)$ in Equation (4), and finding its probability velocity. However this is not necessary because we notice that for $p_{t|1,0}^i(x^i | x_1, x_0) = (1 - k_t) \delta_{x_0^i}(x^i) + k_t \delta_{x_1^i}(x^i)$ the corresponding probability velocity is $u_t^i(x^i, x_t, |x_0, x_1) = \frac{\dot{k}_t}{1 - k_t} [\delta_{x_1^i}(x^i) -$

$\delta_{x_t^i}(x^i)]$ which does not depend on $x_0$, thus $w_t^i(x^i, x_t) = u_t^i(x^i, x_t, |x_1) = u_t^i(x^i, x_t, |x_0, x_1) = \frac{\dot{k}_t}{1-k_t}[\delta_{x_1^i}(x^i) - \delta_{x_t^i}(x^i)]$. Plugging everything into Expression (13), we get that

$$-\log p_t(x_1; \theta) \leq$$

$$\int_0^1 \frac{\dot{k}_t}{1-k_t} \sum_{x_t} p_{t|1}(x_t|x_1) \sum_{i=1}^L \left( -\delta_{x_1^i \neq x_t^i} \log p_{1|t}^i(x_1^i|x_t; \theta) + 1 - p_{1|t}^i(x_t^i|x_t; \theta) - \delta_{x_1^i \neq x_t^i} \right) dt. \quad (101)$$

Therefore, taking the expectation with respect to $p_1(x_1)$, we find that

$$H(p_1, p_1(\theta)) \leq$$

$$\int_0^1 \frac{\dot{k}_t}{1-k_t} \sum_{x_t, x_1} p_{t,1}(x_t, x_1) \sum_{i=1}^L \left( -\delta_{x_1^i \neq x_t^i} \log p_{1|t}^i(x_1^i|x_t; \theta) + 1 - p_{1|t}^i(x_t^i|x_t; \theta) - \delta_{x_1^i \neq x_t^i} \right) dt. \quad (102)$$

Finally, since the part inside the large brackets is not dependent on $x_0$, we can write

$$H(p_1, p_1(\theta)) \leq \mathcal{B} = \int_0^1 \frac{\dot{k}_t}{1-k_t} \sum_{x_1, x_0} \pi(x_1, x_0) \sum_{x_t} p_{t|1,0}(x_t|x_1, x_0) \sum_{i=1}^L \Big($$

$$-\delta_{x_1^i \neq x_t^i} \log p_{1|t}^i(x_1^i|x_t; \theta) + 1 - p_{1|t}^i(x_t^i|x_t; \theta) - \delta_{x_1^i \neq x_t^i} \Big) dt, \quad (103)$$

hence $e^{\frac{\mathcal{B}}{L}}$ is a computable upper bound of the perplexity in practice, as described in Algorithm 2.

### A.3 ALTERNATIVE UPPER BOUND DERIVATION DETAILS

From

$$H(\bar{q}_1, \bar{p}_1) \leq H(\bar{q}_0) - \sum_{x_t} \bar{q}_t(x_t) \sum_{i=1}^L \sum_{x^i \neq x_t^i} \tilde{w}_t^i(x^i, x_t) + D_{KL}(\bar{q}_0 \| \bar{p}_0) + \int_0^1 \sum_{x_0, x_1} \pi(x_0, x_1)$$

$$\sum_{x_t} \bar{q}_t(x_t|x_0, x_1) \sum_{i=1}^L \sum_{x^i \neq x_t^i} \left( \frac{\bar{q}_t^i(x^i|x_0, x_1)}{\bar{q}_t^i(x_t^i|x_0, x_1)} \tilde{w}_t^i(x_t^i, x) \log \frac{\tilde{w}_t^i(x_t^i, x)}{v_t^i(x^i, x_t)} + v_t^i(x^i, x_t) \right) dt \quad (104)$$

by choosing $\bar{q}_t(x)$ to be the flow $p_t$ defined in Equation (2) with the coupling distribution $\pi(x_0, x_1)$, and defining $\bar{p}_t(x)$ to be the learned approximation of this flow $\bar{p}_t(\theta)$ we have

$$H(p_1, p_1(\theta)) \leq H(p_0) - \int_0^1 \sum_{x_t} p_t(x_t) \sum_{i=1}^L \sum_{x^i \neq x_t^i} \tilde{w}_t^i(x^i, x_t) dt + \int_0^1 \sum_{x_0, x_1} \pi(x_0, x_1)$$

$$\sum_{x_t} p_t(x_t|x_0, x_1) \sum_{i=1}^L \sum_{x^i \neq x_t^i} \left( \frac{p_t^i(x^i|x_0, x_1)}{p_t^i(x_t^i|x_0, x_1)} \tilde{w}_t^i(x_t^i, x) \log \frac{\tilde{w}_t^i(x_t^i, x)}{u_t^i(x^i, x_t; \theta)} + u_t^i(x^i, x_t; \theta) \right) dt. \quad (105)$$

which can be interpreted as the discrete flow counterpart of the bound established for discrete diffusion models in Haxholli et al. (2025).

#### A.3.1 MASKED SOURCE SPECIAL CASE

As shown in Gat et al. (2024), the backward probability velocity $\tilde{w}_t$, can be explicitly computed in some important special cases. For example, if coupling distribution is independent $\pi(x_0, x_1) = p_0(x_0)q_1(x_1)$, and if the source distribution is either the masked distribution, or its dimensions are i.i.d. $p_0(x_0) = \prod_{i=1}^N p_0(x_0^i)$. In these cases,

$$\tilde{w}_t(x^i, x_t) = -\frac{\dot{k}_t}{k_t} \left[ \delta_{x_t^i}(x^i) - p_0^i(x^i) \right]. \quad (106)$$

For the special masked dynamics corresponding to the backward probability velocity $\tilde{w}_t$ in Equation 17, we have the following inequality:

$$H(p_1, p_1(\theta)) \leq \mathcal{B} := \int_0^1 \sum_{x_0, x_1} \pi(x_0, x_1) \sum_{x_t} p_t(x_t | x_0, x_1) \sum_{i=1}^L \left( \frac{\dot{k}_t}{1 - k_t} (1 - p_{1|t}^i(x_t^i, x_t; \theta)) \right.$$

$$\left. - \frac{\dot{k}_t}{k_t}(1 - \delta_m(x_t^i)) - \delta_m(x_t^i)\frac{\dot{k}_t}{1 - k_t} \log\left(\frac{k_t}{1 - k_t}(1 - p_{1|t}^i(x_1^i, x_t; \theta))\right)\right) dt. \tag{107}$$

Indeed, the entropy of the source distribution $H(p_0)$ is 0, as all the mass is concentrated in the masked state. The term

$$\sum_{x^i \neq x_t^i} \tilde{w}_t^i(x^i, x_t) \tag{108}$$

on the other hand can be written as

$$\sum_{x^i \neq x_t^i} \frac{\dot{k}_t}{k_t} \left[ p_0^i(x^i) - \delta_{x_t^i}(x^i) \right] \tag{109}$$

and since $p_0(x^i) = \delta_m(x^i)$ we can discern two cases:

1) $x_t^i \neq m$ implying

$$\sum_{x^i \neq x_t^i} \frac{\dot{k}_t}{k_t} \left[ \delta_m(x^i) - \delta_{x_t^i}(x^i) \right] = \frac{\dot{k}_t}{k_t} \tag{110}$$

2) $x_t^i = m$ implying $x^i \neq m$ and thus

$$\sum_{x^i \neq x_t^i} \frac{\dot{k}_t}{k_t} \left[ \delta_m(x^i) - \delta_{x_t^i}(x^i) \right] = 0. \tag{111}$$

Therefore

$$-\int_0^1 \sum_{x_t} p_t(x_t) \sum_{i=1}^L \sum_{x^i \neq x_t^i} \tilde{w}_t^i(x^i, x_t) dt = -\int_0^1 \sum_{x_t} p_t(x_t) \sum_{i=1}^L \frac{\dot{k}_t}{k_t}(1 - \delta_m(x_t^i)) dt. \tag{112}$$

The following two terms remain:

$$\int_0^1 \sum_{x_0, x_1} \pi(x_0, x_1) \sum_{x_t} p_t(x_t | x_0, x_1) \sum_{i=1}^L \sum_{x^i \neq x_t^i} \left( \frac{p_t^i(x^i | x_0, x_1)}{p_t^i(x_t^i | x_0, x_1)} \tilde{w}_t^i(x_t^i, x) \log \frac{\tilde{w}_t^i(x_t^i, x)}{u_t^i(x^i, x_t; \theta)} \right) dt \tag{113}$$

and

$$\int_0^1 \sum_{x_0, x_1} \pi(x_0, x_1) \sum_{x_t} p_t(x_t | x_0, x_1) \sum_{i=1}^L \sum_{x^i \neq x_t^i} u_t^i(x^i, x_t; \theta) dt. \tag{114}$$

The last part of the first one

$$\sum_{x^i \neq x_t^i} \left( \frac{p_t^i(x^i | x_0, x_1)}{p_t^i(x_t^i | x_0, x_1)} \tilde{w}_t^i(x_t^i, x) \log \frac{\tilde{w}_t^i(x_t^i, x)}{u_t^i(x^i, x_t; \theta)} \right) dt \tag{115}$$

can be rewritten as follows:

$$\sum_{x^i \neq x_t^i} \frac{(1 - k_t)\delta_{x_0^i}(x^i) + k_t\delta_{x_1^i}(x^i)}{(1 - k_t)\delta_{x_0^i}(x_t^i) + k_t\delta_{x_1^i}(x_t^i)} \frac{\dot{k}_t}{k_t} \left[ \delta_m(x_t^i) - \delta_{x^i}(x_t^i) \right] \log \frac{\frac{\dot{k}_t}{k_t} \left[ \delta_m(x_t^i) - \delta_{x^i}(x_t^i) \right]}{\frac{\dot{k}_t}{1 - k_t} \left[ p_{1|t}^i(x^i, x_t; \theta) - \delta_{x_t^i}(x^i) \right]}. \tag{116}$$

As before we can distinguish two cases:

1) $x_t^i \neq m$, which combined with $x_0^i = m$ and $x_1^i \neq m$ gives

$$\sum_{x^i \neq x_t^i} \frac{(1-k_t)\delta_{x_0^i}(x^i) + k_t\delta_{x_1^i}(x^i)}{(1-k_t)\delta_{x_0^i}(x_t^i) + k_t\delta_{x_1^i}(x_t^i)} \frac{\dot{k}_t}{k_t} \left[\delta_m(x_t^i) - \delta_{x^i}(x_t^i)\right] \log \frac{\frac{\dot{k}_t}{k_t}\left[\delta_m(x_t^i) - \delta_{x^i}(x_t^i)\right]}{\frac{\dot{k}_t}{1-k_t}\left[p_{1|t}^i(x^i, x_t; \theta) - \delta_{x_t^i}(x^i)\right]}$$

$$= \sum_{x^i \neq x_t^i} \left(\frac{(1-k_t)\delta_{x_0^i}(x_t^i)}{k_t\delta_{x_1^i}(x_t^i)} + 1\right) \frac{\dot{k}_t}{k_t} [0-0] \log \frac{\frac{\dot{k}_t}{k_t}\left[\delta_m(x_t^i) - \delta_{x^i}(x_t^i)\right]}{\frac{\dot{k}_t}{1-k_t}\left[p_{1|t}^i(x^i, x_t; \theta) - \delta_{x_t^i}(x^i)\right]} = 0. \quad (117)$$

2) $x_t^i = m$ implying $x^i \neq m$, which combined with $x_0^i = m$ and $x_1^i \neq m$ gives

$$\sum_{x^i \neq x_t^i} \frac{(1-k_t)\delta_{x_0^i}(x^i) + k_t\delta_{x_1^i}(x^i)}{(1-k_t)\delta_{x_0^i}(x_t^i) + k_t\delta_{x_1^i}(x_t^i)} \frac{\dot{k}_t}{k_t} \left[\delta_m(x_t^i) - \delta_{x^i}(x_t^i)\right] \log \frac{\frac{\dot{k}_t}{k_t}\left[\delta_m(x_t^i) - \delta_{x^i}(x_t^i)\right]}{\frac{\dot{k}_t}{1-k_t}\left[p_{1|t}^i(x^i, x_t; \theta) - \delta_{x_t^i}(x^i)\right]}$$

$$= \sum_{x^i \neq x_t^i} \frac{k_t\delta_{x_1^i}(x^i)}{1-k_t} \frac{\dot{k}_t}{k_t} \log \frac{\frac{\dot{k}_t}{k_t}}{\frac{\dot{k}_t}{1-k_t}p_{1|t}(x^i|x_t)} = \sum_{x^i \neq x_t^i} \frac{\dot{k}_t}{1-k_t}\delta_{x_1^i}(x^i) \log \frac{1-k_t}{k_t p_{1|t}^i(x^i, x_t; \theta)}$$

$$= -\frac{\dot{k}_t}{1-k_t} \log \frac{k_t}{1-k_t} p_{1|t}^i(x_1^i, x_t; \theta). \quad (118)$$

Combining these two cases, one concludes that

$$\int_0^1 \sum_{x_0, x_1} \pi(x_0, x_1) \sum_{x_t} p_t(x_t|x_0, x_1) \sum_{i=1}^L \sum_{x^i \neq x_t^i} \left(\frac{p_t^i(x^i|x_0, x_1)}{p_t^i(x_t^i|x_0, x_1)} \tilde{w}_t^i(x_t^i, x) \log \frac{\tilde{w}_t^i(x_t^i, x)}{u_t^i(x^i, x_t; \theta)}\right) dt$$

$$= -\int_0^1 \sum_{x_0, x_1} \pi(x_0, x_1) \sum_{x_t} p_t(x_t|x_0, x_1) \sum_{i=1}^L \delta_m(x_t^i) \frac{\dot{k}_t}{1-k_t} \log \frac{k_t}{1-k_t} p_{1|t}^i(x_1^i, x_t; \theta). \quad (119)$$

Finally, we derive the last term

$$\int_0^1 \sum_{x_0, x_1} \pi(x_0, x_1) \sum_{x_t} p_t(x_t|x_0, x_1) \sum_{i=1}^L \sum_{x^i \neq x_t^i} u_t^i(x^i, x_t; \theta). \quad (120)$$

As before

$$\sum_{x^i \neq x_t^i} u_t^i(x^i, x_t; \theta) = \frac{\dot{k}_t}{1-k_t} \sum_{x^i \neq x_t^i} \left[p_{1|t}^i(x^i, x_t; \theta) - \delta_{x_t^i}(x^i)\right]$$

$$\frac{\dot{k}_t}{1-k_t} \sum_{x^i \neq x_t^i} p_{1|t}^i(x^i, x_t; \theta) = \frac{\dot{k}_t}{1-k_t}\left(1 - p_{1|t}^i(x_t^i, x_t; \theta)\right).$$

Putting everything together we conclude that

$$H(p_1, p_1(\theta)) \leq$$

$$\mathcal{B} = \int_0^1 \sum_{x_0, x_1} \pi(x_0, x_1) \sum_{x_t} p_t(x_t|x_0, x_1) \sum_{i=1}^L \left(-\frac{\dot{k}_t}{k_t}(1 - \delta_m(x_t^i)) - \delta_m(x_t^i)\frac{\dot{k}_t}{1-k_t} \log \frac{k_t}{1-k_t}\right.$$

$$\left. +\frac{\dot{k}_t}{1-k_t}(1 - p_{1|t}^i(x_t^i, x_t; \theta)) - \delta_m(x_t^i)\frac{\dot{k}_t}{1-k_t} \log p_{1|t}^i(x_1^i, x_t; \theta) dt\right). \quad (121)$$

We can go a step further and calculate the expression

$$\int_0^1 \sum_{x_0, x_1} \pi(x_0, x_1) \sum_{x_t} p_t(x_t|x_0, x_1) \sum_{i=1}^L \left(-\frac{\dot{k}_t}{k_t}(1 - \delta_m(x_t^i)) - \delta_m(x_t^i)\frac{\dot{k}_t}{1-k_t} \log \frac{k_t}{1-k_t}\right) dt$$

$$= \int_0^1 \sum_{x_0, x_1} \pi(x_0, x_1) \sum_{i=1}^L \sum_{x_t} p_t(x_t|x_0, x_1) \left(-\frac{\dot{k}_t}{k_t}(1 - \delta_m(x_t^i)) - \delta_m(x_t^i)\frac{\dot{k}_t}{1-k_t} \log \frac{k_t}{1-k_t}\right) dt$$

$$= L \int_0^1 \left( -\dot{k}_t - \dot{k}_t \log \frac{k_t}{1 - k_t} \right) dt = -L \int_0^1 \dot{k}_t dt$$

$$= -\int_0^1 \sum_{x_0, x_1} \pi(x_0, x_1) \sum_{x_t} p_t(x_t | x_0, x_1) \sum_{i=1}^L \frac{\dot{k}_t}{1 - k_t} \delta_m(x_t^i).$$

Plugging this into $\mathcal{B}$, we get:

$$\mathcal{B} = \int_0^1 \sum_{x_0, x_1} \pi(x_0, x_1) \sum_{x_t} p_t(x_t | x_0, x_1) \sum_{i=1}^L \left( -\frac{\dot{k}_t}{1 - k_t} \delta_m(x_t^i) + \right.$$

$$\left. \frac{\dot{k}_t}{1 - k_t} (1 - p_{1|t}^i(x_t^i, x_t; \theta)) - \delta_m(x_t^i) \frac{\dot{k}_t}{1 - k_t} \log p_{1|t}^i(x_1^i, x_t; \theta) dt \right). \tag{122}$$

In this special dynamic of the masked flow, $x_t^i = m$ is equivalent to $x_t^i \neq x^i$, therefore the bound above matches the first bound:

$$\mathcal{B} = \int_0^1 \frac{\dot{k}_t}{1 - k_t} \sum_{x_0, x_1} \pi(x_0, x_1) \sum_{x_t} p_t(x_t | x_0, x_1) \sum_{i=1}^L \left( -\delta_{x_t^i \neq x^i} + \right.$$

$$\left. (1 - p_{1|t}^i(x_t^i, x_t; \theta)) - \delta_{x_t^i \neq x^i} \log p_{1|t}^i(x_1^i, x_t; \theta) dt \right). \tag{123}$$

### A.4 MD4 SPECIAL CASE

We can define our model $p_{1|t}(x^i, x_t; \theta)$ in the previous subsection to be such that if a given position has been unmasked we always predict that unmasked token. This implies that $p_{1|t}(x_t^i, x_t; \theta) = 1$ when $x_t^i \neq m$. This implies that Equation (122) becomes

$$\mathcal{B} = \int_0^1 \frac{\dot{k}_t}{1 - k_t} \sum_{x_0, x_1} \pi(x_0, x_1) \sum_{x_t} p_t(x_t | x_0, x_1) \sum_{i=1}^L \left( -\delta_m(x_t^i) + \right.$$

$$\left. \delta_m(x_t^i)(1 - p_{1|t}^i(x_t^i, x_t; \theta)) - \delta_m(x_t^i) \log p_{1|t}^i(x_1^i, x_t; \theta) dt \right), \tag{124}$$

that is

$$\mathcal{B} = \int_0^1 \frac{\dot{k}_t}{1 - k_t} \sum_{x_0, x_1} \pi(x_0, x_1) \sum_{x_t} p_t^i(x_t | x_0, x_1) \sum_{i=1}^L \left( -\delta_m(x_t^i) p_{1|t}^i(x_t^i, x_t; \theta) + \right.$$

$$\left. -\delta_m(x_t^i) \log p_{1|t}^i(x_1^i, x_t; \theta) dt \right). \tag{125}$$

However, we can set the probability of $p_{1|t}(m, x_t; \theta)$ to zero, as we know that there are no masked tokens in the data distribution, which implies,

$$\mathcal{B} = \int_0^1 \frac{\dot{k}_t}{1 - k_t} \sum_{x_0, x_1} \pi(x_0, x_1) \sum_{x_t} p_t(x_t | x_0, x_1) \sum_{i=1}^L \left( -\delta_m(x_t^i) \log p_{1|t}^i(x_1^i, x_t; \theta) dt \right). \tag{126}$$

The final bound was originally derived in Shaul et al. (2025) and is simply MD4 from Shi et al. (2024).

### A.5 THE PRECISE PERPLEXITY

Given the equation in Proposition A.1,

$$D_{KL}(\bar{q}_1 \| \bar{p}_1) = D_{KL}(\bar{q}_0 \| \bar{p}_0) +$$

$$\int_0^1 \sum_{x_t} \bar{q}_t(x_t) \sum_{i=1}^L \sum_{x^i \neq x_t^i} \left( w_t^i(x^i, x_t) \log \frac{w_t^i(x^i, x_t)}{v_t^i(x^i, x_t)} + v_t^i(x^i, x_t) - w_t^i(x^i, x_t) \right) dt -$$

$$\int_0^1 \sum_{x_t} \bar{q}_t(x_t) \sum_{i=1}^L \sum_{x^i \neq x_t^i} \left( r_{\bar{q}_t} w_t^i(x_t^i, x) \log \frac{r_{\bar{q}_t} w_t^i(x_t^i, x)}{r_{\bar{p}_t} v_t^i(x_t^i, x)} + r_{\bar{p}_t} v_t^i(x_t^i, x) - r_{\bar{q}_t} w_t^i(x_t^i, x) \right) dt, \quad (127)$$

as in the main text, we choose $\bar{q}_t(x)$ to have the dynamics of the flow $p_t$, but with the coupling distribution $\bar{\pi}(x, y) = p_0(x) \delta_{x_1}(y) = \int \pi(x, z) dz \delta_{x_1}(y)$. Clearly, we have $\bar{q}_0(x) = p_0(x)$, $\bar{q}_1(x) = \delta_{x_1}(x)$ and $\bar{q}_t(x) = p_{t|1}(x|x_1)$.

We notice that since $\bar{q}_0(x) = p_0(x)$ and $\bar{p}_0(x) = p_0(x)$, then $D_{KL}(\bar{q}_0 \| \bar{p}_0) = 0$. Furthermore $D_{KL}(\bar{q}_1(x) \| \bar{p}_1(x)) = D_{KL}(\delta_{x_1}(x) \| p_1(x; \theta)) = -\log p_1(x_1; \theta)$. Thus for such particular choices one gets that

$$-\log p_t(x_1; \theta) =$$

$$\int_0^1 \sum_{x_t} p_{t|1}(x_t|x_1) \sum_{i=1}^L \sum_{x^i \neq x_t^i} \left( w_t^i(x^i, x_t) \log \frac{w_t^i(x^i, x_t)}{u_t^i(x^i, x_t; \theta)} + u_t^i(x^i, x_t; \theta) - w_t^i(x^i, x_t) \right) dt -$$

$$\int_0^1 \sum_{x_t} p_{t|1}(x_t) \sum_{i=1}^L \sum_{x^i \neq x_t^i} \left( r_{p_{t|1}} w_t^i(x_t^i, x) \log \frac{r_{p_{t|1}} w_t^i(x_t^i, x)}{r_{p_t^\theta} v_t^i(x_t^i, x)} + r_{p_t^\theta} v_t^i(x_t^i, x) - r_{p_{t|1}} w_t^i(x_t^i, x) \right) dt,$$

$$(128)$$

where $r_{p_t^\theta} = r_{p_t^\theta}(x, x_t) = \frac{p_t^\theta(x)}{p_t^\theta(x_t)}$, $r_{p_{t|1}} = r_{p_{t|1}}(x, x_t) = \frac{p_{t|1}(x|x_1)}{p_{t|1}(x_t|x_1)}$.

By using the same strategy as in the proof of Proposition 4.4, we get that

$$-\log p_t(x_1; \theta) =$$

$$\int_0^1 \sum_{x_t} p_{t|1}(x_t|x_1) \sum_{i=1}^L \sum_{x^i \neq x_t^i} \left( w_t^i(x^i, x_t) \log \frac{w_t^i(x^i, x_t)}{u_t^i(x^i, x_t; \theta)} + u_t^i(x^i, x_t; \theta) - w_t^i(x^i, x_t) \right) dt$$

$$-H(p_0) + \int_0^1 \sum_{x_t} p_{t|1}(x_t|x_1) \sum_{i=1}^L \sum_{x^i \neq x_t^i} \tilde{w}_t^i(x^i, x_t) dt - \quad (129)$$

$$\int_0^1 \sum_{x_t} p_{t|1}(x_t|x_1) \sum_{i=1}^L \sum_{x^i \neq x_t^i} \left( \frac{p_{t|1}(x|x_1)}{p_{t|1}(x_t|x_1)} w_t^i(x_t^i, x) \log \frac{w_t^i(x_t^i, x)}{r_{p_t^\theta} u_t^i(x_t^i, x; \theta)} + r_{p_t^\theta} u_t^i(x_t^i, x; \theta) \right) dt.$$

$$(130)$$

Taking the expectation with respect to the data distribution

$$H(p_1, p_1(\theta)) = -H(p_0) +$$

$$\int_0^1 \sum_{x_0, x_1} \pi(x_0, x_1) p_{t|1,0}(x_t|x_1, x_0) \sum_{i=1}^L \sum_{x^i \neq x_t^i} \left[ \tilde{w}_t^i(x^i, x_t) + \right.$$

$$\left( w_t^i(x^i, x_t) \log \frac{w_t^i(x^i, x_t)}{u_t^i(x^i, x_t; \theta)} + u_t^i(x^i, x_t; \theta) - w_t^i(x^i, x_t) \right) \quad (131)$$

$$-\left( \frac{p_{t|1}^i(x^i|x_1, x_0)}{p_{t|1}^i(x_t^i|x_1, x_0)} w_t^i(x_t^i, x) \log \frac{w_t^i(x_t^i, x)}{r_{p_t^\theta} u_t^i(x^i, x_t; \theta)} + r_{p_t^\theta} u_t^i(x^i, x_t; \theta) \right) \right] dt. \quad (132)$$

The only terms above that we do not have an explicit form of are the learned-probability ratios between neighbor states. These are the terms missing if we tried to directly calculate the loglikelihood at a point using the continuity equation,

$$\frac{\partial \log p_t^\theta(x_1)}{\partial t} = \frac{1}{p_t^\theta(x_1)} \frac{\partial p_t^\theta(x_1)}{\partial t} = \sum_x \frac{p_t^\theta(x)}{p_t^\theta(x_1)} \sum_{i=1}^L \delta_x(x_1^{-i}) u_t^i(x_1^i, x; \theta). \quad (133)$$

### A.6 TIME-INDEPENDENCE OF PREDICTIVE PROBABILITIES IN MULTIMASKED FLOWS

The following proposition is a generalization of that given in Gat et al. (2024) for masked flows.

**Proposition A.2.** *In the case of multimasked flows, predictive probabilities $p_{1|t}(x^i \mid x_t)$ are time independent.*

*Proof.* From

$$p_{t|0,1}^i(z^i|x_0, x_1) = (1 - k_t)\delta_{x_0^i}(z^i) + k_t\delta_{x_1^i}(z^i) = \begin{cases} (1 - k_t)\delta_{x_0^i}(z^i), & z^i \leq V, \\ k_t\delta_{x_1^i}(z^i), & z^i > V. \end{cases} \tag{134}$$

we have that

$$p_{t|0,1}(z \mid x_0, x_1) = \left[ \prod_{i:\, z^i > V} (1 - k_t) \right] \left[ \prod_{i:\, z^i \leq V} k_t \right] \left[ \prod_{i:\, z^i > V} \delta_{x_0^i}(z^i) \right] \left[ \prod_{i:\, z^i \leq V} \delta_{x_1^i}(z^i) \right] \tag{135}$$

$$p_{0,1|t}(x_0, x_1 \mid z) = \frac{p_{t|0,1}(z \mid x_0, x_1)p(x_0, x_1)}{p_t(z)} = \frac{p_{t|0,1}(z \mid x_0, x_1)p(x_0, x_1)}{\sum_{\tilde{x}_0, \tilde{x}_1} p_{t|0,1}(z \mid \tilde{x}_0, \tilde{x}_1)p(\tilde{x}_0, \tilde{x}_1)} \tag{136}$$

$$= \frac{\left( \left[ \prod_{i:\, z^i > V}(1 - k_t) \right] \left[ \prod_{i:\, z^i \leq V} k_t \right] \right) \left[ \prod_{i:\, z^i > V} \delta_{x_0^i}(z^i) \right] \left[ \prod_{i:\, z^i \leq V} \delta_{x_1^i}(z^i) \right] p(x_0, x_1)}{\sum_{\tilde{x}_0, \tilde{x}_1} \left( \left[ \prod_{i:\, z^i > V}(1 - k_t) \right] \left[ \prod_{i:\, z^i \leq V} k_t \right] \right) \left[ \prod_{i:\, z^i > V} \delta_{\tilde{x}_0^i}(z^i) \right] \left[ \prod_{i:\, z^i \leq V} \delta_{\tilde{x}_1^i}(z^i) \right] p(\tilde{x}_0, \tilde{x}_1)} \tag{137}$$

$$= \frac{\left[ \prod_{i:\, z^i > V} \delta_{x_0^i}(z^i) \right] \left[ \prod_{i:\, z^i \leq V} \delta_{x_1^i}(z^i) \right] p(x_0, x_1)}{\sum_{\tilde{x}_0, \tilde{x}_1} \left[ \prod_{i:\, z^i > V} \delta_{\tilde{x}_0^i}(z^i) \right] \left[ \prod_{i:\, z^i \leq V} \delta_{\tilde{x}_1^i}(z^i) \right] p(\tilde{x}_0, \tilde{x}_1)} = p_{0,1}(x_0, x_1 \mid z) \tag{138}$$

which does not depend on $t$. Thus

$$p_{1|t}(x^i \mid z) = \sum_{x_0, x_1} \delta_{x_1}(x^i)p_{0,1|t}(x_0, x_1 \mid z) = \sum_{x_0, x_1} \delta_{x_1}(x^i)p_{0,1}(x_0, x_1 \mid z) = p_1(x^i \mid z) \tag{139}$$

does not depend on time either. $\square$

# B    ALGORITHMS

---

**Algorithm 1** Discrete Flow Matching with OT Minibatches

---

**Input:** Set of samples $\mathcal{D}$ from $\pi(x_0, x_1)$, model $p_{1|t}^i(x^i|x_t; \theta)$

**repeat**

  1) Sample minibatch $\mathcal{D}_j$ from $\mathcal{D}$.

  2) $\bar{\pi}(x, y) \leftarrow \text{OT}(\mathcal{D}_j)$, s.t. $p(x) = \sum_{y \in \mathcal{D}_j} \bar{\pi}(x, y) = \frac{1}{|\mathcal{D}_j|}$, $q(y) = \sum_{x \in \mathcal{D}_j} \bar{\pi}(x, y) = \frac{1}{|\mathcal{D}_j|}$.

  3) Sample $t$ form $U(0, 1)$.

  4) Sample $x_0, x_1$ from $\bar{\pi}(x_0, x_1)$.

  5) Sample $x_t$ using Equation (4).

  6) Calculate the gradient of the loss $\mathcal{L}$ (e.g. Expression (7))

  7) Update parameters $\theta$

**until** Convergence or stopping criterion

---

**Algorithm 2** Computing the perplexity upper bound

---

**Input:** samples from $\pi(x_0, x_1)$, model $p_{1|t}^i(x^i|x_t; \theta)$

Initialize an empty array: $\mathcal{A} = []$

**repeat**

  1) Sample $t$ form $U(0, 1)$.

  2) Sample $x_0, x_1$ from $\pi(x_0, x_1)$.

  3) Sample $x_t$ using Equation (4).

  4) Append $\frac{\dot{k}_t}{1-k_t} \sum_{i=1}^L \left( -\delta_{x_1^i \neq x_t^i} \log p_{1|t}^i(x_1^i|x_t; \theta) + 1 - p_{1|t}^i(x_t^i|x_t; \theta) - \delta_{x_1^i \neq x_t^i} \right)$ to array $\mathcal{A}$.

**until** Test set is exhausted

Return $\exp\left(\frac{\text{average}(\mathcal{A})}{L}\right)$

---

**Algorithm 3** Computing the alternative perplexity bound

---

**Input:** samples from $\pi(x_0, x_1)$, modeled $u_t^i(x^i, x_t; \theta)$, backward probability velocity $\tilde{w}_t$

Initialize an empty array: $\mathcal{A} = []$

**repeat**

  1) Sample $t$ form $U(0, 1)$.

  2) Sample $x_0, x_1$ from $\pi(x_0, x_1)$.

  3) Sample $x_t$ using Equation (4).

  4) Append $\sum_{i=1}^L \sum_{x^i \neq x_t^i} \left( u_t^i(x^i, x_t; \theta) - \tilde{w}_t^i(x^i, x_t) + \frac{p_t^i(x^i|x_0^i, x_1^i)}{p_t^i(x_t^i|x_0^i, x_1^i)} \tilde{w}_t^i(x_t^i, x) \log \frac{\tilde{w}_t^i(x_t^i, x)}{u_t^i(x^i, x_t; \theta)} \right)$

  to array $\mathcal{A}$.

**until** Test set is exhausted

Return $\exp\left(\frac{H(p_0) + \text{average}(\mathcal{A})}{L}\right)$

---

## C    ADDITIONAL EXPERIMENTAL RESULTS

The foundational architecture of the model we use is based on the diffusion transformer paradigm outlined by Peebles & Xie (2023), which adapts the classic encoder-only transformer structure, such as that introduced in Vaswani et al. (2017); Devlin et al. (2019), by incorporating time-based conditioning. This approach introduces slight architectural modifications, notably the use of rotary positional embeddings as described in Su et al. (2024). Due to the addition of time conditioning, the model's parameter count is approximately 5% higher than that of a typical transformer (e.g., GPT-2). Tokenization and dataset splits are kept consistent with previous work to maintain comparability and minimize confounding variables.

The architecture comprises 12 transformer layers, each equipped with 12 attention heads and a hidden dimensionality of 768, matching the configuration commonly referred to as GPT-2. A dedicated conditioning dimension of 128 is used to capture temporal features essential to the diffusion process. It utilizes conventional scaled dot-product attention and applies a dropout rate of 0.1 to counter overfitting.

Regarding the training setup for OWT experiments, each model was trained with sequence lengths of 128 using a single H200 GPU. The vocabulary includes 50,257 tokens, and the training batch size is fixed at 512. The training schedule encompasses 400,000 steps, and takes 44 hours in the standards case, which increases to 45 when using OT.

The OpenWebText dataset serves as the primary training corpus with local data storage employed to reduce latency. In all cases, we use the schedule $k_t = \epsilon + (1 - \epsilon)t$ with parameter $\epsilon = 0.001$, consistent with settings from Lou et al. (2024). Evaluation samples are generated using 128 or 1024 steps.

Optimization is handled via the AdamW algorithm, set with a learning rate of 3e-4, beta values of (0.9, 0.999), and an epsilon of 1e-8. No weight decay is used, favoring pure learning rate dynamics. A warm-up phase of 2,500 steps is included to enhance training stability, and gradient clipping is applied at a value of 1.

### C.1    CHARACTER LEVEL SHAKESPEARE EXPERIMENT

Table 5 presents the results of the experiment described in Section 5.1, with the sole modification that the training set consists of the original Shakespeare text, without conversion to Morse code.

Table 5: Using minibatch OT reduces the number of jumps by $\sim 5\%$.

| Model (L=128) | Jumps | Relative Jumps | Perplexity |
|---|---|---|---|
| Normal | $113.23 \pm 0.002$ | 1.05 | 5.31 |
| With OT | $\mathbf{107.41} \pm 0.002$ | **1** | **4.89** |

### C.2    TRAINING BOUND COMPARISONS

We train flows wherein the source distribution is chosen to be the Dirac delta at the sequence of all masked tokens. We choose $k_t = t$ in all cases. We tried 3 settings:

a) DFM-O uses cross entropy as the optimization objective as in Gat et al. (2024): $\int_0^1 \sum_{x_1,x_0} \pi(x_1,x_0) \sum_{x_t} p_{t|1,0}(x_t|x_1,x_0) \sum_{i=1}^L [-\log p_{1|t}^i(x_1^i|x_t;\theta)]dt$.

b) DFM-S is the flow matching approach which uses the bound in Equation 16 (as simplified in Appendix A.4): $\int_0^1 \frac{1}{1-t} \sum_{x_1,x_0} \pi(x_1,x_0) \sum_{x_t} p_{t|1,0}(x_t|x_1,x_0) \sum_{i=1}^L -\delta_m(x_t^i) \log p_{1|t}^i(x_1^i|x_t;\theta)dt$.

c) DFM-N is the same as DFM-S but multiplied by $(1 - t)$: $\int_0^1 \sum_{x_1,x_0} \pi(x_1,x_0) \sum_{x_t} p_{t|1,0}(x_t|x_1,x_0) \sum_{i=1}^L -\delta_m(x_t^i) \log p_{1|t}^i(x_1^i|x_t;\theta)dt$.

The model architecture in all cases is identical in design as the one in Section 5.1, but here we use the GPT2 tokenizer and to match related work, we train on OWT (Gokaslan & Cohen, 2019) for 400K

steps with batch size of 512, sequence length of 128. For 'DFM-S', our bound becomes the MD4 of Shi et al. (2024) (see Appendix A.4). The bound is tested on the test sets found in Lou et al. (2024), more precisely: 1BW, LAMBADA, PTB, Wikitext2 and Wikitext103 (Chelba et al., 2013; Paperno et al., 2016; Marcus et al., 1993; Merity et al., 2016). In addition, we compare against SEDD of Lou et al. (2024). The results can be below in Table 6.

Table 6: Perplexity bound results.

| Model (L=128) | Lambada | Wikitext2 | PTB | Wikitext103 | LM1B |
|---|---|---|---|---|---|
| SEDD Absorb | 67.06 | 69.39 | 208.67 | 69.18 | 83.86 |
| DFM-O | 71.90 | 71.23 | 221.62 | 70.80 | 82.60 |
| DFM-N | 67.50 | **67.00** | **204.80** | **66.65** | **80.29** |
| DFM-S | **66.61** | 68.48 | 208.37 | 68.04 | 81.46 |

## C.3 SECTION 5.3 PERPLEXITY BOUND RESULTS

In Table 7 and 8, we provide the perplexity bound results on the five test sets for the models described in Section 5.3.

Table 7: DFM-B perplexity bound results comparing normal training vs OT.

| Dataset | Lambada | Wiki2 | PTB | Wiki3 | LM1B |
|---|---|---|---|---|---|
| DFM-B | **184.81** | 211.66 | 723.15 | 207.73 | 230.87 |
| DFM-B-OT | 190.21 | **204.16** | **654.88** | **204.22** | **222.42** |

Note that bound estimation for OT-trained models is problematic, as minibatch OT defines an implicit coupling we cannot access. Since sampling from this coupling during the calculation of the bound is impossible, we approximate it by sampling minibatches and performing OT on them. This heuristic approach makes such values only approximations.

Table 8: DFM-MMLM perplexity bound results comparing normal training vs OT. Sinkhorn Solver at test time.

| Dataset | Lambada | Wiki2 | PTB | Wiki3 | LM1B |
|---|---|---|---|---|---|
| DFM-O | 71.90 | 71.23 | 221.62 | 70.80 | 82.60 |
| DFM-MMLM | 68.65 | **68.38** | 204.17 | **68.68** | 85.09 |
| DFM-MMLM-OT | **68.63** | 69.33 | **204.06** | 69.21 | **83.45** |

When we use exact OT during testing to get a better estimation of the optimal minibatch coupling and remove the potential repetition of testing samples we get the results in Table 9.

Table 9: DFM-MMLM perplexity bound results comparing normal training vs OT. Exact OT solver at test time.

| Dataset | Lambada | Wiki2 | PTB | Wiki3 | LM1B |
|---|---|---|---|---|---|
| DFM-O | 71.90 | 71.23 | 221.62 | 70.80 | 82.60 |
| DFM-MMLM | 68.65 | **68.38** | 204.17 | **68.68** | 85.09 |
| DFM-MMLM-OT | **68.37** | 69.10 | **202.63** | 69.02 | **83.05** |

C.4  SECTION 5.3 LLAMA-JUDGED GENERATIVE PERPLEXITY, ENTROPY AND STANDARD
     DEVIATIONS

In what follows we present the full generative perplexity results of the experiments described in Section 5.3. That is, we show show resutls when Llama is used as a judge, the entropy values and standard deviations. Such resulrs can be found in tables 10, 11 and 12.

Table 10: Results with and without minibatch OT. GPT-2 Large was used as a judge.

| Generation Steps: | 8 | 16 | 32 | 64 | 128 | 1024 |
|---|---|---|---|---|---|---|
| DFM-B | 345.94 | 241.16 | 211.99 | 197.48 | 191.48 | 185.83 |
| Standard deviation | ±1.71 | ±1.32 | ±1.12 | ±1.10 | ±1.04 | ±1.04 |
| DFM-B-OT | 331.88 | 233.24 | 203.08 | 191.17 | 185.54 | 178.24 |
| Standard deviation | ±1.67 | ±1.26 | ±1.06 | ±1.00 | ±1.01 | ±0.96 |
| DFM-S | 587.80 | 316.25 | 222.46 | 188.62 | 169.81 | 156.81 |
| Standard deviation | ±3.35 | ±1.85 | ±1.39 | ±1.23 | ±1.04 | ±0.97 |
| DFM-N | 556.73 | 296.25 | 210.11 | 176.34 | 160.17 | 147.07 |
| Standard deviation | ±3.17 | ±1.73 | ±1.21 | ±1.08 | ±1.01 | ±0.91 |
| DFM-O | 560.67 | 300.06 | 208.06 | 175.59 | 159.03 | 146.54 |
| Standard deviation | ±3.17 | ±1.78 | ±1.20 | ±1.08 | ±1.01 | ±0.89 |
| DFM-MMLM | 536.50 | 288.38 | 204.77 | 170.61 | 155.45 | 143.48 |
| Standard deviation | ±2.92 | ±1.65 | ±1.16 | ±1.02 | ±0.85 | ±0.95 |
| DFM-MMLM-OT | 525.83 | 283.10 | 199.55 | 167.86 | 153.51 | 141.92 |
| Standard deviation | ±2.87 | ±2.09 | ±1.31 | ±1.02 | ±0.95 | ±0.88 |

Table 11: Results with and without minibatch OT. LLama 3.1 8B was used as a judge.

| Generation Steps: | 8 | 16 | 32 | 64 | 128 | 1024 |
|---|---|---|---|---|---|---|
| DFM-B | 394.67 | 283.71 | 252.04 | 235.98 | 231.61 | 223.77 |
| Standard deviation | ±1.96 | ±1.66 | ±1.54 | ±1.50 | ±1.43 | ±1.41 |
| DFM-B-OT | 380.29 | 274.68 | 243.92 | 230.36 | 225.36 | 216.48 |
| Standard deviation | ±1.96 | ±1.51 | ±1.51 | ±1.42 | ±1.48 | ±1.41 |
| DFM-S | 681.89 | 378.98 | 271.73 | 231.96 | 212.22 | 198.35 |
| Standard deviation | ±3.97 | ±2.34 | ±1.83 | ±1.68 | ±1.60 | ±1.57 |
| DFM-N | 645.79 | 359.97 | 256.33 | 218.68 | 197.46 | 184.23 |
| Standard deviation | ±3.71 | ±2.15 | ±1.61 | ±1.63 | ±1.37 | ±1.40 |
| DFM-O | 652.05 | 361.53 | 253.53 | 217.16 | 198.60 | 184.14 |
| Standard deviation | ±3.76 | ±2.29 | ±1.59 | ±1.50 | ±1.54 | ±1.44 |
| DFM-MMLM | 621.39 | 345.53 | 249.95 | 210.75 | 195.65 | 179.49 |
| Standard deviation | ±3.10 | ±2.07 | ±1.55 | ±1.30 | ±1.65 | ±1.31 |
| DFM-MMLM-OT | 620.84 | 348.39 | 243.31 | 210.87 | 191.42 | 178.62 |
| Standard deviation | ±3.37 | ±1.96 | ±2.08 | ±1.33 | ±1.17 | ±1.45 |

Finally we show that entropy remains unchanged, unlike in the case of improper sampling of SEDD, in which the entropy was shown to drop up to 20% (Zheng et al., 2025).

Table 12: Entropy results with and without minibatch OT.

| Generation Steps: | 8 | 16 | 32 | 64 | 128 | 1024 |
|---|---|---|---|---|---|---|
| DFM-B | 6.30 | 6.27 | 6.26 | 6.25 | 6.25 | 6.25 |
| Standard deviation | ±0.001 | ±0.001 | ±0.001 | ±0.001 | ±0.001 | ±0.001 |
| DFM-B-OT | 6.30 | 6.27 | 6.26 | 6.26 | 6.25 | 6.25 |
| Standard deviation | ±0.001 | ±0.001 | ±0.001 | ±0.001 | ±0.001 | ±0.001 |
| DFM-S | 6.36 | 6.32 | 6.29 | 6.27 | 6.26 | 6.25 |
| Standard deviation | ±0.001 | ±0.001 | ±0.001 | ±0.001 | ±0.001 | ±0.001 |
| DFM-N | 6.35 | 6.31 | 6.28 | 6.26 | 6.25 | 6.24 |
| Standard deviation | ±0.001 | ±0.001 | ±0.001 | ±0.001 | ±0.001 | ±0.002 |
| DFM-O | 6.35 | 6.32 | 6.29 | 6.27 | 6.25 | 6.24 |
| Standard deviation | ±0.001 | ±0.001 | ±0.001 | ±0.001 | ±0.001 | ±0.002 |
| DFM-MMLM | 6.35 | 6.31 | 6.28 | 6.26 | 6.25 | 6.24 |
| Standard deviation | ±0.001 | ±0.001 | ±0.001 | ±0.001 | ±0.001 | ±0.002 |
| DFM-MMLM-OT | 6.35 | 6.31 | 6.28 | 6.26 | 6.25 | 6.24 |
| Standard deviation | ±0.001 | ±0.001 | ±0.001 | ±0.001 | ±0.001 | ±0.002 |

## C.5 TIGHTNESS OF BOUNDS

The expressions of the perplexity bounds are derived by initially dropping the term

$$-\int_0^1 \sum_{x_t} \bar{q}_t(x_t) \sum_{i=1}^L \sum_{x^i \neq x_t^i} \left( \tilde{w}_t^i(x^i, x_t) \log \frac{\tilde{w}_t^i(x^i, x_t)}{\tilde{v}_t^i(x^i, x_t)} + \tilde{v}_t^i(x^i, x_t) - \tilde{w}_t^i(x^i, x_t) \right) dt \quad (140)$$

from the full expression of the KL divergence between the data and the learned distribution in Theorem 4.1. This term can be rewritten as

$$-\int_0^1 \sum_{x_t} \bar{q}_t(x_t) \sum_{i=1}^L \sum_{x^i \neq x_t^i} \left( r_{\bar{q}_t} w_t^i(x_t^i, x) \log \frac{r_{\bar{q}_t} w_t^i(x_t^i, x)}{r_{\bar{p}_t} v_t^i(x_t^i, x)} + r_{\bar{p}_t} v_t^i(x_t^i, x) - r_{\bar{q}_t} w_t^i(x_t^i, x) \right) dt, \quad (141)$$

which shows that it depends on the ratios of induced pathwise probabilities under the model, which are intractable. Unfortunately, this makes this term difficult ot estimate in practice, as computing these ratios would require summing over all possible trajectories that reach a given state at time $t$, which is infeasible due to the uncountably infinite number of such paths.

However, it should be pointed out that when the model learns the flow perfectly, this term becomes zero. Indeed, if $w_t$ matches $v_t$, then the induced probabilities, and therefore the induced ratios match so $r_{\bar{q}_t} = r_{\bar{p}_t}$ implying

$$-\int_0^1 \sum_{x_t} \bar{q}_t(x_t) \sum_{i=1}^L \sum_{x^i \neq x_t^i} \left( r_{\bar{q}_t} w_t^i(x_t^i, x) \log 1 + r_{\bar{p}_t} v_t^i(x_t^i, x) - r_{\bar{p}_t} v_t^i(x_t^i, x) \right) dt = 0. \quad (142)$$

Therefore, we expect this term to decrease as the model improves and more closely approximates the target flow. Even though we cannot estimate the tightness of the bound in real settings, we evaluate it in simplified settings, by conducting the following two analyses.

*Our first analysis* is empirical. The vocabulary consists of three tokens: $M, A, B$ where $M$ is the masked state. The sequence length is two, and the ground truth probabilities over each states are: $P(A, A) = 0.15, P(A, B) = 0.5, P(B, A) = 0.05, P(B, B) = 0.3$.

We assume our model has learned the following imperfect flow:

$p_{1|t}^1(z^1, (M, M); \theta) = [0.9, 0.1]$, (so: $p_{1|t}^1(A, (M, M); \theta) = 0.9$ and $p_{1|t}^1(B, (M, M); \theta) = 0.1$),

$p_{1|t}^2(z^2, (M, M); \theta) = [0.1, 0.9], p_{1|t}^2(z^2, (A, M); \theta) = [0.2, 0.8],$

$p_{1|t}^2(z^2, (B, M); \theta) = [0.3, 0.7], p_{1|t}^1(z^1, (M, A); \theta) = [0.8, 0.2],$

$p_{1|t}^1(z^1, (M, B); \theta) =: [0.5, 0.5],$

and as in the case of DFM-S and DFM-N, once the flow unmasks a token, it always predicts that same token in that position, with a probability of $100\%$. We run a Monte-Carlo simulation to calculate the probability assigned by this flow to each of the four states $(A, A), (A, B), (B, A)$ and $(B, B)$, which returns the following values:

$$\tilde{P}(A, A) = 0.12953, \tilde{P}(A, B) = 0.58568, \tilde{P}(B, A) = 0.02529, \tilde{P}(B, B) = 0.2595$$

Calculating the cross-entropy between the data and the modelled distribution using the ground truth probabilities and the probabilities above, we get $H(P, \tilde{P}) = 1.1626$. Then we use our bound in Equation (14) which in this case becomes the MD4 of Shi et al. The value of the bound is 1.2998 (that is, $H(P, \tilde{P}) \leq 1.2998$), which is about $11\%$ higher then the true value.

The difference between the precise NLL (1.90) and the NLL bound (with value 2.02) from Equation (101) is similar to the differences between the true likelihood and the bound reported in the case of continuous diffusion Song et al. (2021b, Thms. 1 and 3; Table 2).

*The second analysis* is theoretical. As before, the vocabulary consists of three tokens: $M, A, B$ where $M$ is the masked state, and we define a flow that is *independent* of the current state. The sequence length, as previously, is selected to be two. We choose a 'learned' flow such that the probabilities $p_{1|t}^i(x_t^i)$ of jumping to A are $a$ for the first position, and $b$ for the second one. Once a position is unmasked, it never changes just as in DFM-S and DFM-N.

We write the ground truth distribution over states $(A, A), (A, B), (B, A)$ and $(B, B)$ as $p(A, A), p(A, B), p(B, A)$ and $p(B, B)$. The true cross entropy is clearly: $-(p(A, A) \log ab + p(A, B) \log a(1 - b) + p(B, A) \log (1 - a)b + p(B, B) \log (1 - a)(1 - b))$.

Regarding the bound, for $x_1 = (A, A)$, we get $\int_0^1 \frac{1}{1-t} p(A, A)[(1 - t)^2(-\log a - \log b) - (1 - t)t \log a - t(1 - t) \log b] dt = -\int_0^1 p(A, A)[(1 - t)(\log ab) + t \log ab] dt = -p(A, A) \log ab$. Similarly, when calculating the rest, we get $-(p(A, A) \log ab + p(A, B) \log a(1 - b) + p(B, A) \log (1 - a)b + p(B, B) \log (1 - a)(1 - b))$ which is the true cross-entropy, *i.e.*, the bound is tight for this setting. However, this example studies a simple case of a chain whose dynamics are independent of the current state.

### C.6 DYNAMIC AND KANTOROVICH TOTAL COSTS

We generated 3200 samples for each (OT and non-OT), and measured the $L_2$ distance between the changed embeddings at each time steps across all positions, during generation. That is, if some positions change at time $t$ during generation, we add the $L_2$ distance between the embeddings of the changed tokens. We do this for all time points $t$ across all positions, and report the total sum of changes. Based on our first theorem, we expect OT to reduce this quantity, which it does as seen in Table 13.

Table 13: Transport costs for models trained with and without OT

|       | Dynamic | Kantorovich |
|-------|---------|-------------|
| No OT | 6574.68 | 6507.24     |
| OT    | 6328.71 | 6357.15     |

Similarly, for both, the model trained with OT and the one without, we calculate the coupling cost by computing the average of 1200 batches of size 512. This provides the estimated cost of the Kantorovich formulation. Results are shown in Table 13.

## C.7 Correlation Between the Perplexity Bound and Generative Perplexity Results

For each model, we compute two metrics: (1) the average perplexity bound across 5 test sets, and (2) the generative perplexity measured by GPT2-Large with 1024 generation steps. The Pearson correlation coefficient between these metrics is 0.96, indicating very strong correlation. If we normalize the columns of the perplexity results in order to equalize the contribution of each testing set, then the Pearson correlation coefficient changes to 0.958.

## C.8 Ablation Experiments

**Ablation 1: OT Solver.** We replace the Sinkhorn solver with exact OT for multimasked flows. Results appear in Tables 14 and 15. We did not notice differences in training time.

Table 14: Exact OT (MMLM) — 512 OT, 512 dff

| Model | 8 | 16 | 32 | 64 | 128 | 1024 |
|---|---|---|---|---|---|---|
| GPT | 518.86 | 281.39 | 199.68 | 168.12 | 153.51 | 141.46 |
| Standard Deviation | ±2.99 | ±1.57 | ±1.19 | ±0.97 | ±0.95 | ±0.98 |
| LLaMA | 602.34 | 337.54 | 245.29 | 207.90 | 190.66 | 177.82 |
| Standard Deviation | ±3.47 | ±2.03 | ±1.64 | ±1.39 | ±1.35 | ±1.41 |
| Entropy | 6.35 | 6.31 | 6.28 | 6.26 | 6.25 | 6.24 |
| Standard Deviation | ±0.001 | ±0.001 | ±0.001 | ±0.001 | ±0.001 | ±0.002 |

Table 15: Perplexity Results for the Experiment in Table 14

| Set | Lambada | Wikitext2 | PTB | Wikitext103 | LM1B |
|---|---|---|---|---|---|
| Value | 66.27 | 67.60 | 197.05 | 67.62 | 82.98 |

**Ablation 2: OT Batch Size.** We increase the OT batch size from 512 to 4096 while maintaining the diffusion batch size at 512, performing 8 parameter updates per OT batch. We continue using exact OT. Results appear in Tables 16 and 17.

Table 16: Generative Perplexity: Exact OT (MMLM) — 4096 OT, 512 dff

| Model | 8 | 16 | 32 | 64 | 128 | 1024 |
|---|---|---|---|---|---|---|
| GPT | 518.52 | 281.71 | 197.45 | 164.64 | 152.22 | 140.15 |
| Standard Deviation | ±3.03 | ±1.64 | ±1.16 | ±0.96 | ±0.96 | ±0.80 |
| LLaMA | 602.80 | 339.27 | 240.70 | 204.79 | 188.87 | 177.11 |
| Standard Deviation | ±3.67 | ±2.13 | ±1.52 | ±1.35 | ±1.43 | ±1.28 |
| Entropy | 6.34 | 6.30 | 6.27 | 6.25 | 6.25 | 6.23 |
| Standard Deviation | ±0.001 | ±0.001 | ±0.001 | ±0.001 | ±0.001 | ±0.002 |

Table 17: Perplexity Results for the Experiment in Table 16

| Set | Lambada | Wikitext2 | PTB | Wikitext103 | LM1B |
|---|---|---|---|---|---|
| Value | 65.81 | 65.76 | 191.06 | 65.58 | 78.92 |

**Ablation 3: Similarity Metric.** We maintain the 4096 OT batch size and exact solver but replace the $L_2$ metric with the learned metric

$$-\sum_{i=1}^{L} \log p_{1|0}^i(x^i|x_0;\theta)$$

are the backward per token probabilities. Results appear in Tables 18 and 19.

Table 18: Generative Perplexity: Exact OT (MMLM) — $p_{1|0}^\theta$ metric, 4096 OT, 512 dff

| Model | 8 | 16 | 32 | 64 | 128 | 1024 |
|---|---|---|---|---|---|---|
| GPT | 453.59 | 206.86 | 203.69 | 172.14 | 159.52 | 148.73 |
| Standard Deviation | ±2.84 | ±1.62 | ±1.44 | ±1.12 | ±1.01 | ±0.94 |
| LLaMA | 538.47 | 328.03 | 255.03 | 216.38 | 202.21 | 189.07 |
| Standard Deviation | ±3.54 | ±2.32 | ±2.06 | ±1.73 | ±1.69 | ±1.55 |
| Entropy | 6.30 | 6.27 | 6.26 | 6.25 | 6.24 | 6.23 |
| Standard Deviation | ±0.001 | ±0.001 | ±0.002 | ±0.002 | ±0.002 | ±0.002 |

Table 19: Perplexity Results for the Experiment in Table 18

| Set | Lambada | Wikitext2 | PTB | Wikitext103 | LM1B |
|---|---|---|---|---|---|
| Value | 71.98 | 73.36 | 212.62 | 72.80 | 83.54 |

## D INTRODUCTION TO DISCRETE DIFFUSION MODELS

### D.1 DISCRETE-TIME MARKOV CHAINS OVER FINITE-STATE SPACES

A stochastic process $X_1, X_2, \ldots, X_T$, where each state $X_t$ depends solely on the preceding $X_{t-1}$ is called a discrete-time Markov Chain (DTMC). If the states $X_t$ can take any value from the set $\{1, 2, \ldots, S\}$, where $S$ denotes the total number of possible states, and $T$ represents the number of time steps, then we say that this process is a finite-state space DTMC. The probability that at time $t$ we are at $x$ is

$$p_t(X_t = x) = \sum_{y=1}^{S} p(X_t = x, X_{t-1} = y) = \sum_{y=1}^{S} p_{t|t-1}(X_t = x | X_{t-1} = y) p_{t-1}(X_{t-1} = y). \tag{143}$$

If we place all such probabilities $p_t(X_t = x)$ in a vector $s_t$ of shape $S \times 1$, such that $s_t(x) = p_t(X_t = x)$, then from above we can deduce that

$$s_t = P s_{t-1}, \tag{144}$$

where $P(x, y) = p_{t|t-1}(X_t = x | X_{t-1} = y)$. Given an initial probability distribution $s_0$ over states, the equation above fully determines the evolution of the probability over states with respect to time.

### D.2 CONTINUOUS-TIME MARKOV CHAINS OVER FINITE-STATE SPACES (DISCRETE DIFFUSION)

It is possible to define a stochastic process with the Markov property in finite-state spaces, for $t \in [0, T]$, (Doob, 1953). As previously, we can define a discrete-time process, on time points $\{0, \epsilon, \ldots, T - \epsilon, T\}$, such that there is $\epsilon$ probability of activating the previous transition mechanism when progressing from time $t - \epsilon$ to $t$, otherwise we stay where we are with probability $(1 - \epsilon)$. Removing the random variables to simplify notation, we have

$$p_t(x) = (1 - \epsilon) p_{t-\epsilon}(x) + \epsilon \sum_{y=1}^{S} p_{t|t-\epsilon}(x|y) p_{t-\epsilon}(y). \tag{145}$$

We notice that when $\epsilon = 1$ the equation above coincides with Equation (143), and in addition as before we can write Equation (145) in matrix form

$$s_t = (1 - \epsilon) s_{t-\epsilon} + \epsilon P s_{t-\epsilon} = (I + \epsilon(P - I)) s_{t-\epsilon} = (I + \epsilon Q) s_{t-\epsilon} \text{ , where } Q = P - I. \tag{146}$$

From Equation (146), we see that $\frac{s_t - s_{t-\epsilon}}{\epsilon} = Q s_{t-\epsilon}$, which when taking the limit $\epsilon \to 0$ becomes $\frac{ds_t}{dt} = Q s_t$. Given an initial probability distribution $s_0$ over states, the equation above fully determines the evolution (flow) of the probability $p_t$ over states with respect to time. Indeed, the distribution over states at time $t$ is $s_t = e^{tQ} s_0$. This formulation can be generalized, such that $Q$ is allowed to evolve with time,

$$\frac{ds_t}{dt} = Q_t s_t. \tag{147}$$

For the choice $Q_t = \sigma'(t) Q$, where $\sigma$ is monotonically increasing, $\sigma(0) = 0$ and $\lim_{t \to 1} \sigma(t) = T$, we have $s_t = e^{\sigma(t)Q} s_0$. Matrices $Q$ must satisfy the properties of transition-rate matrices (Suhov & Kelbert, 2008), that is, they have non-negative non-diagonal entries, and the elements in each column add to 0. The choice for $Q$ is made such that: $s_1$ is an easy reference distribution to sample from and the matrix exponential $e^{\sigma(t)Q}$ is easy to calculate (Austin et al., 2021; Campbell et al., 2022). Unfortunately, these conditions greatly restrict the design space in this framework.

## E   GENERATED SAMPLES

The following are non-cherrypicked text samples generated from GPT-2–sized models trained under various experimental setups. Outputs may contain hallucinations, inaccuracies, or culturally sensitive content. They are presented solely to illustrate qualitative differences in generation behavior, such as coherence, topical relevance, fluency, and do not reflect the views or endorsements of the authors.

Listing 1: Generated text from DFM-O, with sequence length L=128.

```
 Take a good look at running on ice volleyball ball from the sidelines
    . Do a party crunch once and get bored from another game away.
     Then mess something with a determined and pleasing smoke summon.
     Might not change.

It would have happened if I were busy much less myself.

When your fictional boss feels threatened these dark mysteries are no
    warning to ignore.

Instead ignore what you're working for and watch then acteduate what
    you're doing on view, move around the screen and how your boss
    detected Kung Fu as around you. They are not throwing a police
    officer at your feet. They are just accepting
================================================================
 on Blue Bird" in the Night. Considered a regular occurrence in
    contemporary daytime arts circles, as well as the soundtrack to
    The Breakfast Club (1979), The Heavens Door, Russian-inspired duo
    's nature, Ooboh (and Zoeppo In Peace), and even a German-wave
    song Not To Olmy (1977 album). The highlight of the album's
    Elephant A Ring is the song's Kiss Of Saint John (December 1950),
     in which the island inhabitants embrace a beautiful Viking.

gluk188b - Now the more authentic Azgothic's complex,
================================================================
 folk culture to the masses. In 1900, Dash organized the New
    Draveenjoci Friends Dinah festival, brought together with 50
    local folk groups, including canoeclub, and First Father Township
    .

I spoke with other Dile Dash guardians listening to the show. She's
    the eighth person to stand near church faces. Getting some of the
    staff to volunteer, there were lots of screaming in hopes of
    hearing someone who feels the right to join a church member or
    perform the go-to edition Untitled. To calm their spirits, winners
     brought up the fact that "Polynesicans respect God's
================================================================
 sessions for a down. Even after that, it's all over the board as the
     V&L must-ens. They're also sure to add the other survivors:
     wildcat goal catchers. -Ben Harper, punter reporter (HL)

But it's still challenging to be able to gain a reaction, especially
    with the tack dropping far further down. Instead of matching up
    with position experts, I started to assess how they would perform
    and I started to track nightly games against '80 first-team
    coaches.

The fielders had to look past Dumervil, with Gibbs being
```

Listing 2: Generated text from DFM-N, with sequence length L=128.

```
 the migrant galeslam in Calais.

The Telegraph reported that Lord Dacre's Wembley address included an
    additional briefing on the complaints.

The EU referendum, on which he was asked to vote, said:

'But I am profoundly disappointed with this piece of inquiry that was
    appointed to breach the rules and EU rules and it is unlikely that
     his Labour Government will be affected.

'The Government has lost sight of this blatant interference and has
    attempted to ignore it.' -Ojes

To the Daily Mail, Jimmy once commented: 'Scared to say the verid
===================================================================
s commit investment by defer to the SEC.

The Governor raised serious concerns about stopping the proposed
    measure by arriving to New York City on the day of the pact's July
     31 deadline - if legal - though it would probably do little to
    cut any gains for his state's most successful investors.

Brown administration officials have ruled out complying with short-
    term hedge funds trading rules. That still appears to be only a
    possibility as any OIRP deals seem to crash or soon come into
    force.<|endoftext|>There is "no chance we either profit from the #
    LossLiveup." - the MMQB

===================================================================
 good.

30 Cole Springfield 2016

Springfield alone averaged a superb .667 in his junior season with a
    6-inch pitcher, 6-foot hoop, a 14-curry well and a close
    connection.

As close as any player can have in a baseball academy (the last time
    he had a game) is Gavin Bentley. Less former WSU defensive lineman
    . But Mayau had his playing style over someone else.

31 James Wood, 1925-2002

As well known Oxford export, the Tigers "There Were None" for his
    fellow topronouncement of Juneau, who was the
===================================================================
, and did choke off the second one to show the new media coverage. I'm
     just going to do the rest of our work and ask the city of
    Montreal to discontinue the multi-year tradition of photography.
    That's my last piece. Here in Montreal, we're excited to try to
    work our cities way our working-class citizens.

The The Tonge Room, Le Grand Le Collective is a celebration of various
     global libertarian and anarchist events. Read more on live music
    from our first event. Read more about our team. We spoke with the
    Chanesque Art Project director about the theme we set out for
    Rockavaloon
```

Listing 3: Generated text from DFM-S, with sequence length L=128.

```
 motion of a catcher in which the mechanical properties of the cooling
     fluid's electrical discharge are sure to be overcome of depth"
     says Benfeldt, half year undergraduate in medical tics, in the
     2005 semester, "where we needed to develop a more comprehensive
     model of the precipitation of motion and equality of motion
     general to animal dimensions, there has been a discussion about
     deluge, Form, spin and Motion".[3]

Field motion has played a natural role that mimics parallel movements
     in the laying and loading of a field-dependency container, and
     therefore change the accepted realism of motion. Intuitively, when
     one can demonstrate non
     ================================================================
 the help of Indian FA Dr. Natalia Sekuni to help NYC full backs
     Lilian Balfour and Remis Elijah Mahrez.

Maryab Kardy also had three league games throughout his career with
     Toronto FC.

Korian scored 4.5 goals and 2 assists in 24 Bundesliga appearances
     last season, first for FC Nordsbank Leiburg and has 10.4 goals, 5
     assists in 7 starts this season. He collected a 1-0 assist for
     MacLilleux in 1914-19.

Korian scored one goal against FC Seattle minutes into the game and
     led the Reds to a 21-
     ================================================================
 DeVos delivered various policy and campaign announcements for him.

Cuomo later claimed that he had seen "thousands" of potential voters
     in the state.

Sanders, who spoke in the city in November, charged whether Trump
     would boost the economy, saying, "This is the way I use something
     I know because everybody in America has a great choice and both
     parties today. He had said that the system worked for some but
     that there were a better solutions for the voters."

Trump may present himself as he most likely to have a marquee issue.

Still, Trump did not just hear thunder from his previous candidate
     Trump,

     ================================================================
 in the countryside. Even though the government's actions were however
     far out to be heartening the protesters so deeply, many peasants
     who were intending to give the poor the title instead had asked
     why they should choose to be a representative citizen and
     therefore stand up to defend the peasant family as well as
     society. Immediately, after we saw the working class and even the
     middle-level intellectuals in one segment, they had little doubt
     that the class that raised them was all or part by them of
     resisting the situation, showing why working classes can be an
     irritable about the bourgeois who participated irresponsible
     actions and drove the country up to chaos.
```

Listing 4: Generated text from the DFM-B model with BoW source distribution and normal training, with sequence length L=128.

```
 but it certainly doesn't exclude modules for employees from sand
     Reels resorts," said Ziefen. Those boutique area stores will also
      draw the attention out of local rushers and American area
     brokers.

"This office doesn't see it as a part of proving that a profitable
     stand-up representative, clean, independent business."<|endoftext
     |>Keith Young's secret of the worms' DNA may come from the
     Cparagon Green while studying the region's biodiverse flora.
     Draggio early cartilaginian creatures, from one of their native
     heights to the earth to corn and barley leaves, stirred their
==================================================================
's program was broken down into monopoly vehicle lending. Wells Fargo
     and The New York Times are the largest auto lenders seeking
     infinite certainty on loans. And despite the design principles
     they must comply with the law Wisconsin auto lenders are requiring
      Wells Fargo because no one denies compliance.

Last year, the federal government closed its loan to college and micro
     -urch, said the group of governors. Families across the state also
      have concerns that while the costs of the loans are "viable,
     federal lenders were allowed to prevent borrowers from using
     acceptable financing policies because federal loans, including
     seeking credit default, are denied."

Even if
==================================================================
 and I'm good at learning a few more stuff, I bet that he's the second
      roaster supreme in authority in school that doesn't do any
      arithmetic at all."Christopher Goldstein and Marc Cruz

A friend Jonathan has done quite a little art, and I am sure you've
     heard of English Roles, Almor and Sacnegramald.

It's obvious that we have found guilty of a terrible English plot at
     this time and so are 22-year-olds. It might be instructive to get
     on blowing the sand and doing masters-levels without getting
==================================================================
 I call itself a farmland: "hot habitat garden enticiest that our
     civic/boxing advantage won't have to dismantle overnight;

Mayer that life is simply being ecologically ec

Note: That sets me out to it: stupidly conscious beautiful plants
     versus stupidly conscious living beings. If we leave Human space
     then it will feel much less secure.

And grafts down over solutions down on imperfect.

Which is terrible in my research, & which is why we have a political
     ecologic.

A key inner dilemma in thinking of ecological phenomena is aging.
     Their vitality involves increasing drastically
```

Listing 5: Generated text from the DFM-B-OT model with BoW source distribution and minibatch-OT training, with sequence length L=128.

```
fuel festivals should serve as their short-term goals.

Only one of many Charity & Human Advocates have been written in the
    past to promote free free markets. They can give invitations and
    help repute any who describe the offer and apply their own
    informational refinements. The resistance to or even being that
    the FairMormon group will have to come to educational causes and
    careers and thinkers from not only in Millionaires Web Groups,
    educational and family activities. These groups can also donate by
     mailing lists to kiosks by Comeback and drive by the same
    publisher Samples from that advocacy group by Oxfam. For many
    birthday auctions, groups
    ==================================================================
 Queen City Building in Albine, Romania, the United House said.

Agrini's Airport-blocking congested Charles Avenue area was Baldini's
    first free kick when his 10-yard header gave the Italian side the
    league first of the World Cup, and won them two World Championship
     and trophies.

The Italian native, aged just 20, won his first World Cup title, West
    Club Athletes Player of the Year, and received the Interim Di'Solo
     from the Udinese club's run of P2.5 million, a deal that will be
    considered a move in a Napoli bid.

Speaking
    ==================================================================
 in a way.) Excellent, by e-mails me (good) Shihuan - and its reader -
     for this question, I have already translated this piece into
    fantasy literary thriller. This book is phenomenally interesting
    and amusing and is not really even in writing. The explanation is
     particularly fascinating to add to the fan community and this
    book contains lots of spoilers.

I started writing earnest Vegan Essentials. That book was attached to
    the Rules of My Feast! My first reviews were seven years ago and
    is still a category ninth. This is due to ongoing vegan activism
    and loyalty to other affected readers.

Vegan everyone at the
    ==================================================================
 book entry based on Midnight Symphony. It's an addictive decision,
    and it's going to become what's deciding actions it is going to
    take to mirror what I said to people to Love Your Experience-a
    kind of confluence.

Here's the first theatrical teaser trailer, high-speed footage from
    Rex Arena Theatre with Howard Aller and Marshall Carter at the New
     Sound Day Festival this summer.

But with all the scenes in actual driving mode without going in front
    of a production vehicle, I think there's a huge difference where
    you're essentially in cycling mode; I
```

Listing 6: Generated text from the DFM-MMLM model with multi-mask source distribution and normal training, with sequence length L=128.

```
 laws were part of the way to keep everybody related to one type of
    plants in their social roles."

Mr. Zhou shushed, saying, "My vocals won't show up for weeks, but we'
    ll be showing civil expressions. We wanted to call for community
    involvement."

At 26, Noonan was really pushing for contentious statements.

To prove the point, Tonelli participated in a group meeting in
    Hannamkel, Georgia (you won't find room between sour'' and
    screwhead.k spoke toward all of these dairy farmers). The four
    always told each

====================================================================
 is through lots of reporting and reerforming ways on the realised
    check.

In essence, a simple move gives the developer a rescall of mutable and
    push limits for wethers which is typical in code. To start and
    alleviate checksums and other exceptions. I find this technique is
    especially actually interesting, when times are changing and a
    design change for push limits a degree away from mutable and
    wethers:

Implementation

The code explicitly gives the right to namespace crash if the dynamic
    codebase's changing anything, and nothing that does change
    triggers entryulating. On the other hand, providing

====================================================================
 somewhat), and police encryption systems discussed in detail don't
    with this approach have very high level security.

The most significant hole would be close between the fake Syed GED
    listing and the bogus public AR-15 in restriction that they were
    unaware of NSA activity in the past. Instead, they enlisted
    isardars like Shin Intel's George Singleton TP Program to gain
    access to a small subset of unknowns Syed before April 2002.

Borse and spoofing is never wise enough, however. There was a whole
    site beside that old swatch and telly when it first was instigated
    by the feds. While a

====================================================================

 behaviour changes, resulting in a some degree of glacial DNA
    diversity in the signaling system. The research results suggest
    similar variations in the system evolved, at least since
    2004.[48]

and the initial development of military radio networks began. In
    January 2008, to fund his experiments, the US founded Jo Kutus, a
    micro-channel engineer and orthopedic surgeon near Ulisz, south-
    west to decodel dynamic television imagery and dropped it rain.
    The total cost would be US$135 million for the I-3 plan and 10%
    before TV broke.

Except an ideal early model, most countries
```

Listing 7: Generated text from the DFM-MMLM-OT model with multi-mask source distribution and minibatch-OT training, with sequence length L=128.

```
 " came out earlier this year.

Absent Films begins production in partnership with the Entertainment
    Agency Europe (MEGO), the Italian news agency Gazeta and Spain's
    Forza National Investigation Agency (ASIO. It is said to be
    producing about 21 films worldwide, titled "Nobody Loves Worries."

Absency Films' CEO, Michael Agiloh, offers an explanation of why many
    of his characters appear on screen -- "In these many shows, each
    of us are in the center of our heads, filled with energy to do
    that go outside our cells to stimulate, stimulate, and recreate
    love

============================================================
 Dudley on the brink of a Joakier contract, the Knicks could be more
    optimistic this summer, not on any physical trade for Thomas.

[An MLB free trade period. Here's what we have here.]

They are also no longer in the trade market for center Raymond Felton,
    which was traded to Andrea Bargnani and was busy signing out a
    2021 deal. Ono, who has been a productive player on the roster,
    would get significant financial relief with a new deal. Thomass
    agent signing would mean he leaves an expiring contract after the
    NBA season in 2017.

For longer, they

============================================================
 random two Anthrax databases, when marked cases are cleaned up, and
    one still has access to records from within the center of a case.

Researchers say they have concocted an elaborate system of polygraphs
    that process documentions, original polygraphs, which can then be
    used to sort and review on file case in a bid to preserve the
    files.

Public Citizen, which organized the document,, learned of the
    recording of phone conversations between President George H. W.
    Bush of Odessa, Conn., during a February 2005 trip to New York
    City.

The FBI is using its investigative techniques to shut down a

============================================================
Repeat this after under Run as menu and under Advanced. Now we've
    obtained the empty Zone_X file so application requires synchronous
     and Authentication to search for those within Zone. After doing
    so, the field name for the Zone_X and the abbreviation the Zone_X
    field are activated.

The first and second column or column change Zone's current property.
    Bring it back from the new view to complete the Forms.

Above will show some settings generated in the previous view that was
    enabled by bot-originator . Here they appear in a Parameters
    screen :

Siren Bot Name Screen Off-screen Name Dimensions
```