# OpenReview forum: "Minibatch Optimal Transport and Perplexity Bound Estimation in Discrete Flow Matching"
_ICLR.cc/2026/Conference — Submitted to ICLR 2026_

### Official Review · Reviewer_Yqsk · 2025-10-26

**Soundness:** 3
**Presentation:** 3
**Contribution:** 2
**Rating:** 6
**Confidence:** 4

**Summary:**

This paper proposes a minibatch OT for discrete flow matching (DFM), which corresponds to OT coupling strategy in continuous FM. They adapt the Kantorovich transport with quadratic cost in continuous setting into DFM and define the categorical dynamic objective. They further derive two practical upper bounds on perplexity for DFM, enabling principled raining, evaluation and model comparison. Finally, they implement the discrete minibatch OT to multitask flow. By experiments on several text datasets, they show that the OT coupling outperforms regular masked DFM.

**Strengths:**

1. The paper is well-written and easy to follow.
2. The OT in discrete setting is well motivated and the proposed OT objective is clear and intuitive.
3. The derived perplexity bounds are useful for model comparisons.

**Weaknesses:**

1. Transition from continuous to discrete case may raise some novelty issue (don't take points off on this)
2. Minibatch-OT depends on token embeddings (for L2) that co-evolve during training; stability/selection (e.g., EMA embeddings) is only briefly discussed.
3. Bounds’ tightness is shown in narrower settings; calibration vs exact NLL remains partially open at scale.

**Questions:**

In continuous setting, although OT-FM is promising in theory, it is not very widely used in practice. This may come from the bias of minibatch effect (approximating continuous OT via samples). In the discrete setting, I guess this issue will even be more severe. How sensitive of OT in DFM according to batch size?

---

> ### Author Response · Authors · 2025-11-19
>
> We thank the reviewer for their time and careful evaluation. We are pleased that they found the paper well written and easy to follow, and that they appreciated our two main contributions: (i) the clear motivation of OT in the discrete setting and the intuitive formulation of the proposed OT objective, and (ii) the derived perplexity bounds, which are useful for rigorous model comparisons.
>
> **(W1) Transition from continuous to discrete case may raise some novelty issue (don't take points off on this)**
>
> We thank the reviewer for the comment as it allows us to explain in more detail why our contribution is actually novel, and why we believe not taking points off is the correct decision. The differences of our formulation and theorem with those in the classical discretized OT are detailed below:
>
> 1. *Restricted but structured dynamics.*
>    We deliberately work in a restricted setting: first, we consider *per-position* discrete flows on sequence space (this is the discrete flow matching framework of Gat et al. 2024 [1]), and then a particular subclass of those flows given by convex interpolants. Precisely because the dynamics are so structured (specific), it is reasonable to expect stronger results to hold, and in particular to hold for a broader class of cost functions than in the classical Benamou–Brenier setting.
>
> 2. *General similarity and path-length, not kinetic energy.*
>    Within this restricted dynamic class, we introduce a general similarity function $s(\cdot,\cdot)$ as a weighting term (which already makes our formulation different) and define a dynamic objective based on (expected) path length rather than kinetic energy (yet another essential difference). That is, we work with discrete jump rates and weighted jump counts instead of a quadratic velocity field, which naturally accommodates more general notions of discrepancy than $\|\cdot\|^2$.
>
> 3. *“Stronger” Benamou–Brenier-type result: general $s$ and additive cost.*
>    Our hypothesis that this restriction of the dynamics allows a result with more flexible metrics is confirmed: our Benamou–Brenier-type theorem holds for *arbitrary* choices of $s$, not just squared Euclidean distances. In particular, Kantorovich cost is
> $c(x_0,x_1) = \sum_{i=1}^L s(x_0^i,x_1^i),$
>    so the static objective is simply the sum of per-position similarities. The fact that the dynamic path-length objective collapses exactly to this additive Kantorovich cost for general $s$ is, we believe, a particularly elegant outcome.
>
> 4. *Fixed dynamics, optimization over couplings, and equality without extra infima.*
>     In the case of discrete flows with convex interpolants, since the flow dynamics are fully determined once a coupling $\(\pi\)$ is chosen, the optimization in our framework is taken only with respect to the coupling (not the whole flow itself). For every $\(\pi\)$, the associated convex-interpolant flow induces a *dynamic* loss and a *static* Kantorovich loss that are exactly equal for that same $\(\pi\)$; taking the infimum over couplings on both sides is then purely formal and not necessary to establish equality. In other words, the coupling automatically determines matching dynamic and Kantorovich losses, and no additional infimum over flows is required. This pointwise equality for each coupling makes our result structurally stronger than the classical formulation, where one typically only proves equality after taking infima over both dynamic paths on one side and couplings on the other.

---

> > ### Author Response · Authors · 2025-11-19
> >
> > **Minibatch-OT depends on token embeddings (for L2) that co-evolve during training; stability/selection (e.g., EMA embeddings) is only briefly discussed.**
> >
> > We thank the reviewer for allowing us to clarify this point. We raised this point as a hypothetical limitation and proposed it as a direction for future work. We do not claim, nor do we have proof, that allowing token parameters to co-evolve necessarily harms model performance. Rather, we believe this would make an interesting standalone study, where one could systematically investigate such effects by comparing fixed embeddings, pretrained embeddings, EMA embeddings, and related approaches. Ideally, this topic would be explored in a dedicated paper, as doing it justice would require substantial additional analysis. Given that the current submission already includes several theoretical contributions alongside extensive experiments and ablations, we chose not to treat this issue only partially here. We thank again the reviewer for their careful reading of the paper and hope this clarifies our intent.
> >
> >
> > **Bounds’ tightness is shown in narrower settings; calibration vs exact NLL remains partially open at scale.**
> >
> > We agree that obtaining sharper characterizations of the tightness of our bounds at larger scales would be highly desirable. However, to the best of our knowledge this has remained an open problem since SEDD of Lou et al. [3], even in the simpler cases of masked and uniform diffusion. We are not aware of prior work that systematically evaluates the tightness of bounds in discrete diffusion/flow models, so we view our analysis in more controlled settings as a valuable first step that may inspire further research in this direction.
> >
> > It is particularly encouraging that, in our small-scale experiments, the discrepancy between the true and estimated NLL closely mirrors the behavior reported by Song et al. [2] for high-dimensional continuous diffusion. Moreover, we provide an explicit perplexity formula in the paper, in which one of the terms is currently computationally intractable; we hope that making this structure explicit will motivate future work on more efficient or tighter estimators. In this sense, we believe our paper pushes the frontier of bound estimation in this area.
> >
> > **In continuous setting, although OT-FM is promising in theory, it is not very widely used in practice. This may come from the bias of minibatch effect (approximating continuous OT via samples). In the discrete setting, I guess this issue will even be more severe. How sensitive of OT in DFM according to batch size?**
> >
> > We agree with the reviewer that [4] is promising in theory and an active area of research. We belive that a main factor that limits it somewhat in practical, industrial adoption is the competition from consistency models, which are not applicable in the discrete case, a fact that motivated our approach.
> >
> >
> > We also examined how performance scales with batch size. Since the OT batch size must be at least as large as the diffusion batch size, we focused on scaling up and then measuring the effect of reducing the batch from 4096 to 512. We found that this reduction does not significantly degrade the generative perplexity. This suggests that even with relatively small batches, one can still obtain substantial benefits from incorporating OT. Please see Appendix C8, Table 14, 15, 16, 17 for detailed results.
> >
> > We thank again the reviewer for their time and careful evaluation. We hope to have addressed the stated concerns and are happy to discuss anything that has remained unclear.
> >
> > [1] Gat et al. Discrete Flow Matching. 2024
> >
> > [2] Song et al. Maximum Likelihood Training of Score-Based Diffusion Models. 2021
> >
> > [3] Lou et al. Discrete Diffusion Modeling by Estimating the Ratios of the Data Distribution. 2024
> >
> > [4] Tong et al. Improving and generalizing flow-based generative models with minibatch optimal transport. 2023

---

### Official Review · Reviewer_i4qU · 2025-10-28

**Soundness:** 2
**Presentation:** 2
**Contribution:** 3
**Rating:** 4
**Confidence:** 3

**Summary:**

The paper explores shortest distance transitions in Discrete Flow Matching (DFM) for categorical data and proposes probability estimation by perplexity upper bounds. It proposes a weighted path-length dynamic OT objective whose Kantorovich form depends only on token-level similarity. Under convex interpolants this yields a categorical Benamou–Brenier result. Practically, this enables minibatch OT training with either Hamming or embedding-L2 costs. The paper further derives two computable upper bounds on perplexity for DFM, usable as training losses and evaluation metrics. Empirically, minibatch-OT reduces steps by up to 8× at matched generative perplexity, and the authors introduce Multimask Flow which outperforms masked flows, especially with OT, while adding ~3.4% training overhead in their setup.

**Strengths:**

- The categorical Benamou–Brenier-style equivalence (Theorem 3.1) is interesting. The dynamic jump-minimization equals a Kantorovich problem with cost $c(x_0,x_1)=\sum_i s(x_0^i,x_1^i)$, recovering Hamming distance and L2-embedding costs for different choices of $s$.
- On OpenWebText with GPT-2–sized models, minibatch-OT cuts steps by ~8× to match the non-OT model’s generative perplexity. Training overhead is reported at ~3.4% for L=128, with favorable scaling to longer sequences.

**Weaknesses:**

- The paper uses generative perplexity judged with external LMs. This could cause judge bias with the choice of external LMs.
- Reported headline gains are emphasized on the judged metric rather than the bounds themselves.
- Most results are GPT-2-scale on BoW settings. It’s unclear how benefits translate to larger LMs or real web texts. The method likely generalizes, but evidence is limited.

**Questions:**

1. Could you report correlation between bound values and the generative perplexity across settings, and clarify which bound you recommend for model selection?
2. Can authors detail hyperparameters choice or provide ablations?
3. Can authors explore results at >1B parameters? The overhead scaling looks promising. Can you document wall-clock times at larger scales?
4. How do Hamming vs L2-embedding vs learned metrics compare for step reduction and final quality?

---

> ### Author Response · Authors · 2025-11-19
>
> We thank the reviewer for their time and careful evaluation, as reflected in the concise and accurate summary of our paper. We are pleased that they found the categorical Benamou–Brenier-style equivalence interesting and appreciated that we can recover important cost metrics from intuitive similarity measures. We are also grateful that the reviewer valued the reduction in generation steps and the favorable scaling with larger batch sizes. Finally, we are encouraged that the reviewer considers the paper presents valuable contributions, as indicated by the good contribution score. We address the concerns below.
>
> **(W1) The paper uses generative perplexity judged with external LMs. This could cause judge bias with the choice of external LMs.**
>
> The reviewer is right to be concerned about potential bias from the external LLM evaluator. This was also a key concern for us, which is why we
>
> (i) used two different LLMs to estimate generative perplexity, and
>
> (ii) computed an entropy score for every model to quantify token-level diversity and guard against artifacts or biased sampling (e.g., issues arising from incorrect sampling as discovered in [1]).
>
> In addition, for every experiment we reported the corresponding perplexity bound values, which do not depend on any external evaluator and instead measure the model’s intrinsic ability to approximate the data distribution.
>
> **(W2) Reported headline gains are emphasized on the judged metric rather than the bounds themselves.**
>
> We thank the reviewer again for their careful reading. In the main paper, we focus on generative perplexity, as our original motivation for using optimal transport was to reduce the number of generation steps. However, we did compute the perplexity bound for every experiment and reported these results in the Appendix (C2, C3, and now the added C8). As we explain in more detail below, the bounds and generative perplexity are in close agreement, with a Pearson correlation of 0.96, indicating a very strong correlation.
>
> **(W3) Most results are GPT-2-scale on BoW settings. It’s unclear how benefits translate to larger LMs or real web texts. The method likely generalizes, but evidence is limited.**
>
> We agree that it would be very valuable to validate our approach on larger models and more realistic web-scale datasets. However, given the already substantial number of experiments in this work and the significant computational cost of scaling them to industrial-size models, we chose to focus on scales that are standard in the discrete diffusion/flow literature ([2,3,4]), on which our paper builds. While these are not production-scale models with 1B+ parameters trained on hundreds of billions or trillions of tokens, the setting we use is nonetheless nontrivial: our models have on the order of 200M parameters, and the dataset comprises approximately 8 billion tokens.
>
> We share the reviewer’s assessment that all indications in the paper suggest the method should generalize to larger models and more realistic text settings, and we strongly agree that such large-scale experiments are an important next step. In line with common practice, where initial methodological or theoretical work is followed by dedicated scaling studies (e.g., [5,6]), we believe this would be best addressed in a separate follow-up paper focused specifically on empirical scaling properties.
>
> In this submission, we view our theoretical results as the main contributions, with several experiments serving to validate and illustrate the theory. Adding large-scale experiments on top of the current theoretical and empirical content would risk overextending the paper and harming its coherence. We thank the reviewer for emphasizing the importance of this direction.
>
>
>
> **(Q1) Could you report correlation between bound values and the generative perplexity across settings, and clarify which bound you recommend for model selection?**
>
> Thank you very much for the suggestion; it led to a valuable addition to our paper (Appendix C7). For each model in Table 3, we compute two metrics:
>
> (1) the average perplexity bound across 5 test sets, and
>
> (2) the generative perplexity measured by GPT2-Large with 1024 generation steps.
>
> The Pearson correlation coefficient between these metrics is 0.96, indicating very strong correlation. If we normalize the columns of the perplexity results in order to equalize the contribution of each testing set, then the Pearson correlation coefficient changes to 0.958.
>
> Regarding the choice of bound, the second bound is applicable only when the source distribution is either masked or has i.i.d. dimensions. In such cases, we recommend computing both bounds and selecting the smaller one. Since they are both upper bounds on perplexity, the smaller value will be closer to the true perplexity.

---

> > ### Author Response · Authors · 2025-11-19
> >
> > **(Q2) Can authors detail hyperparameters choice or provide ablations?**
> >
> > We use the setting of Lou et al [2], and the parameters stated therein, which we list in Appendix C. We did not perform ablation studies on these diffusion related parameters and instead relied on the work of previous authors. Motivated by the reviewer’s suggestion, we focused our ablation studies on the OT-related hyperparameters. We conduct ablation studies on three key factors: OT batch size, OT similarity metric, and OT solver. Switching from Sinkhorn to exact solver in multimask flow yields minor improvements for small step sizes (8 and 16), with more pronounced gains in bound values. Increasing minibatch OT size to 4096 (while maintaining diffusion batch size) slightly improves results at 32 and 64 steps, again with more pronounced improvements in bound values. Replacing the $L_2$ metric with a heuristically chosen learned metric $-\sum_{i=1}^L \log{p^i_{1|0}(x^i|x_0;\theta)}$ greatly improves generative perplexity at small step sizes but underperforms overall. More details and the results can be found in Appendix C8. We thank the reviewer for this suggestion, as we believe it has further enriched our paper.
> >
> >
> > **(Q3) Can authors explore results at >1B parameters? The overhead scaling looks promising. Can you document wall-clock times at larger scales?**
> >
> > Regarding experiments with billion-parameter models, as we explained in our response to (W3), we are unfortunately not able to run such studies. We agree with the reviewer that the scaling trends we observe are promising. However, on our hardware (GH200), we already saturate memory at larger batch sizes, so exploring scaling at substantially larger regimes would require proper multi-GPU parallelization.
> >
> > We would like to emphasize that the batch sizes for which we report wall-clock times are in line with those commonly used in production. For example, batch sizes for GPT-3 are reported to range roughly between 245 and 1560, for PaLM 540B between 512 and 2048, and for LLaMA 3 405B around 1000. Thus, we do not expect the overhead we report to grow significantly for modern large-scale models beyond what we already observe. In fact, we believe it could be further reduced with production-level engineering and optimization.
> >
> > **(Q4) How do Hamming vs L2-embedding vs learned metrics compare for step reduction and final quality?**
> >
> > We advise against using Hamming distance for large vocabulary sizes, even though its performance is quite strong in the small-vocabulary regime reported in the paper. We were positively surprised by the effectiveness of the $L_2$-embedding strategy, and in the ablation section (Appendix C8) we also evaluate a learned metric
> > $-\sum_{i=1}^L \log{p^i_{1|0}(x^i|x_0;\theta)},$
> > where $p^{i}_{1|0}(x^i \mid x_0; \theta)$ denotes the one-step transition probability of token $x^i$ given $x_0$ under the model parameters $\theta$.
> >
> > We chose this metric heuristically, as we wanted to pair source points to the datapoint that the one-step flow considers most likely to generate. Unfortunately, as shown in Appendix C8, this learned metric underperforms the $L_2$ metric except in terms of generative perplexity at small numbers of steps (8 or 16), where the gains are nevertheless quite significant. The space of possible metrics is large, and that the best choice will in practice depend on the context, task, and objective.
> >
> >
> >
> > We thank again the reviewer for their time and careful evaluation. We hope to have addressed the stated concerns and are happy to discuss anything that has remained unclear.
> >
> >
> > [1] Zheng, Kaiwen, et al. "Masked diffusion models are secretly time-agnostic masked models and exploit inaccurate categorical sampling." 2024
> >
> > [2] Lou et al. Discrete Diffusion Modeling by Estimating the Ratios of the Data Distribution. 2024
> >
> > [3] Haxholli, Etrit, et al. "Efficient perplexity bound and ratio matching in discrete diffusion language models." 2025
> >
> > [4] Subham Sekhar Sahoo et al. “Simple and Effective Masked Diffusion Language Models”. 2024
> >
> > [5] Ye et al. Dream 7B: Diffusion Large Language Models. 2025
> >
> > [6] Nie et al. Scaling up Masked Diffusion Models on Text. 2025

---

> > > ### Comment · Reviewer_i4qU · 2025-11-27
> > >
> > > Thank you for your response. I'll keep my current rating decision.

---

> > > > ### Author Response · Authors · 2025-11-27
> > > >
> > > > We thank the reviewer for their response. From the unchanged rating, we understand that the main outstanding concern relates to the lack of experiments using large models trained on extensive datasets. If other issues remain unaddressed, we would welcome clarification, as such feedback helps us strengthen the paper. With respect to this point, we realize we did not emphasize in our rebuttal that Section 5.3 experiments utilize OpenWebText, which is a large-scale dataset, and that our model processes this dataset multiple times during training, totaling 26 billion tokens, which is comparable to recent language modeling research. Thank you again for your time.

---

> > > > > ### Author Response · Authors · 2025-12-02
> > > > >
> > > > > We would like to further expand our response to **(Q3)** to address how the overhead changes relative to flow training as the transformer itself scales: OT computation scales linearly in embedding dimension D, while transformer computation scales quadratically in D. Consequently, as the MLPs and embedding dimensions in the transformer grow, the relative overhead of our proposed approach decreases compared to the flow optimization cost. Furthermore, our method is agnostic to the number of transformer layers, which means the relative overhead is expected to decrease even further as models scale.

---

### Official Review · Reviewer_heAo · 2025-10-31

**Soundness:** 2
**Presentation:** 1
**Contribution:** 1
**Rating:** 2
**Confidence:** 3

**Summary:**

The paper is arguably built on two parts. The first one is centred around a discussion of optimal transport for discrete data, where the authors show that the dynamic formulation of the transport problem is the same as the Kantorovich formulation. In the second part, the authors show multiple upper bounds on the perplexity of a discrete flow matching model.

**Strengths:**

- The provided proofs seem mostly fine.
- The connections between the Hamming distance and the metric are interesting.
- The connection between the discrete dynamic problem and the Kantorovich formulation seems novel.
- The authors introduce Multimask Flow, where the vocabulary now also contains multiple mask tokens.

**Weaknesses:**

- Poor typesetting overall. Examples include: line 127 (should not be numbered, nor in align/gather environments), line 159 – 161 (unindented), all tables could benefit from using booktabs, proofs in appendix have lines ending with equality sign – which should really be at the beginning.
- It seems that the contribution in section 3 is poor. I must admit that I am not even certain of what is exactly proved. It does seem that it introduces the dynamic formulation for the discrete optimal transport in this exact form, but similar formulations exist (with flows on graphs, for instance), and, overall, it is a pretty well-studied area [1, 2, 3]. In general, discrete optimal transport is well-known, and standard libraries such as POT [4] includes the Hamming distance in the library. Other papers also use discrete OT [5]. Overall, I am not certain at all about the novelty in this part whatsoever, but it could be that the presentation, which I found lacking, hinders what the contributions really are.
- Discrete OT has only been motivated by the fact that it works well in continuous settings. I do not see why we should expect a similar trend in the discrete setting. Moreover, because of the poor scaling, OT is not used in the continuous settings.
- The alleged improvement enabled by discrete OT is measured terribly in the first experiment: only the number of “jumps” on an obscure dataset which I could not find (citation needed!) is referred to, without controlling for any quality metric. The constant flow of zero has zero jumps, so is arguably better by this sole metric.
- What are the relative jumps? I believe the metric is not introduced anywhere.
- Perplexity makes only sense in the autoregressive, so I shall assume that the authors meant the NLL.
- As for section 4, upper bounds on the NLL exist in MDLM [6], for instance. Discrete flow matching, MDLM and other methods alike are essentially equivalent, and are all trained with the cross-entropy loss; it is quite easy to compute a bound on the test NLL, as the loss itself typically constitutes an (N)ELBO. I am not certain about the novelty; if the tightness is improved upon, it should have been somewhat motivated theoretically.
- I am not certain how the tightness is proved in the experiments whatsoever.
- In experiment 5.2, it is claimed that SEDD has an NLL of about 80, whereas in the original paper the authors get it down to about 33.
- Multimask Flows are never properly introduced, only very briefly in section 5.3. The empirical evidence is weak for “outperforming masked flows”, and I believe that the generative perplexities for these models are much lower in the other works.

**Questions:**

- What is the novelty in your discrete OT section?
- Why is OT for discrete settings important? As you have seen in your own experiments, it scales poorly for vocabularies larger than… 3. How do you expect to scale it to 50k tokens?
- You claim that similarity in the embeddings should be used to weigh the cost differently. What embeddings are you referring to? that of your untrained model? an auxiliary, pre-trained model?
- Do we agree that you meant NLL and not perplexity?
- For 5.1, do you have quality metrics interacting with the number of jumps?
- Why Multimask Flows allegedly work better? Could it just be that they have more parameters? Do they use (that is to say, generate) the special tokens introduced? If so, how? Or did I misunderstand the proposed method altogether?
- What is novel in your new NLL bound?
- When do you use the bound in the experiments? At all times? for all methods? If you did for SEDD as well, for instance, then your bound is definitely not tighter.
- Could you please expand on the claims about the tightness of your bounds?
- How do you estimate the minibatch OT? POT?

Overall, feels like 3 different ideas put together but without the depth that they might all necessitate. The presentations certainly hinders the understanding of the results.

### References

[1] Gabriel Peyré, Marco Cuturi. “Computational OT”.

[2] Ravindra K. Ahuja, Thomas L. Magnanti, James B. Orlin. “Network Flows”.

[3] Robert M. Gray. “Transport distance, Shannon information and source coding”. https://ee.stanford.edu/~gray/gretsi.pdf

[4] Flamary R., Vincent-Cuaz C., Courty N., Gramfort A., Kachaiev O., Quang Tran H., David L., Bonet C., Cassereau N., Gnassounou T., Tanguy E., Delon J., Collas A., Mazelet S., Chapel L., Kerdoncuff T., Yu X., Feickert M., Krzakala P., Liu T., Fernandes Montesuma E. POT Python Optimal Transport (version 0.9.5). URL: https://github.com/PythonOT/POT

[5] Xiaoyang Hou & Tian Zhu, Milong Ren, Dongbo Bu, Chunming Zhang, Xin Gao, Shiwei Sun. “GGFlow: A Graph Flow Matching Method with Efficient Optimal Transport”.

[6] Subham Sekhar Sahoo, Marianne Arriola, Yair Schiff, Aaron Gokaslan, Edgar Marroquin, Justin T Chiu, Alexander Rush, Volodymyr Kuleshov. “Simple and Effective Masked Diffusion Language Models”.

---

> ### Author Response · Authors · 2025-11-19
>
> We thank the reviewer for their time. Before addressing the technical points, we would like to briefly comment on the tone of the review. Phrases such as “OT is measured terribly” and the overall tone of other comments are not aligned with the usual standards of ICLR and could be phrased more constructively. In several places, strong, confident claims are made that are factually incorrect or at best based on misunderstandings of the paper, and are contradicted by the content of the submission (some of which is emphasized multiple times throughout) and, in places, by statements elsewhere in the same review. Furthermore, we are concerned as the review indicates lack of familiarity with standard language processing terminology such as the proper definition of perplexity, standard datasets such as Shakespeare and conflates the general discrete flow framework with the specific masked diffusion dynamics. We also note that other reviewers (Nso4, Yqsk) found the presentation clear and awarded high presentation scores. Given the substantial theoretical nature of the work, we invested significant effort in making the paper accessible, and we are happy to clarify any remaining ambiguities.
>
>
> **(W1) It seems that the contribution in section 3 is poor. I must admit that I am not even certain of what is exactly proved. It does seem that it introduces the dynamic formulation for the discrete optimal transport in this exact form, but similar formulations exist (with flows on graphs, for instance), and, overall, it is a pretty well-studied area [1, 2, 3]. In general, discrete optimal transport is well-known, and standard libraries such as POT [4] includes the Hamming distance in the library. Other papers also use discrete OT [5]. Overall, I am not certain at all about the novelty in this part whatsoever, but it could be that the presentation, which I found lacking, hinders what the contributions really are.**
>
> First, we point out that this comment contradicts the one given in the *Strengths* Section: "The connection between the discrete dynamic problem and the Kantorovich formulation seems novel."
>
> After checking the references provided by the reviewer, we could not find anything close to our work. We would be grateful if the reviewer could point to the precise results/definitions in the provided references, which the reviewer claims to exist and cover our results. It is extremely unlikely for such material to have covered our results, as discrete flow matching was introduced in 2024, while the most recent OT reference is from 2019. We explain the differences with the classical discretized OT in detail below.
>
> 1. *Restricted but structured dynamics.*
>    We deliberately work in a restricted setting: first, we consider *per-position* discrete flows on sequence space (this is, the discrete flow matching framework [11]), and then a particular subclass of those flows given by convex interpolants. Precisely because the dynamics are so structured (specific), it is reasonable to expect stronger results to hold, and in particular to hold for a broader class of cost functions than in the classical Benamou–Brenier setting.
>
> 2. *General similarity and path-length, not kinetic energy.*
>    Within this restricted dynamic class, we introduce a general similarity function $s(\cdot,\cdot)$ as a weighting term (which already makes our formulation different) and define a dynamic objective based on (expected) path length rather than kinetic energy (yet another essential difference). That is, we work with discrete jump rates and weighted jump counts instead of a quadratic velocity field, which naturally accommodates more general notions of discrepancy than $\|\cdot\|^2$.
>
> 3. *“Stronger” Benamou–Brenier-type result: general \(s\) and additive cost.*
>    Our hypothesis that the restriction of the dynamics allows a result with more flexible metrics is confirmed: our Benamou–Brenier-type theorem holds for *arbitrary* choices of $s$, not just squared Euclidean distances. In particular, the induced Kantorovich cost is
> $c(x_0,x_1) = \sum_{i=1}^L s(x_0^i,x_1^i),$
>    so the static objective is simply the sum of per-position similarities. The fact that the dynamic path-length objective collapses exactly to this additive Kantorovich cost for general $s$ is, we believe, a particularly elegant outcome.

---

> > ### Author Response · Authors · 2025-11-19
> >
> > 4. *Fixed dynamics, optimization over couplings, and equality without extra infima.* In the case of discrete flows with convex interpolants, since the flow dynamics are fully determined once a coupling $\pi$ is chosen, the optimization in our framework is taken only with respect to the coupling (not the whole flow itself). For every $\pi$, the associated convex-interpolant flow induces a *dynamic* loss and a *static* Kantorovich loss that are exactly equal for that same $\pi$; taking the infimum over couplings on both sides is then purely formal and not necessary to establish equality. In other words, the coupling automatically determines matching dynamic and Kantorovich losses, and no additional infimum over flows is required. This pointwise equality for each coupling makes our result structurally stronger than the classical formulation, where one typically only proves equality after taking infima over both dynamic paths on one side and couplings on the other.
> >
> > Finally, [5] just gives the classical definition of the Kantorovich formulation, and heuristically sets the cost to be the Hamming distance, therefore there is no formulation of dynamic OT or any Benaou-Brenier-like connection between the Kantorovich and the dynamic formulation. Moreover, they apply their method to molecular generation, wherein the size of the vocabulary is very small: 4 and 9 nodes for the first and the second dataset, and 3 edges in both cases. This is likely why the Hamming distance is useful in their setting. We apply our method to language tasks with vocabulary sizes of 50 thousand to 100 thousand.
> >
> > We have added these explanations to the paper.
> >
> > **(W2) Discrete OT has only been motivated by the fact that it works well in continuous settings. I do not see why we should expect a similar trend in the discrete setting. Moreover, because of the poor scaling, OT is not used in the continuous settings.**
> >
> > We do not just rely on an expectation that discrete OT will behave similarly to continuous OT. Instead, we:
> >
> > 1. Motivate discrete OT via its dynamic formulation in the discrete-flow setting, and derive its Kantorovich formulation,
> >
> > 2. Test it empirically,
> >
> > 3. And observe that, in discrete settings, we obtain a similar beneficial trend: significant improvements for only a moderate increase in computation.
> >
> > Regarding “poor scaling” of OT in continuous settings: the peer-reviewed work of Tong et al. [7] has had substantial impact (over 500 citations in less than 1.5 years), and is regarded to be promising.
> >
> > **(W3) The alleged improvement enabled by discrete OT is measured terribly in the first experiment: only the number of “jumps” on an obscure dataset which I could not find (citation needed!) is referred to, without controlling for any quality metric. The constant flow of zero has zero jumps, so is arguably better by this sole metric.**
> >
> > We focus on the jump count in the first experiment because we expect the quality of the flow to be at least as good: we are training a flow with identical marginals at the source and target distributions, so quality is unlikely to be traded off. Second, and more importantly, the goal of the section is purely to show how OT reduces the number of jumps in a learned flow, not to beat the state of the art on Shakespeare. We never allege that the quality of the flow will improve, simply that the number of steps will be reduced to show the effect of OT. We leave the quality improvements for the experiments in realistic settings in Section 5.3. Nevertheless, we agree that complementing this with a quality metric causes no harm, and we have now measured and reported the perplexity bound as well (see updated results in the revised paper, Tables 1 and 5). The perplexity bound results are significantly improved when OT is used.
> >
> > The main purpose of this experiment is as a proof of concept on a toy dataset. The dataset as stated in the paper is the standard Shakespeare dataset, which is one of the most famous toy data sets in language modeling. In one of the experiments, we apply a Morse-code representation, which can be seen simply as a different tokenization of the same underlying text.
> >
> > **(W4) What are the relative jumps? I believe the metric is not introduced anywhere.**
> >
> > In general, by relative, we mean that if that method A gives result x, method B gives result y and method C gives result z, then the relative results are x/z for A, y/z for B and 1=z/z for C. We now have spelled this out in the revised version.
> >
> > **(W5) Perplexity makes only sense in the autoregressive, so I shall assume that the authors meant the NLL.**
> >
> > We do not mean the NLL. We indeed mean perplexity, which is a general notion. Perplexity is defined as $e^{\frac{1}{L}H(p,q)}$ where $H(p,q)$ is the cross entropy between the true and learned distributions. Thus our formula $e^{\frac{B}{L}}$ bounds the perplexity. This is precisely the perplexity definition used in [8], [6], [9] and all relevant work in this field as well as in AR models.

---

> > > ### Author Response · Authors · 2025-11-19
> > >
> > > **(W6) As for section 4, upper bounds on the NLL exist in MDLM [6], for instance. Discrete flow matching, MDLM and other methods alike are essentially equivalent, and are all trained with the cross-entropy loss; it is quite easy to compute a bound on the test NLL, as the loss itself typically constitutes an (N)ELBO. I am not certain about the novelty; if the tightness is improved upon, it should have been somewhat motivated theoretically.**
> > >
> > > Section 4 extends the existing perplexity upper bound from MDLM [6] to the general class of discrete flow matching dynamics. MDLM corresponds to a very specific case of diffusion models, which themselves are a special case of discrete flows. Therefore, MDLM is an extremely restricted instance within the broader discrete-flow framework, and it is not equivalent to general discrete flow matching.  In fact the original paper [11] that introduced discrete flows relied on generative perplexity and downstream tasks to evaluate their models, as the masked diffusion bound is not transferable.
> > >
> > > Our contribution is precisely to show that the MDLM-style bound naturally extends to arbitrary discrete flow dynamics. For MDLM, our general bound reduces to the MDLM bound in [6]. Beyond that, we also derive an expression for the exact perplexity, which, while theoretically interesting, is unfortunately too computationally expensive to use in practice. Hopefully future research will be fruitful in this direction.
> > >
> > > While one can heuristically train flows with cross-entropy alone, this is not theoretically motivated in our general discrete-flow setting (and in fact is suboptimal as demonstrated in Section 5.2 and Appendix C.2). Our bounds are necessary to estimate the perplexity for flow types such as the bag-of-words and multimask flows studied in the paper, where the MDLM-specific bound and cross-entropy do not apply.
> > >
> > > In short: " MDLM and other methods alike are essentially equivalent" is incorrect.
> > >
> > > Further, the bounds in Equations (14) and (16) are not the cross entropy formula. In other words: the statement "and are all trained with the cross-entropy loss; it is quite easy to compute a bound on the test NLL, as *the loss itself typically constitutes an (N)ELBO*" is incorrect as well.
> > >
> > > **(W7) I am not certain how the tightness is proved in the experiments whatsoever.**
> > >
> > > We do not claim a general proof of tightness of the bound. What we do is:
> > >
> > > 1. derive the exact expression for the perplexity,
> > >
> > > 2. derive a bound by dropping a single term from this exact expression,
> > >
> > > 3. show that when the model learns the flow perfectly, this dropped term becomes zero,
> > >
> > > 4. empirically evaluate the tightness of the bound in simple settings where we can estimate the true perplexity very accurately.
> > >
> > > The percentage differences we observe between the bounded NLL (in this case we also measure NLL in order to compare with [10]) and the true NLL are similar to the NLL differences reported in Song et al. [10] for continuous diffusion in high dimensions, where the true NLL can be computed precisely.
> > >
> > > **(W8) In experiment 5.2, it is claimed that SEDD has an NLL of about 80, whereas in the original paper the authors get it down to about 33.**
> > >
> > > This is incorrect. The results we report for SEDD in Section 5.2 are:
> > >
> > > 52.18| 42.02| 117.00| 41.83| 80.79
> > >
> > > for the small model, which are nearly identical to those in Table 1 of the original SEDD paper [8] (whose exact architecture we replicate):
> > >
> > > 50.92| 41.84| 114.24| 40.62| 79.29
> > >
> > > So our numbers are in close agreement with the original paper for the same model size and setup.
> > >
> > > We should emphasize here that the reason we include such results is to show that masked flow and diffusion have similar performances.
> > >
> > >
> > > **(W9) Multimask Flows are never properly introduced, .... models are much lower in the other works.**
> > >
> > > We agree that an earlier introduction of multimask flows improves readability, and we have now revised the paper accordingly, introducing them before the experimental section, using the additional page allowed.
> > >
> > > Empirically, the performance gap between multimask flows and masked flows in terms of generative perplexity is substantial in our experiments. Regarding the claim “I *believe* that the generative perplexities for these models are much lower in the other works”, this seems to conflate settings with different network sizes, sequence lengths, training budgets, and other hyperparameters. To make a fair comparison, one must control for these factors.
> > >
> > > To the best of our knowledge, there is no previous work that, under the same architecture, sequence length, batch size, training steps, and other hyperparameters, demonstrates masked diffusion outperforming our multimask flows in terms of generative perplexity. Many recent works indeed report lower perplexities, but they typically use much larger/different models, more tokens, and longer contexts, and unchecked entropy which is not a controlled comparison. We controlled for such factors.

---

> ### Author Response · Authors · 2025-11-19
>
> **(Q1) What is the novelty in your discrete OT section?**
>
> We addressed this point in detail in **(W1)**.
>
>
> **(Q2) Why is OT for discrete settings important? As you have seen in your own experiments, it scales poorly for vocabularies larger than… 3. How do you expect to scale it to 50k tokens?**
>
> This is again not accurate. We are unsure what the reviewer means by “it scales poorly for vocabularies larger than 3.” Our results show the opposite: OT improves performance with only a small increase in compute time. The remark likely conflates our proof of concept in Section 5.1, where we deliberately used the Hamming distance to show that even a suboptimal metric reduces jump counts by up to 15% on a small-vocabulary toy setup. In that same section we also tested our approach on a vocalulary of 87, and notice significant benefits as well. Finally, in Section 5.3 we used the L2 metric over token embeddings and observe substantial gains. These experiments already ran at realistic scales: a vocabulary of 50,257 for the BoW setting and 100,514 for the multimask setting. In short, there is no need to speculate about scalability... our original submission already demonstrated it.
>
>
> **(Q3) You claim that similarity in the embeddings should be used to weigh the cost differently. What embeddings are you referring to? that of your untrained model? an auxiliary, pre-trained model?**
>
> This part appears to have been misunderstood by the reviewer and we are glad to clarify it. The $L_2$ metric that we use throughout Section 5.3. scales with vocabulary size, and is defined as:
> $\sum_{i=1}^L ||\text{emb}(x^i_0)-\text{emb}(x^i_1)||^2$, where $\text{emb}(x^i)$ are the vector embeddings of token $x^i$. The embeddings are simple initializations of tokens at the beginning and they train as the model trains, meaning they become more discriminative as time passes.
>
>
> **(Q4) Do we agree that you meant NLL and not perplexity?**
>
> No, as we discussed this in detail in (W5).
>
> **(Q5) For 5.1, do you have quality metrics interacting with the number of jumps?**
>
> Although this is not the focus of the section, we have now updated the paper to include the bound results as well.
>
> **(Q6) Why Multimask Flows allegedly work better? Could it just be that they have more parameters? Do they use (that is to say, generate) the special tokens introduced? If so, how? Or did I misunderstand the proposed method altogether?**
>
> We have now added another section (3.3) to explain the multi-masked flows in more details. The masked tokens are fed as normal tokens, and do not participate in the layers of the model itself. In addition, these parameters do not add compute and memory-wise they become negligible as the transformer scales. We believe that the reason for the improvement is that since there are many starting points, it creates the opportunity for the network to define a flow which sends a given sequence $x_0$ to a distribution $p(x_1|x_0;\theta)$ with smaller entropy. While for the masked one, $p(x_1|[M,M,...,M];\theta)$ should be the entire $p(x_1;\theta)$. Nonetheless, given there are no downsides to this approach, the empirically demonstrated (not alleged) improvements make this is a valuable contribution.
>
> **(Q7) What is novel in your new NLL bound?**
>
> We addressed this point in detail in **(W6)**.
>
>
> **(Q8) When do you use the bound in the experiments? At all times? for all methods? If you did for SEDD as well, for instance, then your bound is definitely not tighter.**
>
> Our bounds become the same as the one in [9] in the case of masked flows. We do not claim our bounds are tighter than in SEDD/masked-diffusion. Our main claim is that the bounds provided cover general discrete flows, not just the masked flow/diffusion. We use our bound in our own BoW and Multimasked flows whose perplexity otherwise we would not be able to measure (the masked diffusion bound does not apply here).
>
> We consistently use our first bound for evalution:
>
> For our experiments in Section 5.1, we *now* use the bound in Eq. 14.
>
> For experiments in section 5.2, when defininig the network to predict the same token at position $i$ as the one fed to it if that token is unmasked, then Equation 14 becomes:
> $\int_0^1 \frac{1}{1-t} \sum_{x_1, x_0 }\pi(x_1,x_0)\sum_{x_{t}}p_{t|1,0}(x_{t}|x_1,x_0) \sum_{i=1}^L -\delta_m(x_t^i)\log{p_{1|t}^i(x^i_1| x_{t};\theta)}dt.$ where $\delta_m(x_t^i)$ is 1 anytime $x_t^i$ is the mask token, and 0 otherwise.
>
> For multimaksed-experiments in section 5.3, when forcing the network to predict the same token at position $i$ as the one fed to it if that token is not one of the masked tokens, then Equation 14 becomes:
>
> $\int_0^1 \frac{1}{1-t} \sum_{x_1, x_0 }\pi(x_1,x_0)\sum_{x_{t}}p_{t|1,0}(x_{t}|x_1,x_0) \sum_{i=1}^L -\delta_\bar{m}(x_t^i)\log{p_{1|t}^i(x^i_1| x_{t};\theta)}dt.$ where $\delta_\bar{m}(x_t^i)$ is 1 anytime $x_t^i$ is one of the masks, and 0 otherwise.
>
> While for the BoW experiments in section 5.3, we use the bound in Eq.14.

---

> > ### Author Response · Authors · 2025-11-19
> >
> > Regarding training:
> >
> > In section 5.1 and section 5.3 we always use cross entropy loss for consistency.
> >
> > In section 5.2 we use cross entropy (DFM-O), our first bound (DFM-S), and the scaled bound (DFM-N) to compare and show that the performance differences the mirror those in masked diffusion.
> >
> >
> > **(Q9) Could you please expand on the claims about the tightness of your bounds?**
> >
> > We believe the reviewer refers to the claims regarding the case when the model matches perfectly the real flow. In this case, we have the following:
> >
> > The expressions of the perplexity bounds are derived by initially dropping the term
> > $$-\int_0^1 \sum_{x_t}\bar{q_t}(x_t) \sum_{i=1}^L \sum_{x^i\neq x_t^i} \left( \tilde{w}_t^i(x^i, x_t)\log{\frac{\tilde{w}_t^i(x^i, x_t)}{\tilde{v}_t^i(x^i, x_t)}}+ \tilde{v}_t^i(x^i, x_t) - \tilde{w}_t^i(x^i, x_t)\right)dt$$
> >
> > from the full expression of the KL divergence between the data and the learned distribution in Theorem 4.1. This term can be rewritten as *(by Proposition A.1)*
> >
> > $$-\int_0^1 \sum_{x_t}\bar{q_t}(x_t) \sum_{i=1}^L \sum_{x^i\neq x_t^i}\left(aw_t^i(x_t^i, x)\log{\frac{aw_t^i(x_t^i, x)}{bv_t^i(x_t^i, x)}}+ bv_t^i(x_t^i, x)  - aw_t^i(x_t^i, x)\right)dt,$$where $a = r_{\bar{q_t}}$ and $b = r_{\bar{p}_t}$.
> >
> >
> > If the model learns the flow perfectly, this term becomes zero. Indeed, if $w_t$ matches $v_t$, then the induced probabilities, and therefore the induced ratios match so $a=r_{\bar{q_t}}=r_{\bar{p_t}}=b$ implying
> >
> > $$-\int_0^1 \sum_{x_t}\bar{q_t}(x_t) \sum_{i=1}^L \sum_{x^i\neq x_t^i}\left(bv_t^i(x_t^i, x)\log{\frac{bv_t^i(x_t^i, x)}{bv_t^i(x_t^i, x)}}+  bv_t^i(x_t^i, x)  - bv_t^i(x_t^i, x)\right)dt.$$
> >
> > $$-\int_0^1 \sum_{x_t}\bar{q_t}(x_t) \sum_{i=1}^L \sum_{x^i\neq x_t^i} \left(bw_t^i(x_t^i, x)\log{1}+ bv_t^i(x_t^i, x) - bv_t^i(x_t^i, x)\right)dt=0.$$
> >
> > We would be happy to clarify further if something is unclear, or if the reviewer is referring to different claims.
> >
> > **(Q10) How do you estimate the minibatch OT? POT?**
> >
> >
> > Yes, in practice we used POT for minibatch OT. We named the method in Table 4, but we have now stated this even more explicitly.
> >
> >
> >
> > **Overall, feels like 3 different ideas put together but without the depth that they might all necessitate. The presentations certainly hinders the understanding of the results.**
> >
> > The paper does not present three unrelated ideas. It targets two components that are central in continuous flow matching and that were not present in the discrete flow framework [11]: first, a principled way to measure perplexity enabling theoretically backed training along with an evaluation metric, and second, a mechanism to rectify paths. These are not optional details; they are foundational tools for evaluating and improving flows. Our contributions supply these missing pieces for discrete flows and integrate them into a single framework. Theorem 3.1, Theorem 4.1, and Proposition 4.4 provide the main results that study these problems in depth and rectify them.
> >
> > *On OT experimental depth.* For brevity, we list only the main factors here. We ran nine experiments, seven at realistic vocabulary sizes from 50,257 to 100,514. We utilized two widely used training datasets, and evaluated on five test sets, used two metrics, different tokenizations, several flow types, and we varied vocabulary size. We studied compute scaling with respect to batch size and sequence length, and we note that with an L2 cost over token embeddings the cost is independent of vocabulary size. We also calculated the entropy of the generated texts to measure diversity and used two different LLMs as evaluators to assess model quality. Following the constructive suggestions of Reviewer i4qU, we have now added ablations that vary the metric, the solver including exact OT, and the batch size.
> >
> > *On the bounds' experimental depth.* We evaluated every flow introduced using our bound and reported the resulting perplexities in the appendix. This enables fair comparisons across discrete diffusion models, autoregressive models, and different discrete flow dynamics. Again due to the constructive comments of Reviewer i4qU, we have also now measured the correlation between our bound and generative perplexity and observed a very strong correlation with Pearson r = 0.96.
> >
> > Finally, multimask flows are not a third, disconnected idea. They directly arose from the need to extend OT to the strongest performing instance of flows, namely masked flows. Their performance is then assessed with our bound, which makes the comparison to masked flows possible. To conclude, all components belong to one framework, are interconnected and explored in depth.
> >
> >
> > **(Strength 1) The provided proofs seem mostly fine.**
> >
> > A mathematical proof unfortunately cannot be mostly fine. It either is correct or it is incorrect. If the reviewer has found any mistakes then we would be grateful for sharing them. If they are rectifiable during the rebuttal we will update the paper and maintain our submission.

---

> > > ### Author Response · Authors · 2025-11-19
> > >
> > > [7]  Tong et al. Improving and generalizing flow-based generative models with minibatch optimal transport. 2023
> > >
> > > [8]  Lou et al. Discrete Diffusion Modeling by Estimating the Ratios of the Data Distribution. 2024
> > >
> > > [9]  Shi et al. Simplified and Generalized Masked Diffusion for Discrete Data. 2024
> > >
> > > [10] Song et al. Maximum Likelihood Training of Score-Based Diffusion Models. 2021
> > >
> > > [11] Gat et al. Discrete Flow Matching. 2024

---

### Official Review · Reviewer_Nso4 · 2025-10-31

**Soundness:** 3
**Presentation:** 4
**Contribution:** 3
**Rating:** 6
**Confidence:** 4

**Summary:**

The paper advances the discrete flow matching framework for training and sampling of CTMC processes over discrete state spaces. The main contribution of the paper are two fold:

i) formulating dynamical optimal transport (OT) for discrete spaces along with empirical validation with mini-batch OT.

ii) proving two upper bounds on the perplexity.

Additionally, the authors propose a new multimask source distribution. This distribution augments the vocabulary with approximately 50k additional mask tokens (i.e., tokens that have zero probability under the target distribution), and a uniform probability is assigned to each token. Combing the mini-batch OT with multimask source results in improved generative perplexity.

**Strengths:**

1. The paper is very well written, mathematical theorem clear and well presented.

2. The paper formulate dynamical OT for discrete state spaces and prove the relation to  Kantorovich formulation. This allows training with  minibatch-OT.

3. Table 3 shows new and interesting results for minibatch-OT on generative perplexity.

**Weaknesses:**

1. in subsection 5.2 If the only difference between Table 2 and the table presented in [1] is the model trained with (1-t) time weighting then it is of low novelty compared to previous works.

2. As reported in the appendix, while multimask source gives an improvement in results of generative perplexity with minibatch-OT, it does hurt the perplexity of both with and without minibatch-OT (table 8).

3. The authors states that computing minibatch OT on a batch of 1000 samples adds approximately 3.4% to training cost. However, for larger batches the authors states it can increase training cost by 10-15% which might not be tolerable for large scale models.

4. The paper presents two bounds on perplexity in Equations (13) and (16). As acknowledged by the authors, these results overlap with prior work ([1–4]). While the paper provides a clear and unified exposition, the claimed novelty in enabling comparisons with autoregressive and discrete diffusion models seems somewhat limited given these existing results.

[1] Shi, Jiaxin, et al. "Simplified and generalized masked diffusion for discrete data." Advances in neural information processing systems 37 (2024): 103131-103167.

[2] Zheng, Kaiwen, et al. "Masked diffusion models are secretly time-agnostic masked models and exploit inaccurate categorical sampling." arXiv preprint arXiv:2409.02908 (2024).

[3] Shaul, Neta, et al. "Flow matching with general discrete paths: A kinetic-optimal perspective." arXiv preprint arXiv:2412.03487 (2024).

[4] Haxholli, Etrit, et al. "Efficient perplexity bound and ratio matching in discrete diffusion language models." arXiv preprint arXiv:2507.04341 (2025).

**Questions:**

1. In section 5.3, the minibatch-OT is evaluated. However the similarity metric $s(x_0^i, x_1^i)$ is not specified. Could the authors please provide this information?

2. The computational complexity of OT solvers typically scales quadratically or cubically with respect to the number of samples. In the paper, the authors mention that the OT solver increases the training cost by 10–15% for larger batch sizes. Could the authors elaborate on this observation? In particular, an estimation of the expected computational overhead for batch sizes comparable to those used in large-scale LLM training would be greatly appreciated.

---

> ### Author Response · Authors · 2025-11-19
>
> We thank the reviewer for their time and efforts invested in writing a high quality review that in our opinion correctly identifies the strengths and the potential concerns of the paper. We are glad that the reviewer finds the paper to be very well written as we did put extra effort on this aspect. We are also happy the reviewer recognizes and appreciates the extension of Kantorovich and dynamic formulation to discrete flows, as well as their Benamou-Brenier-like theorem between them. Finally we are grateful that the reviewer finds the experimental findings interesting. We address the concerns below:
>
>
> **(W1) In subsection 5.2 If the only difference between Table 2 and the table presented in [1] is the model trained with (1-t) time weighting then it is of low novelty compared to previous works.**
>
> Actually we do not consider the removal of (1-t) as novel. We added Section 5.2 for 2 main reasons:
> 1) We wanted to show some empirical proof that the bounds derived are the proper ones for discrete flow matching, as a significant number of readers in the field prefer to also see empirical results. We provide this empirical proof by showing that our bounds in discrete masked flow matching give roughly the same results as SEDD in discrete diffusion. At the same time this shows that we can use our bounds for comparing with AR models.
>
> 2) We wanted to check the difference between cross entropy of [1] and our scaled/unscaled bound within the discrete flow matching itself.
>
> 3) We wanted to show the differences of DFM-N and DFM-S mirror those reported for masked diffusion in [4], more precisely: CEDD, and CEDDT.
>
> We recognize this is not mentioned in the paper. We have used the additional page to explain this in more detail.
>
> **(W2) As reported in the appendix, while multimask source gives an improvement in results of generative perplexity with minibatch-OT, it does hurt the perplexity of both with and without minibatch-OT (table 8).**
>
>
> We again thank the reviewer for their careful reading. We are happy to clarify this point, as it is somewhat subtle in the text. At first glance, it may appear that the multimask source worsens perplexity with and without OT compared to the DFM-N results. However, throughout Section 5.3 we use cross-entropy as the training loss (as in DFM-O), in order to remain consistent with the BoW experiments. Thus, the relevant comparison is actually with DFM-O, and in that setting the perplexity is substantially improved. We have updated Table 8 in Appendix C3 to make this clearer. In all cases, we used the appropriate corresponding bounds when evaluating model perplexities.
>
> In addition, we observed that using the exact solver at test time slightly improves performance and, importantly, guarantees that no test points are repeated in a given batch (Table 9, Appendix C3).
>
>
> **(W3) The authors states that computing minibatch OT on a batch of 1000 samples adds approximately 3.4% to training cost. However, for larger batches the authors states it can increase training cost by 10-15% which might not be tolerable for large scale models.**
>
> We thank the reviewer for the comment. Based on the data of open LLMs with 70+ billion parameters, the vast majority of them use batches of 512, 1024 and 2048 sequence lengths. All of these values hover at <13% increase in compute time. In addition, at production level the performance of OT is likely to be optimized further. For example, it appears suboptimal that POT with GPU performs roughly the same for batch sizes 256, 512, 1024.

---

> > ### Author Response · Authors · 2025-11-19
> >
> > **(W4) The paper presents two bounds on perplexity in Equations (13) and (16). As acknowledged by the authors, these results overlap with prior work ([1–4]). While the paper provides a clear and unified exposition, the claimed novelty in enabling comparisons with autoregressive and discrete diffusion models seems somewhat limited given these existing results.**
> >
> > We are grateful for the opportunity to clarify this further: [1, 2] provide bounds on masked diffusion. [4] and [5] each provide a bound through ratios (not likelihoods) on general discrete diffusion models. Our first bound (Eqs. 13) extends that of [5] to discrete flow matching, while our second bound (Eqs. 16) extends the one in [4] to discrete flow matching. This enables us to estimate the perplexity on the flows we define: BoW and multimasked, which are not defined in discrete diffusion (and likely cannot due to the limited space of transition matrices in the discrete diffusion framework). In fact the original paper [6] that introduced discrete flows relied on generative perplexity and downstream tasks to evaluate their models, as the bounds from diffusion are not transferable.
> >
> > On the relation to [3], we agree that there is overlap between our first bound and that in [3]. We view [3] as concurrent independent work with respect to Eqs. (13): both this line of work and [3] were developed during late 2024. Nonetheless, our approach to reaching the bound is different: instead of taking an ELBO approach we followed the strategy of [5, 4] and studied the KL between the learned and real path distributions. This enabled us to derive a *precise* expression for the perplexity (by carrying the substitutions that provide Eq 13, to Theorem 4.1). However, one of the terms is computationally intractable. After dropping it, we end up with the bound as in [3]. We hope that future work can build on the precise perplexity formula to derive tighter bounds.
> >
> > **(Q1) In section 5.3, the minibatch-OT is evaluated. However the similarity metric $s(x^i,x_t^i)$ is not specified. Could the authors please provide this information?**
> >
> > Thank you for the question. The similarity measure we use in this section is $s(x^i,x_t^i)=\Vert e_m(x^i)-e_m(x_t^i)\Vert^2$ which then implies $c(x_0,x_1)=\sum_{i=1}^L \Vert e_m(x_1^i)-e_m(x_0^i)\Vert^2$ that is $c(x_0,x_1)=\Vert e_m(x_1)-e_m(x_0)\Vert^2$. We have now added this to the paper.
> >
> > **(Q2) The computational complexity of OT solvers typically scales quadratically or cubically with respect to the number of samples. In the paper, the authors mention that the OT solver increases the training cost by 10–15% for larger batch sizes. Could the authors elaborate on this observation? In particular, an estimation of the expected computational overhead for batch sizes comparable to those used in large-scale LLM training would be greatly appreciated.**
> >
> > This is an important question, and while we partially addressed it in (W3) we expand further here. For GPT3 the batch sizes are reported to vary between 245 and 1560. For PaLM 540B, 512, 1024 or 2048. For LLama3 405B they are reported to be 1000. Therefore we do not expect this overhead to increase more for modern large scale models than what we reported. In fact we believe it can be reduced with further production-level optimization.
> >
> >
> >
> > We thank again the reviewer for their time and careful evaluation. We hope to have addressed the stated concerns and are happy to discuss anything that has remained unclear.
> >
> > [5] Lou et al. Discrete Diffusion Modeling by Estimating the Ratios of the Data Distribution. 2024
> >
> > [6] Gat et al. Discrete Flow Matching. 2024

---

> > > ### Comment · Reviewer_Nso4 · 2025-11-27
> > >
> > > I thank the authors for the detailed response.
> > >
> > > I would appreciate if the authors could further clarify the definition of the similarity metric $s(x^i,x_t^i)=\Vert e_m(x^i)-e_m(x_t^i)\Vert^2$ , it is unclear from where the embeddings $e_m(x^i)$ are obtained?

---

> > > > ### Author Response · Authors · 2025-11-27
> > > >
> > > > We thank the reviewer for their reply and the additional effort invested in reviewing our work.
> > > >
> > > > Our approach employs learnable embeddings as in the non-OT case, that is, for a token with identifier $i$, its embedding corresponds to the $i$-th column of the learnable embedding matrix $E \in \mathbb{R}^{d \times V}$, where $d$ denotes the embedding dimension and $V$ the vocabulary size. During training, these embeddings progressively diverge from one another to capture the distinct semantic properties of different tokens, at which point the OT objective becomes increasingly relevant. We have updated the manuscript to include this fact.
> > > >
> > > > We really appreciate the reviewer's careful and thorough review.

---

### Meta-Review · Area_Chair_6Tpn · 2025-12-11

**Summary:**

Reviewers express very mixed feelings about the paper. I believe that most of the misunderstandings are caused by the initial writeup. The authors put a lot of effort in addressing the reviewer comments as best as they could, however, given the great deal and variety of different issues raised by the reviewers, the submission appears to leave too many open questions and my conclusion is that the paper is not matured enough for publication at ICLR yet.

**Reviewer Concerns:**

Most of the minor questions are likely settled by the rebuttal but there are simply so many comments, questions and remarks by the reviewers that their sheer number cannot be ignored. This is shown also in the ratings which contain many "fair" and "poor" marks. One of the reviewers actually refused to change the score after reading the rebuttal which is another indicator for me that the initial writeup was too far off from being included in the conference.

**Reviewer Scores:**

Again, I believe that the authors did the best they could to convince the reviewers but the amount of items and necessary changes of the submission is just too large.

---

### Decision · Program_Chairs · 2026-01-26

Reject